# Low pH-induced conformational change and dimerization of sortilin triggers endocytosed ligand release

Nadia Leloup [1], Philip Lössl [2], Dimphna H. Meijer[1], Martha Brennich [3], Albert J.R. Heck [2],
Dominique M.E. Thies-Weesie[4] & Bert J.C. Janssen [1]

Low pH-induced ligand release and receptor recycling are important steps for endocytosis. The transmembrane protein sortilin, a β-propeller containing endocytosis receptor, internalizes a diverse set of ligands with roles in cell differentiation and homeostasis. The molecular mechanisms of pH-mediated ligand release and sortilin recycling are unresolved. Here we present crystal structures that show the sortilin luminal segment (s-sortilin) undergoes a conformational change and dimerizes at low pH. The conformational change, within all three sortilin luminal domains, provides an altered surface and the dimers sterically shield a large interface while bringing the two s-sortilin C-termini into close proximity. Biophysical and cell-based assays show that members of two different ligand families, (pro) neurotrophins and neurotensin, preferentially bind the sortilin monomer. This indicates that sortilin dimerization and conformational change discharges ligands and triggers recycling. More generally, this work may reveal a double mechanism for low pH-induced ligand release by endocytosis receptors.

[1] Crystal and Structural Chemistry, Bijvoet Center for Biomolecular Research, Faculty of Science, Utrecht University, 3584 CH Utrecht, The Netherlands. [2] Biomolecular Mass Spectrometry & Proteomics and Netherlands Proteomics Center, Bijvoet Center for Biomolecular Research and Utrecht Institute for Pharmaceutical Sciences, Faculty of Science, Utrecht University, 3584 CH Utrecht, The Netherlands. [3] European Molecular Biology Laboratory, Grenoble Outstation, Grenoble 38000, France. [4] Van't Hoff Laboratory for Physical and Colloid Chemistry, Debye Institute for Nanomaterials Science, Faculty of Science, Utrecht University, 3584 CH Utrecht, The Netherlands. Correspondence and requests for materials should be addressed to B.J.C.J. (email: b.j.c.janssen@uu.nl)

Receptor-mediated endocytosis is an essential mechanism for eukaryotic cells. Ligands are bound extracellularly to endocytosis receptors, internalized and subsequently discharged from the receptors by low pH. The free receptors can then be recycled to the cell surface for new cycles of endocytosis. Different families of endocytosis receptors have been identified but our understanding of how their ligands are released at low pH after endocytosis remains incomplete. For the low-density lipoprotein (LDL) endocytosis receptor it has been shown that an intramolecular conformational rearrangement of domains discharges ligands[1] but whether such a conformational change mechanism applies to other endocytosis receptors is not clear[2].

Sortilin (Sort1, neurotensin receptor-3) is a type I transmembrane endocytosis receptor that has a multifunctional role in protein sorting and cell signaling[3, 4] in a diverse range of cell types such as neurons, hepatocytes and white blood cells[5]. Sortilin can trigger internalization of ligands from the cell surface via endocytosis and sorts ligands between several intracellular compartments, such as the trans-Golgi network (TGN), endosomes, lysosomes, and the secretory pathway[6–8]. For example the neuropeptide neurotensin that is implicated in hormonal and dopaminergic regulation[9] is internalized by sortilin[6]. In addition, sortilin signaling induced by binding proneurotrophins leads to neurodegeneration via the p75-sortilin-proneurotrophin complex, and neurotensin has been suggested to inhibit this pathway by preventing proneurotrophin binding to sortilin[10].

Sortilin binds and sorts a broad range of ligands, and dysfunction of sortilin has been linked to a wide range of disorders[4]. Upregulation of sortilin is a risk factor in cardiovascular diseases, such as hypercholesterolemia through its role in the clearance of low-density lipoprotein (LDL)[11] and secretion of Proprotein convertase subtilisin/kexin type 9 (PCSK9)[12], as well as obesity through the stimulation of fatty acid absorption via neurotensin signaling[13]. Sortilin is also associated with neurodegenerative diseases, such as Huntington's[14] and Alzheimer's disease[15] due to its involvement in the proneurotrophin pathway[16] and progranulin sorting[17].

The 49 residue-spanning C-terminal cytosolic tail of sortilin is required for endocytosis and shuttling between cellular compartments[18]. It harbors sorting motifs that can be recognized and bound by adaptor proteins such as the clathrin adaptors Golgi-localized, γ-ear containing, Arf binding proteins 1-3 (GGA1-3), thereby mediating sortilin endocytosis and shuttling.

The N-terminal luminal region of sortilin (s-sortilin) is N-linked glycosylated and essentially formed by the Vps10p domain. Previously solved crystal structures of s-sortilin revealed that the Vps10p domain in fact consists of three domains; a ten-bladed β-propeller and two 10CC domains (10CC-a and 10CC-b) that have substantial interactions with the β-propeller[19]. Sequence analysis indicates that these three domains are followed by a transmembrane helix and the aforementioned cytosolic tail that is predicted to be predominantly disordered. The available structures of human s-sortilin are either in complex with the 13 amino acid peptide neurotensin[19, 20] or small molecule inhibitors[21]. Each of the ten β-propeller blades consists of four anti-parallel β-strands with strand-linking loops forming the two propeller faces. The two 10CC domains are stabilized by disulfide bonds and interactions with the β-propeller. These structures reveal a rigid and compact conformation, in which the 10CC domains stabilize the 10-bladed β-propeller, and indicate that s-sortilin is a monomer.

The large surface formed by the sortilin luminal segment provides a platform for ligand binding. Sortilin interacts with a diverse set of ligands such as signaling receptors (e.g., Tropomyosin receptor kinase B, Epidermal Growth factor Receptor), enzymes (e.g., PCSK9), adaptor proteins (e.g., Apolipoprotein E[22] and sphingolipid adaptor proteins[8]) and signaling proteins, such as Sonic Hedgehog, (pro)neurotrophins, progranulin, and neurotensin. Sortilin also interacts with its own propeptide (called spadin or Sort-pro) that is generated by furin cleavage. It is not well understood what defines the promiscuity of sortilin for interaction with such a large collection of interacting partners but an important role for the large β-propeller has been suggested[19]. Neurotensin binds in the central tunnel formed by the β-propeller and competes for binding with (pro)neurotrophins and spadin[10, 19, 22]. Spadin, like neurotensin, competes with receptor-associated protein (RAP) for sortilin binding[22] and is believed to prevent ligand binding to sortilin in the TGN[22].

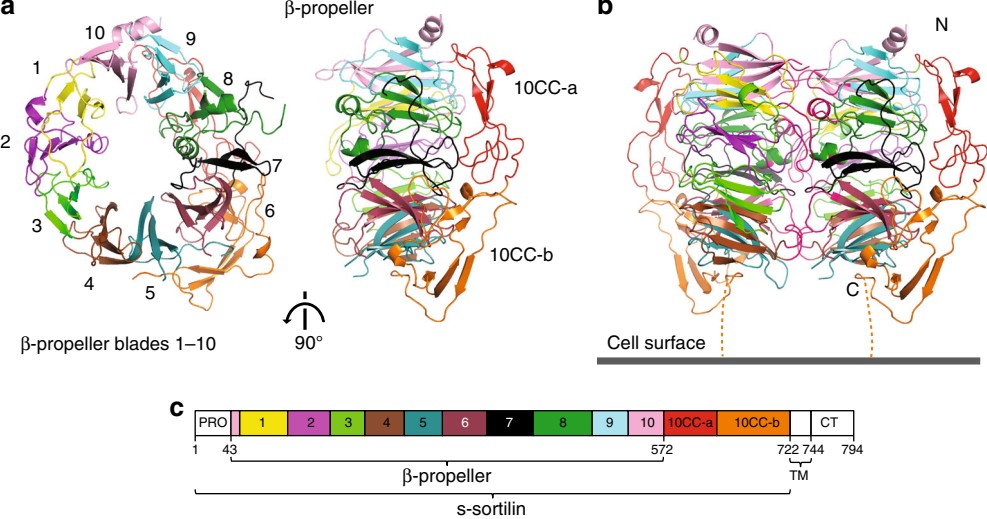

**Fig. 1** Crystal structures of s-sortilin at acidic pH reveal a dimer. **a** Cartoon representation of a single chain from the mouse s-sortilin dimer viewed from the top face of the β-propeller (left) and from the side (right), domains numbered and colored according to scheme C. **b** Side-view of the crystallographic s-sortilin dimer and its proposed orientation on the cell surface. Termini are indicated by N and C. The nine residues missing from the structure to the cell surface are indicated with a dotted line. **c** Schematic domain organization of mouse sortilin. PRO, prodomain; TM, transmembrane region; CT, cytosolic tail. The domains present in the structures are colored

**Table 1 Data collection and refinement statistics**

| Crystal form | 1 | 2 | 3 | 4 |
|---|---|---|---|---|
| *Data collection* | | | | |
| Space group | $P2_12_12_1$ | $P2_12_12_1$ | $P2_12_12_1$ | $C2$ |
| Cell dimensions | | | | |
| $a, b, c$ (Å) | 98.0, 132.3, 154.8 | 150.3, 151.8, 162.7 | 79.8, 137.2, 147.6 | 200.6, 62.6, 67.3 |
| $\alpha, \beta, \gamma$ (°) | 90, 90, 90 | 90, 90, 90 | 90, 90, 90 | 90, 101, 90 |
| Resolution (Å) | 69.64–2.30 (2.34–2.30)[a] | 71.72–4.00 (4.22–4.00)[a] | 62.20–3.21 (3.40–3.21)[a] | 65.99–2.10 (2.16–2.10)[a] |
| $R_{merge}$ | 0.096 (1.158) | 0.199 (1.506) | 0.073 (0.325) | 0.099 (0.499) |
| $I/\sigma I$ | 9.8 (1.4) | 7.7 (2.0) | 14.1 (4.8) | 6.3 (2.6) |
| Completeness (%) | 99.4 (98.7) | 100.0 (100.0) | 99.7 (99.6) | 97.5 (95.7) |
| Redundancy | 6.3 (6.0) | 12.0 (11.0) | 6.3 (6.4) | 3.1 (2.9) |
| $CC_{1/2}$ | 0.996 (0.432) | 0.998 (0.778) | 0.997 (0.964) | 0.985 (0.710) |
| *Refinement* | | | | |
| Resolution (Å) | 69.64–2.30 (2.34–2.30)[a] | 71.72–4.00 (4.22–4.00)[a] | 62.20–3.21 (3.40–3.21)[a] | 65.99–2.10 (2.16–2.10)[a] |
| No. of reflections | 87,424 (8568)[a] | 32,054 (3137)[a] | 27,057 (2655)[a] | 46,656 (4489)[a] |
| $R_{work}/R_{free}$ | 0.205/0.234 | 0.186/0.240 | 0.227/0.255 | 0.207/0.234 |
| No. of atoms | | | | |
| Protein | 10,313 | 20,403 | 10,237 | 4624 |
| Ligand/ion | 192 | 368 | 173 | 72 |
| Water | 159 | 0 | 0 | 226 |
| *B*-factors (Å²) | | | | |
| Protein | 63.1 | 199.1 | 114.2 | 37.9 |
| Ligand/ion | 86.3 | 227.6 | 133.8 | 51.8 |
| Water | 51.7 | n/a | NA | 38.9 |
| R.m.s. deviations | | | | |
| Bond lengths (Å) | 0.003 | 0.004 | 0.002 | 0.003 |
| Bond angles (°) | 0.633 | 0.745 | 0.686 | 1.39 |
| R.m.s.z. | | | | |
| Bond lengths | 0.27 | 0.30 | 0.24 | 0.25 |
| Bond angles | 0.47 | 0.48 | 0.43 | 0.45 |
| Ramachandran favored (%) | 96 | 95 | 94 | 96 |
| Ramachandran outliers (residuals) | 0 | 4 | 2 | 0 |
| Rotamer outliers (residuals) | 10 | 8 | 10 | 2 |
| MolProbity score | 1.48 | 1.60 | 1.74 | 1.11 |
| PDB ID | 5NMT | 5NNJ | 5NNI | 5NMR |

[a]Values in parentheses are for highest-resolution shell

Release of ligands from endocytosis receptors is generally believed to be induced by the increasing acidity within compartments along the endocytic pathway while going from early endosomes to late endosomes and lysosomes[23]. Indeed, for sortilin it has been shown that interaction with several ligands is lost at acidic pH[22, 24]. For example, the interaction between sortilin and RAP is lost at pH 4.0[24], that of spadin and Amyloid Precursor Protein interactions are substantially weakened at pH 5.0[22, 25] and binding of PSCK9 and the conotoxin-TxVI propeptide to sortilin is almost completely abrogated at pH 5.5[12, 26]. These ligands belong to different protein families that are structurally and functionally unrelated, and probably do not all bind to the same site on the sortilin surface.

How sortilin is capable of binding such a diverse set of ligands, different in size and structure, and release them at acidic pH is not clear. The structures available for s-sortilin do not inform on the molecular mechanism that underlies this discharging mechanism nor on the signal that triggers shuttling between cellular compartments. In this study, we detail that sortilin undergoes a conformational change within the three luminal domains and dimerizes while transitioning from neutral to acidic pH. This transition provides an altered sortilin structure and surface. We show that this double mechanism, of dimerization and conformational change, triggers the release of ligands representative of two common sortilin ligand families, the neuropeptide neurotensin and (pro)neurotrophins. This ligand discharging mechanism is strikingly different from that of the LDL

receptor and represents a release mechanism that could apply to a wide diversity of endocytosis receptors and ligands. In addition, the pH-induced dimerization brings the s-sortilin cytosolic segments in close proximity of each other which could provide the signal for cytosolic adaptor proteins to shuttle sortilin to various intracellular compartments. Our results indicate how discharging of receptor-bound ligands in the endosomes by pH-induced conformational change and dimerization can be recognized at the cytosolic side by adaptor proteins for subsequent regulation of receptor shuttling.

## Results

**Crystal structures of sortilin reveal a dimer at low pH**. We determined the structures of the glycosylated luminal segment of mouse sortilin, s-sortilin, from crystals grown at neutral pH (pH 7.5, one crystal form, 2.1 Å maximum resolution) and acidic pH (pH ranging from 5.0 to 6.2, three crystal forms, maximum resolution ranging from 2.3 to 4.0 Å) (Fig. 1, Table 1, Supplementary Fig. 1). The structures reveal two distinct conformations, a monomer at neutral pH and a dimer at acidic pH. The sortilin monomer adopts a nearly identical structure to that of human sortilin previously determined from crystals grown at pH 7.2–7.9[20, 21, 27] (Supplementary Fig. 1c) with a root mean square deviation (r.m.s.d.) of 0.67 Å for 586 Cα atom positions, of a total of 590 Cα atoms modeled (see Methods section for details). This is similar to the differences that are apparent when comparing the four human s-sortilin structures to each other. The eight

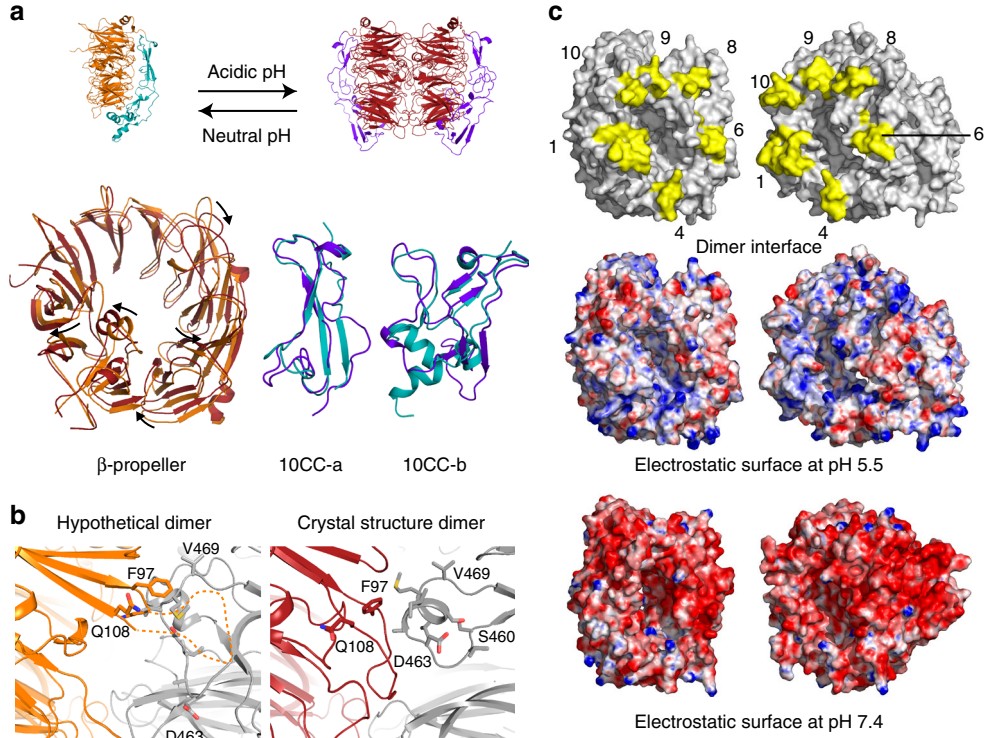

**Fig. 2** A conformational change accompanies the monomer-dimer transition. **a** Cartoon representation of the dimer (red) and monomer (orange, PDB 3F6K) conformations. Superpositions of the monomer and dimer conformation of each domains reveal substantial conformational changes within the domains (bottom). **b** A conformational change is required to prevent steric clashes. Monomer s-sortilin conformation as hypothetical dimer leads to clashes predominantly arising from loop 97–108 (left panel). This loop is flexible and not visible in the monomer mouse structure. It transitions to a structured form and provides interactions in the dimer (right panel). **c** Open book view of the dimer in surface representation (top) with the interface colored yellow. Blades involved in the dimerization are numbered. Below, gradient visualization from red (−10 kT/e) to blue (10 kT/e) of the electrostatic surfaces at pH 5.5 and 7.4, calculated with the APBS server[53, 54] based on residue pKa determined by PROPKA[28]

independent s-sortilin molecules in the three crystal forms grown at acidic pH all form homodimers with an identical interface and highly similar structure (Supplementary Fig. 1b). The r.m.s.d. between the s-sortilin chains in the dimers is 0.80 Å for all of the 650 Cα atoms modeled. S-sortilin dimerizes through the top face of the β-propeller, i.e., the large side of the β-propeller disk opposite to the bottom face at which the 10CC domains are located. This provides an outward and separated position for the 10CC domains in the two s-sortilin chains (Fig. 1b). The predominantly hydrophobic dimer interface is large with a buried surface area of 4882 Å² and is formed mainly by loops protruding from the blades (Fig. 1b, Fig. 2). β-Propeller blades 1, 4, 6, 7, 8, 9, and 10 are involved in dimer formation. The twofold symmetry axis that describes the s-sortilin homodimer passes through the dimer parallel to the dimerization interface and exits at blades 4 and 5 on one side and blades 9 and 10 on the other side of the dimer. As a consequence, the following β-propeller blades interact with each other across the dimerization interface: blade 1 interacts with blades 7 and 8 of the other chain, blade 4 interacts with blade 6 of the other chain, and blade 9 interacts with blade 10 of the other chain.

The s-sortilin dimer structures reveal how the full-length transmembrane sortilin could be oriented on the cell-surface (Fig. 1b). In the s-sortilin dimer the C-termini are in close proximity to each other with 37 Å distance between the two N713 Cα atoms. Each C-terminus has extensive interactions with its own β-propeller that limits their flexibility (Supplementary Fig. 2). The dimer crystal structures lack nine residues to the transmembrane helix. Most likely the twofold axis that describes the sortilin

dimer is oriented perpendicular to the cell surface. In this orientation, the sortilin β-propellers face the cell surface in a perpendicular fashion and the C-termini, the 10CC-b domains and β-propeller blades 4 and 5 are closest to the cell surface whereas blades 9 and 10 would be furthest away from it (Fig. 1b). Interestingly, the interface on the sortilin dimer that faces the cell-surface in this proposed orientation is lined by ten lysine residues that may aid surface adhesion by interacting with negatively charged glycolipids in the cell-membrane (Supplementary Fig. 2).

**Sortilin undergoes a conformational change**. The s-sortilin monomer-dimer transition is accompanied by a substantial conformational change (Fig. 2a, Supplementary Fig. 2, Supplementary Videos 1–4). The conformational change is an unusual rearrangement within the β-propeller and the two 10CC domains. No rigid body movement is observed, i.e., the positions of the domains do not change with respect to each other when comparing the s-sortilin monomer with a dimer chain (Supplementary Fig. 1a). In the dimer structure, the β-propellers are more compact and all blades are more evenly distributed around the center of the propeller. In the transition of the β-propeller from monomer to dimer, blade 1 moves in towards the center of the propeller and away from blade 2 (the blade 1—propeller-center distance decreases from 13.5 to 10.9 Å). Blades 8–10 are pushed away from the center of the propeller while blade 6 moves in and blades 3–5 remain relatively unperturbed. The distance between the centers of mass of opposing blades 1 and 6 thus increases from 39.0 Å in the monomer to 42.1 Å in the dimer, while all other opposing blades are further away from each other in the

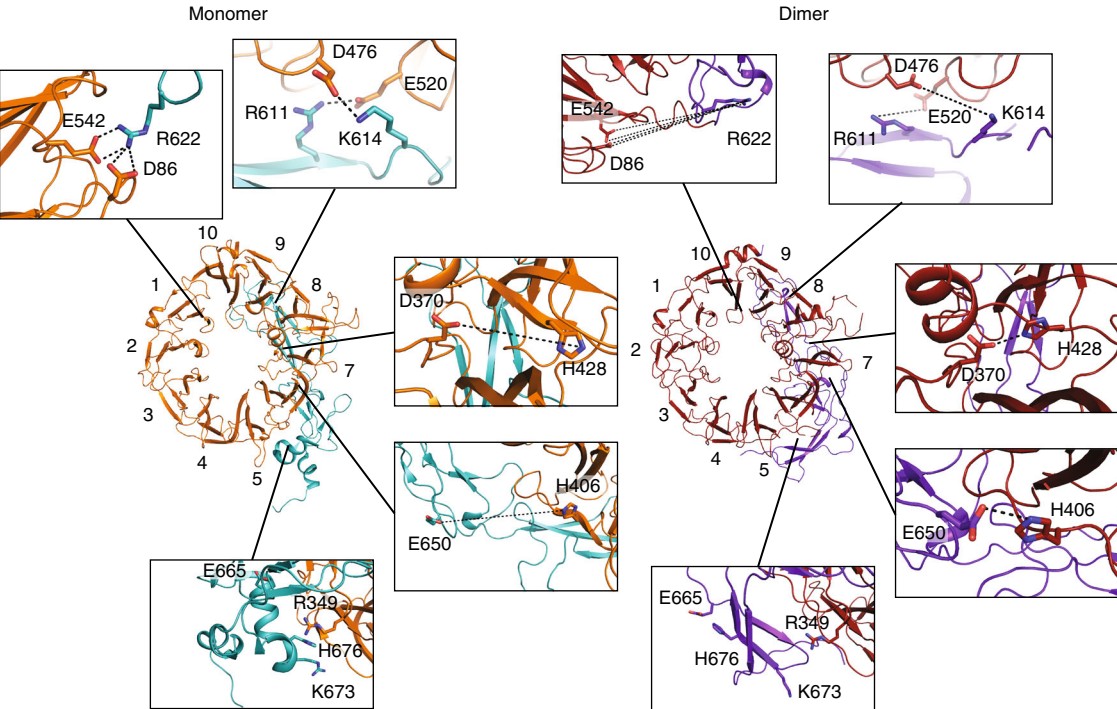

**Fig. 3** Different charge-dependent salt bridges stabilize the monomer and dimer conformation of s-sortilin. Monomer β-propeller represented in orange and 10CC domains (PDB 3F6K) in light blue; dimer β-propeller represented in red and 10CC domains in purple. Interacting residues are shown in sticks. Three salt bridges stabilize the monomer structure (two top panels at the left) while these interactions are broken and the involved residues have moved apart in the dimer structure (two top panels at the right) (distances >7.1 Å; middle two panels). On the other hand, two salt bridges in the dimer (two middle panels at the right) are disrupted and far apart in the monomer (two middle panels at the left). These new salt bridges in the dimer are enabled by the protonation of H406 and H428 at acidic pH. Interestingly, protonation of H676 (bottom panels) would lead to a coulombic clash between the β-propeller and the 10CC-b domain in the monomer structure (left), while after rearrangement H676 takes part in a charged interaction with E665, away from the β-propeller (right)

monomer compared to the dimer s-sortilin. The biggest blade center of mass distance decrease, from 46.4 Å in the monomer to 44.5 Å in the dimer, arises from opposing blades 4 and 9. This conformational change seems to be necessary for dimer formation as otherwise a few steric clashes would occur. The most predominant steric clash would arise from loop 97–107 in blade 1 of one dimer chain to loop 459–469 in blade 8 of the other chain (Fig. 2b). The loop 97–101 is flexible in the monomer and, together with associated β strands, undergoes a rearrangement to provide substantial hydrophobic interactions in the dimerization interface (Fig. 2b).

In addition to the rearrangements within the β-propeller also domains 10CC-a and 10CC-b undergo a conformational change. The core β-sheet in 10CC-a, formed by three β-strands in the monomer, has largely disappeared in the dimer structure. Nonetheless, both conformations, as adopted in the monomer and in the dimer, seem to be relatively rigid (Fig. 2a, Supplementary Fig. 1), most likely due to the stability provided by extensive interactions with the β-propeller. The 10CC-b domain in our mouse monomer structure is partially disordered but adopts a conformation similar to those in the human monomer s-sortilin structures (Supplementary Fig. 1). The analysis of the rearrangements within the 10CC-b domain is therefore based on comparison of the human monomer to the mouse dimer versions (89% sequence identity). Despite three internal disulfide bonds in 10CC-b, the three α-helices that form a small hydrophobic core in the s-sortilin monomer change their conformation into a three-stranded β-sheet in the dimer (Fig. 2a, Supplementary Fig. 1).

This conformational change also impacts the position of the sortilin C-terminus. In the monomer conformation, the s-sortilin C-terminus is exposed and pointing away from the β-propeller; whereas in the dimer conformation, it is interacting, through conserved hydrophobic interactions, with the rim of the β-propeller at blades 4 and 5 (Supplementary Fig. 2). This change in positioning of the C-terminus during the monomer to dimer transition may be important for the shuttling of sortilin between cellular compartments since it would bring the cytosolic domains of the dimer into close proximity (Supplementary Fig. 2).

**pH-dependent interactions**. The sequence of s-sortilin contains a total of 66 negatively charged residues (combined glutamic and aspartic acids) against 50 positively charged residues (combined arginines and lysines). Most of these charged residues are exposed, so at neutral pH the surface of sortilin is predicted to be negatively charged on patches scattered on the ß-propeller top face (residues 313–345, 366–368, 386–387, 397–403, 461–475, and 518–542) (Fig. 2c). This likely results in Coulombic repulsion at neutral pH and prevents dimerization (besides the steric clashes described earlier). At pH 5.5 the dimerization interface of s-sortilin is predicted to become neutral, likely aiding the dimerization (Fig. 2c). Five histidines (residues H182, H239, H406, H428, and H506) are predicted to change from neutral at pH 7.4 to positively charged at pH 5.5, and three aspartic acids (residues D333, D399, and D476) as well as seven glutamic acids (residues E154, E405, E448, E496, E520, E442, and E638), are predicted to change from negatively charged at pH 7.4 to neutral at pH 5.5. It is worthwhile to note that the negative patch present on 10CC-a

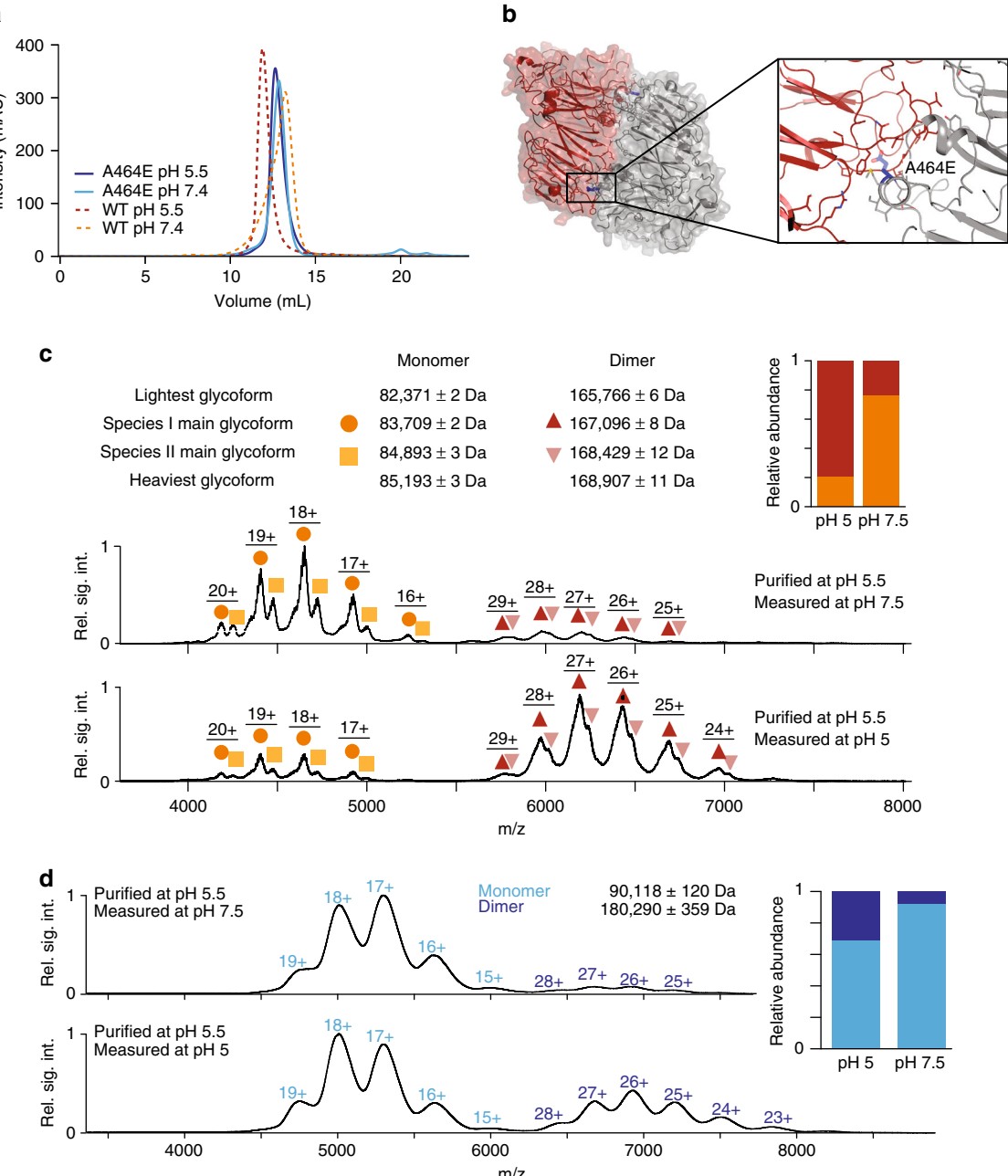

**Fig. 4** Wt s-sortilin is in a pH-dependent dimerization equilibrium while dimerization interface mutant A464E is predominantly a monomer at acidic pH. **a** Superose 6 size exclusion chromatography traces show that wt s-sortilin appears larger (smaller retention volume) at pH 5.5 (red) than at pH 7.4 (orange), while there are no difference in retention volume between A464E at pH 5.5 (dark blue) and pH 7.4 (light blue). **b** Based on the s-sortilin dimer structure mutant A464E (stick representation, blue) was constructed to prevent dimer formation. The A464E mutation prevents hydrophobic interactions by introducing a charged residue. **c** Native MS of wt s-sortilin produced in HEK293-ES cells (resulting in shorter, more homogeneous oligomannose glycans) shows that at pH 5, s-sortilin is mainly present as a dimer, while at pH 7.5, s-sortilin is predominantly a monomer (wt s-sortilin produced in HEK293-E cells, resulting in larger, less homogeneous hybrid glycans, gives a similar result, Supplementary Fig. 5b). **d** Native MS of s-sortilin A464E produced in HEK293-E cells shows that both at pH 5 and pH 7.5, s-sortilin A464E is predominantly a monomer

at residues 580–596 that is not involved in the dimerization interface, is predicted to remain negatively charged at acidic pH. The changes in surface charge explain the propensity of sortilin to transition from a monomer at neutral pH to a dimer at acidic pH.

The conformational rearrangement and pH-induced charge transitions allow disruption and formation of salt bridges. For example, within the β-propeller, at pH 7.4 residue H428 in the monomer is predicted to be neutral[28] and positioned away from

D370, at pH 5.5 H428 in the dimer is predicted to be positively charged and now forms a buried salt bridge with D370 (Fig. 3, Supplementary Fig. 3). Also in the interface between the β-propeller and the 10CC domains charge-related conformational transitions occur. At neutral pH, salt bridges between E542/D86 and R622 as well as E520-R611 and D476-K614 stabilize the interface between 10CC-a and the propeller (Fig. 3, Supplementary Fig. 3). At acidic pH, E542, E520 and D476 are predicted to

**Table 2 SE-AUC parameters**

| Sample | Glycans | pH | Model | Molecular weight | Molecular Weight (Da) | $K_D$ ($\mu$M) | Global reduced $\chi^2$ |
|---|---|---|---|---|---|---|---|
| wt s-sortilin | deglycosylated | 5.5 | monomer-dimer | fixed | 79400 | 0.5 | 1.54 |
| wt s-sortilin | deglycosylated | 7.4 | monomer-dimer | fixed | 79400 | 841 | 0.97 |
| wt s-sortilin | native[a] | 5.5 | single species | floated | 162578 | n.a. | 1.25 |
| wt s-sortilin | native[a] | 5.5 | single species | fixed | 180200 | n.a. | 2.96 |
| wt s-sortilin | native[a] | 7.4 | monomer-dimer | fixed | 90100 | 4 | 1.21 |
| wt s-sortilin, 2 x neurotensin | native[a] | 7.4 | monomer-dimer | fixed | 93500 | 259 | 1.71 |
| A464E s-sortilin | native[a] | 5.5 | single species | floated | 123704 | n.a. | 1.99 |
| A464E s-sortilin | native[a] | 7.4 | single species | floated | 101914 | n.a. | 1.97 |

[a] produced in HEK293-E cells

be neutral[28], and in the dimer structure these residues are positioned away from each other at distances larger than 7.1 Å (Fig. 3, Supplementary Fig. 3). H406 that is predicted to be neutral[28] at pH 7.4 and positively charged at pH 5.5, forms a new salt bridge (with E650 in 10CC-b) in the dimer structure. The substantial rearrangement within 10CC-b also prevents Coulombic repulsion at acidic pH between H676 (though not predicted to be positively charged at acidic pH) and R349 or K673, as H676 instead forms a salt bridge with E665 at neutral pH (Fig. 3, Supplementary Fig. 3). These observations indicate that the large-scale pH-induced structural rearrangement in s-sortilin may be the result of local, residue level, changes in charge.

**Sortilin is in a pH-dependent dimerization equilibrium**. We verified that s-sortilin undergoes a reversible, pH-induced, monomer-dimer transition in solution. At pH 5.5, the monomer-dimer equilibrium is shifted toward dimer; at similar concentration, the size exclusion chromatography (SEC) retention volume of s-sortilin at pH 5.5 is decreased compared to its retention volume at pH 7.4 (Fig. 4a). The weight average masses, as determined by multi-angle light scattering (MALS), are $84 \pm 2$ kDa for the SEC peak at pH 7.4 and $149 \pm 7$ kDa at pH 5.0 (Supplementary Fig. 4). Native MS analysis, which retains non-covalent interactions in the gas phase[29], shows that at similar s-sortilin concentrations, the dimer is the minor species at pH 7.5 but the predominant species at pH 5.0 (Fig. 4c) and that the s-sortilin monomer has a mass of $89.6 \pm 0.2$ kDa and the dimer a mass of $179.0 \pm 0.2$ kDa (Supplementary Fig. 5). Note that the differences in mass between the s-sortilin presented in Fig. 4c and Supplementary Fig. 4 are from differences in N-linked glycosylation states that are arising from the version of HEK293 cells used to produce the s-sortilin. S-sortilin used for the MALS experiments (Supplementary Fig. 4) is identical to that of the native mass spectrometry experiment in Supplementary Fig. 3. These data indicate that at acidic pH s-sortilin has more propensity to dimerize (Fig. 4a).

We quantified the affinity of s-sortilin dimerization at two different pH conditions with sedimentation equilibrium analytical ultra centrifugation (SE-AUC). For each sample a global analysis was performed at three concentrations, three centrifugal speeds and two wavelengths. Deglycosylated s-sortilin (Supplementary Fig. 5), displays a three orders of magnitude difference in the $K_D$ of dimerization; $8.4 \times 10^2$ $\mu$M at pH 7.4 and 0.5 $\mu$M at pH 5.5 (Table 2, Supplementary Fig. 6). The fit of the data to a monomer-dimer equilibrium model indicates that dimer formation is reversible and concentration dependent. SE-AUC data for glycosylated s-sortilin could not be modeled with a monomer-dimer model at pH 5.5 but the mass calculated with a single

species model indicates that most of the s-sortilin is in a dimer conformation (Table 2). At pH 7.4 the glycosylated sample has a $K_D$ of dimerization of 4 $\mu$M, which is a substantially higher affinity than that of deglycosylated s-sortilin, and suggests that N-linked glycans stabilize the s-sortilin dimer. Using s-sortilin proteolytic digestion followed by LC–MS/MS-based peptide analysis, we identified five N-linked glycan sites (N129, N241, N373, N549, and N651) and one O-glycan at position T715 (Supplementary Tables 1, 2, Supplementary Fig. 7). Of these residues, only N241 is near the dimerization interface. The glycan on this residue is not resolved in any of our crystal structures and, due to very low expression levels we have not been able to produce a N241Q glycan mutant for further experiments. Limited s-sortilin secretion due to inhibition of N-linked glycosylation has been observed by others before, indicating that some of the glycans are required for proper folding and/or intracellular transport of the receptor[20]. Taken together these results reveal that s-sortilin dimerizes more readily at acidic pH compared to neutral pH and that N-linked glycans might help s-sortilin dimerization.

**Interface mutant A464E disrupts dimerization of s-sortilin**. To validate the dimer interface observed in the crystal structures, we generated an interface mutant, s-sortilin A464E. This mutant is expected to interfere with dimerization but not with folding (Fig. 4). The large and negatively charged glutamate is likely to disturb a hydrophobic pocket in the dimerization interface (Fig. 4b). S-sortilin A464E has similar size exclusion retention volumes at pH 7.4 and 5.5. These curves fall in-between the monomer and dimer wild-type (wt) s-sortilin, but are closest to s-sortilin wt at pH 7.4 (Fig. 4a). Native MS shows some dimers of s-sortilin A464E are present at both pH conditions but much less compared to wt s-sortilin (Fig. 4c, d). We attempted to quantify the s-sortilin dimerization affinity with SE-AUC at pH 5.5 and pH 7.4 (Table 2). It is however not possible to fit the data to a monomer-dimer equilibrium model. The fit to a single species model indicates an average mass for s-sortilin A464E of 102 kDa at pH 7.4 and 124 kDa at pH 5.5. In comparison, an identical analysis for wt s-sortilin at pH 5.5 indicates an average mass of 163 kDa, much closer to the expected value of 180 kDa. Taken together, these data show that, as predicted, s-sortilin A464E forms less dimers compared to wt s-sortilin but that the dimerization properties are not completely disrupted by the mutation.

**S-sortilin shape depends on pH**. To characterize the solution structure of s-sortilin and s-sortilin A464E at different pH, we performed SEC-SAXS experiments of both constructs at pH 7.4 and pH 5.5 (Fig. 5, Supplementary Fig. 8). For wt s-sortilin, at pH

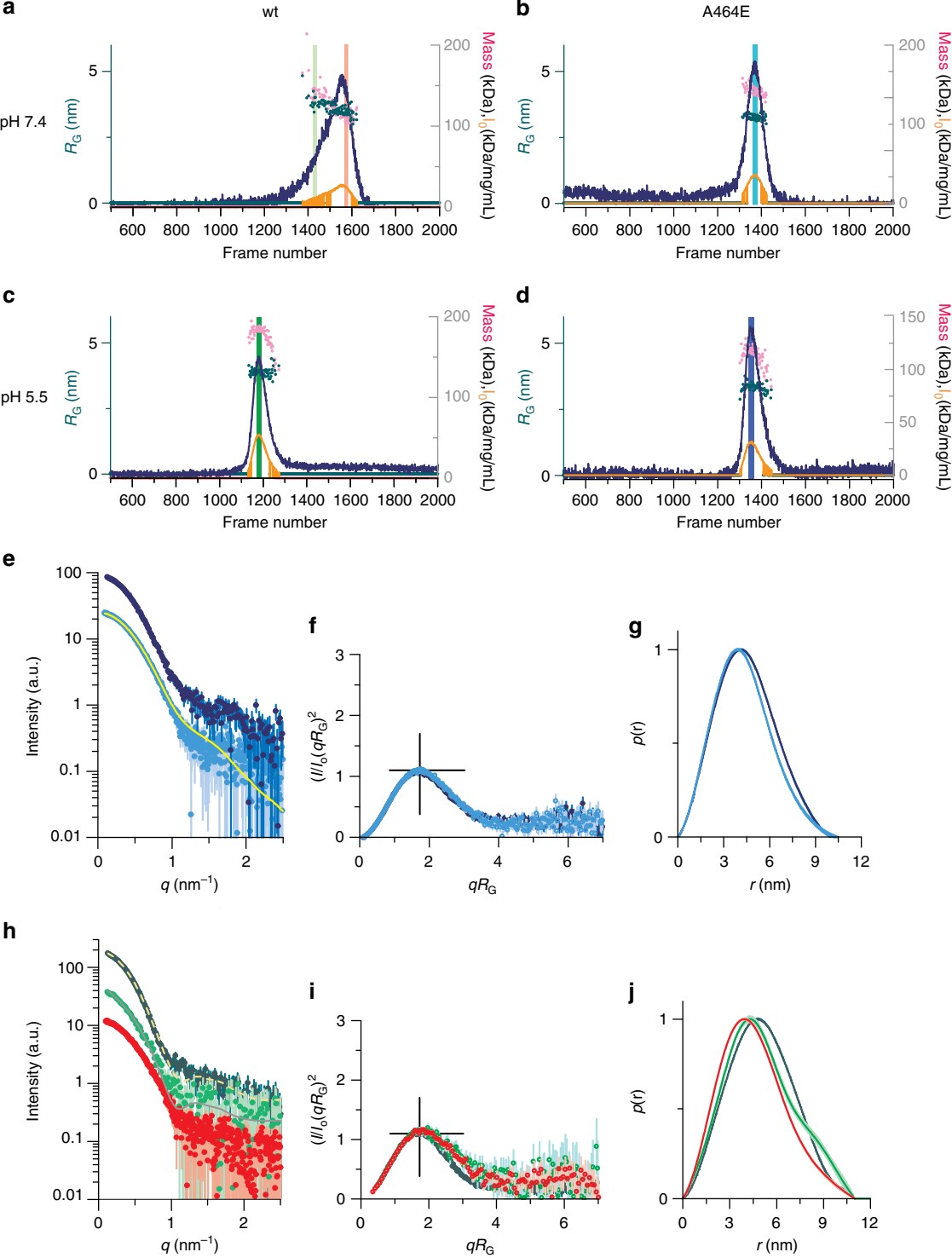

**Fig. 5** SEC-SAXS allows identification of different solution states of s-sortilin. **a–d** SEC-SAXS chromatograms of wt s-sortilin **a**, **c** and s-sortilin A464E **b**, **d** at pH 7.4 **a**, **b** and 5.5 **c**, **d**. The blue lines denote the total summed scattering intensity, orange lines the forward scattering intensity, pink dots the estimated mass based on the correlated volume and green dots the radius of gyration. The boxes denote regions used for subsequent analysis. Light green: dimer at pH 7.4, dark green: dimer at pH 7.4, red and light blue: monomer at pH 7.4, dark blue: monomer. **e–g** The shape of the s-sortilin monomer is pH dependent. **e** SAXS curves of s-sortilin A464E at pH 7.4 (light blue, yellow line: coral fit) and pH 5.5 (dark blue). The curves are shifted by an arbitrary offset for better comparison. **f** Corresponding normalized Kratky plots. **g** Corresponding pair distance distribution functions. **h–j** The shape of the s-sortilin dimer is pH dependent. **h** SAXS curves of wt s-sortilin at pH 7.4 in monomeric (red) and dimeric state (light green, gray line: CORAL fit) and pH 5.5 (dark green, yellow line: CORAL fit). The curves are shifted by an arbitrary offset for better comparison. **i** Corresponding normalized Kratky plots. **j** Corresponding pair distance distribution functions

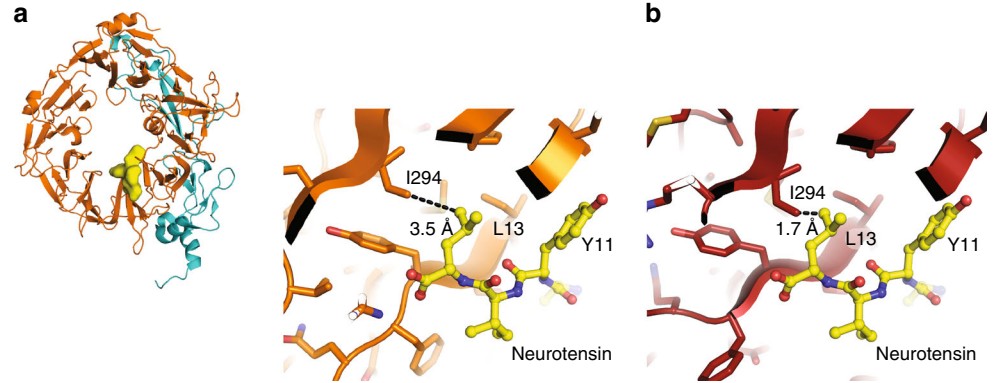

**Fig. 6** The conformational change accompanying the dimerization would trigger release of neurotensin. **a** Neurotensin (sticks, yellow) bound to monomer s-sortilin (orange) based on the neurotensin-s-sortilin complex (PDB 3F6K)[19]. **b** Neurotensin modeled on s-sortilin dimer (red) by superposing the monomer-neurotensin complex onto the dimer based on blade 6. In this model L13 and Y11 of neurotensin are clashing with I294 and β-strand 1 of blade 6 of the s-sortilin dimer, respectively.

7.4, the elution peak is very broad and asymmetric. The SAXS signal varies significantly throughout the peak and both the radius of gyration and the estimated mass decrease at higher retention volumes. All other runs, i.e., s-sortilin at pH 5.5 and of s-sortilin A464E at pH 7.4 and pH 5.5, exhibit relatively narrow and more homogeneous peaks with stable scattering signal throughout the peak. Taking into account the previous mentioned observations of reversible and pH regulated s-sortilin dimerization, in particular from AUC and MALS experiments, and the relative positions of the peaks, the peak of wt s-sortilin at pH 5.5 most likely represents s-sortilin dimer and those of s-sortilin A464E at both pH values s-sortilin monomer. Based on the protein concentrations and forward scattering intensities at the top of the peak, the apparent molecular weight of the dimer at pH 5.5 is $168 \pm 17$ kDa and that of monomer s-Sortilin A464E is $98 \pm 10$ kDa at pH 5.5 and $86 \pm 9$ kDa at pH 7.4 (Supplementary Table 3). The high molecular weight shoulder peak of wt s-sortilin at pH 7.4 most likely represents s-sortilin dimer although we cannot exclude some monomer is present (Supplementary Fig. 8d). The scattering curves corresponding to the s-sortilin wt and A464E mutant monomer at pH 7.4 match, implying that the suppression of dimerization does not affect the shape of the monomer. The SAXS data indicate that the shapes of the s-sortilin dimers at pH 7.4 and pH 5.5 are not identical nor are the shapes of the s-sortilin monomers at pH 7.4 and pH 5.5 (note that we observe s-sortilin monomer at pH 5.5 for the A464E mutant but not for the wt s-sortilin). This either suggests that s-sortilin can adopt four different conformations, with the monomer and dimer conformations at pH 7.4 different to the monomer and dimer conformations at pH 5.5 or, alternatively, that the s-sortilin dimer at pH 7.4 contains some monomer and the A464E monomer at pH 5.5 contains some dimer. The A464E pH 5.5 monomer SAXS data can be described reasonably well as a mixture of 77% A464E pH 7.4 monomer scattering plus 23% wt pH 5.5 dimer scattering ($p = 0.0074$, A464E pH 7.4 monomer scattering plus 61% wt pH 5.5 dimer scattering ($p = 0.007$, The radii of gyration of both the monomer and the dimer are larger at pH 5.5 than at pH 7.4 ($3.38 \pm 0.02$ nm vs. $3.26 \pm 0.03$ nm and $3.89 \pm 0.01$ nm vs. $3.69 \pm 0.05$ nm) (Supplementary Table 3). The normalized Kratky plots (Fig. 5f, j) all display one symmetric peak, but do not completely return to zero. This shape matches well with a generally globular protein with some local flexibility, e.g., arising from flexible glycans; we have identified five N-linked glycan in total on each s-sortilin chain and these glycans are heterogeneous and flexible. In contrast to the radius of gyration,

the maximum distance $D_{max}$ seems to be unaffected by either dimerization or pH change and stays constant at about 10.5 nm (Fig. 5g, k, Supplementary Table 3). However, the position of the peak of the distribution shifts to larger distances at pH 5.5 (3.89–4.10 nm for the monomer and 4.26–4.81 nm for the dimer), in accordance with the increase in radius of gyration.

The changes in the scattering between the monomer at pH 7.4 and the monomer at pH 5.5 are larger than the changes in the theoretical curves calculated from the monomer crystal structure and one chain of the dimer. In addition, the observed scattering curves do not match the predicted curves of the monomer or the dimer crystal structures (Supplementary Fig. 8c). Possibly these differences can be explained by differences in flexibility of the C-terminal 10CC-a and 10CC-b domains (Supplementary Fig. 8) or, at least partly, by the presence of monomer in the s-sortilin wt pH 5.5 shoulder fraction.

**S-sortilin homodimerization prevents ligand binding**. We explored the role of s-sortilin dimerization in modulating ligand binding. The high-affinity neurotensin binding site is located in the tunnel of the s-sortilin β-propeller[19] and mainly involves loop 317–320 and R292 with some contribution from residues K227, F273, G274, F281, S283, and I294 (Fig. 6). This site is not at the s-sortilin dimerization interface, but the neurotensin binding site does undergo a conformational change in the monomer-dimer transition, which would prohibit binding of neurotensin to the dimer conformation (Fig. 6). Indeed, neurotensin drives the monomer-dimer equilibrium at pH 7.4 toward the monomeric form: SE-AUC data show that s-sortilin dimerization at pH 7.4 is reduced 65-fold in the presence of neurotensin ($K_D$ of dimerization shifts from $4 \mu M$ for s-sortilin to $2.6 \times 10^2 \mu M$ for s-sortilin with neurotensin) (Table 2). Furthermore, in SEC-SAXS addition of neurotensin narrows the peak by reducing the size of the shoulder stemming from the s-sortilin dimer (Supplementary Fig. 9). The shape of the monomer does not change (Supplementary Fig. 9). Thus, neurotensin stabilizes the monomer form of s-sortilin and prevents dimer formation. It can be inferred from this data that pH-induced sortilin conformational change and dimerization will trigger release of neurotensin from sortilin.

We tested the propensity of three members of a second sortilin ligand family, the (pro)neurotrophins, to interact with wt s-sortilin and the monomer mutant A464E at neutral and acidic pH using Surface Plasmon resonance (SPR) (Supplementary Fig. 10, Supplementary Table 4). At neutral pH, the A464E mutation does not impact the binding of proBrain-Derived Neurotrophic Factor

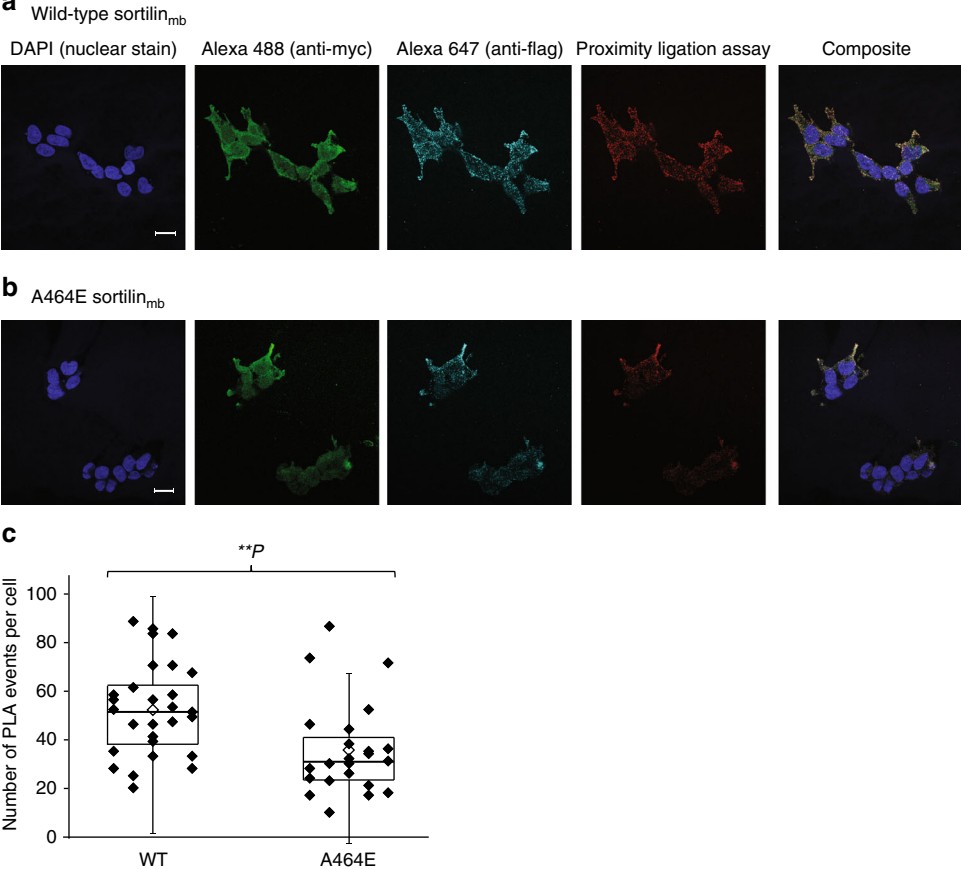

**Fig. 7** Sortilin dimerizes on the cell surface. **a** Analysis of the interaction of wt sortilin$_{mb-myc}$ with sortilin$_{mb-flag}$ by Duolink PLA in adherent HEK cells that co-express wt sortilin$_{mb-myc}$ and sortilin$_{mb-flag}$. Flag-tagged proteins were detected with mouse anti-flag antibodies (in cyan) and myc-tagged proteins with rabbit anti-myc antibodies (in green). Protein interactions were detected with Duolink PLA labeled in red. Each red spot is regarded as a single interaction. Images were collected by confocal microscopy. Scalebars, 20 μm. **b** Similar to A but now with sortilin A464E. **c** Distribution of the number of PLA events per cell (N > 22) expressing wt or mutant A464E sortilin$_{mb-myc}$ and sortilin$_{mb-flag}$. The upper and lower quartiles of each sample are represented by the upper and lower sides of the boxes; the medians are represented by the black horizontal lines, and the means by hollow diamonds. The range of the whiskers indicate the statistical outliers with a coefficient 1.5. Mann–Whitney test was used, **$p < 0.01$

(proBDNF) to s-sortilin, as indicated by the similar affinities of proBDNF for wt s-sortilin (0.31 μM) and s-sortilin A464E (0.40 μM). On the other hand, Nerve Growth Factor (NGF) and proNGF have less affinity for s-sortilin A464E (0.32 and 0.81 μM for NGF and proNGF, respectively) compared to wt s-sortilin (0.06 and 0.28 μM for NGF and proNGF, respectively). This indicates that this mutation interferes with NGF and proNGF binding. In addition, it points to a difference in binding specificity between NGF or proNGF and proBDNF. The affinity of NGF and proNGF for sortilin has been determined by others but differed substantially; 0.09 μM (NGF) and 5 nM (proNGF)[30] and 8 μM (NGF) and 0.77 μM (proNGF)[10]. The value that we determined for NGF-sortilin interaction agrees with that of Nykjaer et al.[30] and that of proNGF with sortilin is similar to that of Feng et al.[10] but in contrast to these earlier reports we find that NGF interacts with higher affinity to s-sortilin than proNGF does. The differences in affinity may come from differences in protein origin. We expressed and purified proNGF in HEK293 cells compared to proNGF produced in *Escherichia Coli*[30] and Sf9 cells[10], and we used mouse proteins instead of the human versions. Unfortunately, the affinity of wt s-sortilin to (pro) neurotrophins in acidic conditions could not be determined due to nonspecific binding of wt s-sortilin to the SPR sensor surface. The A464E mutant however, had substantially less nonspecific

binding and showed that (pro)neurotrophins are able to interact with s-sortilin A464E albeit with weakened affinity at pH 5.0 compared to pH 7.4 (Supplementary Table 4). Binding of NGF and proNGF to s-sortilin A464E at acidic conditions was reduced by a factor of 2 compared to neutral pH while binding of proBDNF was reduced by a factor 4. The somewhat weakened ligand affinity for s-sortilin A464E at pH 5.0 compared to pH 7.4 may be due to the conformational change, the remaining albeit much reduced dimerization propensity, or both properties of s-sortilin A464E. Taken together, our data in combination with that of others that show drastically reduced wt sortilin ligand interactions at acidic pH[12, 22, 24, 26, 31] indicate that dimerization and conformational change of sortilin at acidic pH prevents ligand binding.

**Membrane bound sortilin dimerizes in cells**. We tested the ability of wt and A464E mutant versions of cell membrane bound sortilin, sortilin$_{mb}$ (containing a human GPA33 transmembrane α-helix and lacking the cytosolic segment) to dimerize in cells using an in situ proximity ligation assay (PLA)[32] in adherent HEK cells (Fig. 7). We omitted the sortilin cytosolic tail in the sortilin$_{mb}$ construct to exclude dimerization effects arising from the cytosolic tail and to limit internalization with concomitant acidic pH-induced dimerization[33]. PLA events will only be

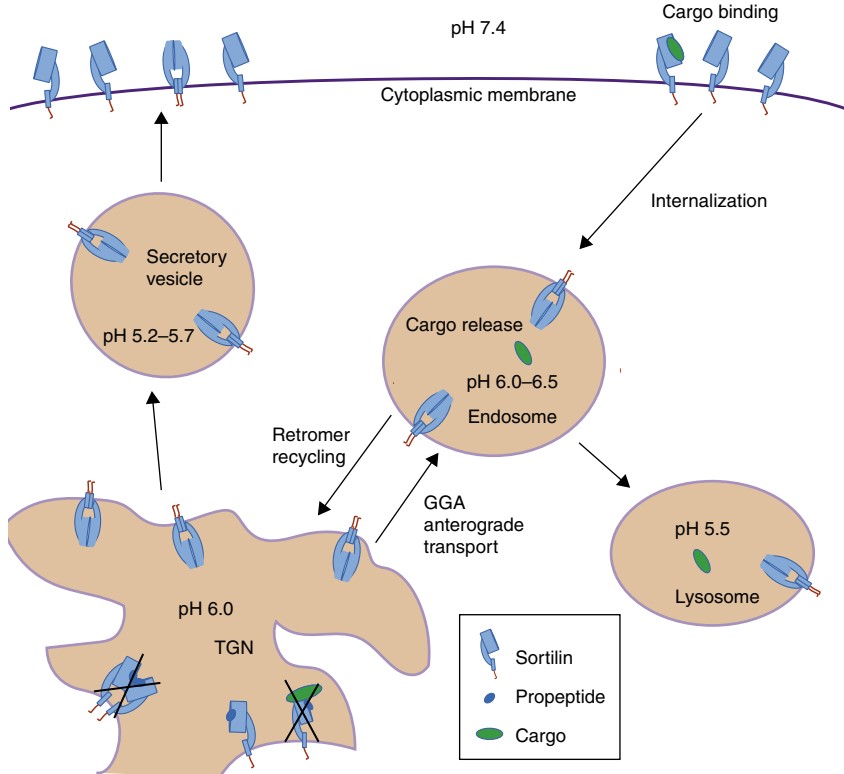

**Fig. 8** Proposed role of dimerization for sortilin function. After expression, sortilin is a monomer due to the presence of its propeptide. Once the pH drops in the late TGN, the propeptide is released and sortilin is transported to the cell surface or the endosome as a dimer. At the cell surface, sortilin is in an equilibrium between monomer and dimer, but ligands are bound preferentially to the monomer form. Ligand bound sortilin is internalized. Upon endocytosis, sortilin dimerizes due to the acidification along the endosomal pathway and thereby releases its ligand. Sortilin dimers may control recycling by the retromer and GGA

observed if the cytosolic tags (myc and flag) are within 40 nm distance of each other[34]. Co-expressed wt sortilin$_{mb-myc}$ and sortilin$_{mb-flag}$ were able to form dimers in cells. In addition, the A464E mutant versions of sortilin$_{mb-myc}$ and sortilin$_{mb-flag}$ showed a significant decrease of sortilin dimer formation in cells. Since A464E is an interface mutant based on the s-sortilin dimer structure, these data indicate that membrane bound sortilin homodimerizes in cells via the top face of the β-propeller, in a fashion similar to s-sortilin in the crystals. The dimerization is however not completely abrogated by the A464E mutation, an effect we also observe for s-sortilin A464E in solution. Alternatively, it is possible that two monomer sortilin molecules are within 40 nm proximity to give rise to a PLA event and that the counts from the sortilin A464E mutant are mainly from monomer protein. On average, $52 \pm 19$ PLA events per cell were detected in cells transfected with the wt construct ($N = 28$) against $36 \pm 19$ PLA events for the mutated construct ($N = 23$). An unpaired *t*-test indicated a *p*-value of 0.0036 for the difference between the wt and mutant sortilin. Sortilin is thus able to dimerize in cells through its luminal domain, and this dimerization brings its cytosolic domains into <40 nm proximity.

## Discussion
Progressive acidification of compartments along the endocytic pathway (from pH 6.0 to 6.5 in early endosomes to 4.5–5.5 in late endosomes and lysosomes) plays a role in release of ligands from endocytosis receptors, in protein sorting and targeted ligand degradation in the lysosome[35]. The mechanisms underlying ligand release are largely unresolved. For the LDL endocytosis receptor a low pH-induced conformational change consisting of

domain rearrangements of the luminal segment discharges ligands[1]. It is not clear if this mechanism is used by other endocytosis receptors to discharge ligands after endocytosis. Low pH-induced conformational changes have been shown for diverse receptors such as epidermal growth factor and asiaglycoprotein receptors and LDL receptor-related protein 1[36, 37]. On the other hand low pH-induced dimerization has only been observed in a few instances for virus and plant proteins[38, 39]. To identify if there are other mechanisms, besides the low pH-induced domain rearrangement described for the LDL receptor to discharge ligands, we have focused on sortilin, which has been implicated in the endocytosis of a broad range of ligands.

We detail that sortilin dimerizes and undergoes a conformational change at acidic pH. All structures of s-sortilin determined previously were either of s-sortilin-neurotensin or s-sortilin-inhibitor complexes and indicated s-sortilin to be a monomer[19–21, 27]. The pH of the crystallization conditions (between 7.2 and 7.9) or the bound ligands most likely limited formation of sortilin dimers. In particular, the ligands bind in a hydrophobic pocket in the central hole of the β-propeller and are sterically hindering the required conformational change in the sortilin β-propeller and thus stabilize the monomer conformation. Our mouse s-sortilin structure, crystallized at pH 7.5, also adopts a monomer structure of nearly identical conformation compared to the human s-sortilin structures. Unmodelled poorly resolved electron density is present in this mouse structure at the well-resolved neurotensin binding site. This additional electron density may correspond to a low occupancy small molecule co-purified from the expression medium, but this has not been modeled (Supplementary Fig. 11). The s-sortilin dimer structures, crystallized at acidic pH, reveal that dimerization is accompanied by a substantial conformational

rearrangement within the ß-propeller and two 10CC domains. The conformational change within the ß-propeller is unexpected and, to our knowledge, such structural plasticity has not been observed for the supposedly stable ß-propeller fold before. This adds a new structural dimension to the large ß-propeller containing protein family[40] and may provide additional control for these proteins over catalytic function, ligand binding or signaling roles.

Sortilin dimerization and conformational change may provide a double mechanism for ligand release at acidic pH. Both processes provide an altered surface for sortilin; dimerization shields a large interaction interface on the top face of the β-propeller and the conformational change modifies the surface properties locally but over a much larger area (except for blades 3–5). Hydrophobic loops on the top face of the sortilin β-propeller, previously hypothesized to be interacting with the cell membrane[20] are buried in the sortilin dimerization interface. The conformational change at acidic pH is required for release of neurotensin and, probably, other peptide-based ligands such as spadin, whereas the dimerization may sterically hinder other sortilin ligands (Fig. 8). A linear epitope in blade 2 of sortilin has previously been identified as a binding site for proNGF[41]. This epitope is located 28 Å away from the dimerization interface and the NGF part of proNGF is 60 Å long. It is currently not known how proNGF exactly interacts with sortilin but residues that become buried in the dimer may contribute to binding proNGF in the sortilin monomer. The ligand-release mechanism of sortilin is different from that of the LDL receptor in two ways; the conformational change in sortilin occurs within the domains as opposed to a domain movement described for LDL receptors and sortilin dimerizes whereas the LDL receptor does not change its oligomeric state between different pH conditions. The sortilin dimer presents a new surface that may be preferentially recognized by ligands in acidic cellular compartments to enable sortilin mediated sorting of these ligands between acidic cellular compartments. Such ligands would, presumably, be released by monomerization of sortilin at neutral pH. The concomitant change of conformation and oligomerization state, described here for sortilin, may be a general ligand release mechanism for sorting and endocytosis receptors.

Ligand release by sortilin dimerization could be coupled to shuttling and recycling of sortilin by cytosolic adaptor complexes such as GGA1 (for transport from the TGN to endosomes) and the retromer complex (for transport from endosomes to the TGN or the plasma membrane). In the sortilin dimer crystal structure the C-termini at the luminal side are in close proximity and at a distance of 37 Å. The sortilin dimerization also brings the transmembrane α-helices and cytosolic tails in proximity of each other. Nine residues, between the C-terminus in the structure and the α-helix are disordered and the orientation of the α-helices and structure of the cytosolic tail is unresolved. Both the GGA and retromer complex are homodimers of which each of the two chains individually recognize and bind a sortilin cytosolic tail[42,43]. Possibly, the dimerization of sortilin on the luminal side enhances the binding of the adaptor complexes to sortilin on the cytosolic side by the avidity effect arising from sortilin homodimer to adaptor homodimer interactions[43]. In this mechanism, dimerization of sortilin would provide the trigger for shuttling of sortilin to different compartments (Fig. 8), but this hypothesis has not been experimentally verified.

Sortilin dysfunction is associated with numerous pathologies due to its multifunctional role in protein sorting, and sortilin requires tight regulation for proper function. Dimerization of sortilin is not completely abrogated at neutral pH, and our SEC-SAXS data indicates that the shape of the sortilin dimer at neutral pH may be different from the dimer at acidic pH. These shape differences may arise from differences in flexibility and position of the 10CC-a and 10CC-b domains. However, our observations that the A464E dimerization interface mutation also shifts the equilibrium of dimerization toward more monomers at neutral pH in native MS experiments and on the cell surface in the PLA analysis, indicate that the sortilin dimerization interface is likely similar independent of pH. Possibly, the conformational change required for dimerization provides an extra level of control to keep the dimerization in check. In addition, cells may have mechanisms to fine-tune the sortilin dimerization process by modifying the glycosylation pattern, either at the biosynthesis level or by glycan trimming by glycosidase enzymes. We have shown that glycosylated s-sortilin forms dimers more readily compared to the deglycosylated form. Indeed, two forms of sortilin that differ in N-linked glycosylation have different signaling and transport roles in HT29 cells; a higher glycosylated form is responsible for neurotensin endocytosis whereas a less glycosylated form binds neurotensin in the TGN[6]. Another control mechanism is provided by the sortilin propeptide spadin that prevents ligand binding to newly synthesized sortilin in the ER or cis-Golgi network at neutral pH[22]. Spadin competes with neurotensin for sortilin binding, indicating an overlapping binding site, and is released from sortilin at acidic pH[22], most likely in a fashion similar to neurotensin release. Possibly spadin is able to limit the amount of sortilin dimerization, in a similar fashion as we have shown for neurotensin, and can thus regulate the transport of sortilin between cellular compartments. At more acidic pH during receptor secretion, dimerization of sortilin may take over the ligand-binding inhibiting role of spadin to prevent overzealous ligand binding to sortilin that has been newly produced or is being recycled (Fig. 8). Thus, sortilin dimerization is regulated at the cellular level by pH, processing and glycosylation.

## Methods

**Generation of protein constructs and mutagenesis.** The sequence of the Mouse sortilin luminal segment, residues 1–722 (numbering excluding the signal sequence), was obtained from DNA 2.0 as codon-optimized version for expression in human cell lines (Supplementary Table 5). The A464E point mutation, in the dimerization interface, was introduced by a two-step PCR with overlapping primers (Supplementary Table 6). The sequences of furin-resistant mouse proNGF and proBDNF (with all furin sites modified from RR/KR to AA), were obtained from DNA 2.0 as codon-optimized versions for expression in human cell lines. All constructs were subcloned using BamHI/NotI sites in pUPE107.03 (cystatin secretion signal peptide, C-terminal His$_6$-tag, U-Protein Express), unless indicated otherwise.

**Protein expression and purification.** Constructs were transiently expressed as secreted version either in Epstein-Barr virus nuclear antigen I (EBNA1)-expressing HEK293 cells (HEK293-E)[44] or in N-acetylglucoaminyltransferase I-deficient (GnTI−) EBNA1-expressing HEK293 cells (HEK293-ES) (U-Protein Express). HEK293-ES cells produce proteins with shorter, more homogeneous high mannose glycans ("short" glycan type), while HEK293-E cells produce native-like protein with hybrid glycans ("native" glycan type). Proteins produced in HEK293-ES cells were used for crystallization and deglycosylation. Proneurotrophins produced in HEK293-ES cells were used in Surface Plasmon Resonance experiments. S-sortilin produced in HEK293-E cells was used for all other experiments unless stated otherwise. Medium was collected 6 days after transfection and cells were spun down by 10 min of centrifugation at 1000×g. Supernatant was concentrated fivefold and diafiltrated against 500 mM NaCl, 25 mM 4-(2-hydroxyethyl)-1-piperazineethanesulfonic acid (HEPES) pH 7.5 (IMAC A) using a Quixstand benchtop system (GE Healthcare) with a 10 kDa molecular weight cutoff (MWCO) membrane. Cellular debris were spun down for 10 min at 9500×g and the concentrate was filtered with a glass fiber prefilter (Minisart, Sartorius). Protein was purified by Nickel-nitrilotriacetic acid (Ni-NTA) affinity chromatography and eluted with a mixture of 60% IMAC A and 40% of 500 mM NaCl, 500 mM imidazole and 25 mM 4-(2-hydroxyethyl)-1-piperazineethanesulfonic acid (HEPES) pH 7.5 (IMAC B). For crystallization experiments, this was followed by size exclusion chromatography (SEC) on a Superdex 200 Hiload 16/60 column (GE Healthcare)in 150 mM NaCl, 20 mM HEPES pH 7.0, for all other experiments the SEC was performed in 25 mM MES pH 5.5, 150 mM NaCl because in our hands s-sortilin does not form aggregate or precipitate at pH 5.5, while contaminants precipitate, forming a white powder on the side of the tube which is easily spun down and separated from pure

s-sortilin which remains in solution. Protein was concentrated to 14.2 mg mL$^{-1}$ for s-sortilin, 11.3 mg mL$^{-1}$ for proNGF and 9.9 mg mL$^{-1}$ for proBDNF using a 30 kDa MWCO (10 kDa MWCO for proNGF and proBDNF) concentrator before plunge freezing in liquid nitrogen and storage at −80 °C.

**Crystallization and data collection of mouse s-sortilin**. Samples were concentrated to 14.2 mg mL$^{-1}$ in buffer 25 mM HEPES pH 7.0, 150 mM NaCl. Sitting-drop vapor diffusion at 18 °C was used for all crystallization trials, by mixing 150 nL of protein solution with 150 nL of reservoir solution. S-sortilin was also set-up for crystallization after deglycosylation, in which case it was deglycosylated using EndoHf 1:100 O/N at RT in buffer pH. Crystal forms 2, 3, and 4 (also see Table 1) were grown from a 1:1 molar ratio mixture of s-sortilin with proneurotrophins, but proneurotrophins were not present in the crystals. Crystal form 1 was obtained from deglycosylated s-sortilin concentrated to 14.2 mg mL$^{-1}$ in a condition containing 0.18 M magnesium formate dihydrate pH 7.0, 18% polyethylene glycol (PEG) 3350 (w/v) and 10 mM tris(2-carboxyethyl)phosphine hydrochloride; final pH 6.2. Crystal form 2 was obtained from deglycosylated s-sortilin at a final concentration of 5.6 and 2.0 mg mL$^{-1}$ proBDNF in a condition containing 0.2 M NH$_4$Cl, 1 mM CaCl$_2$ and 20% PEG 3350 (w/v), final pH 5.0. Crystal form 3 was obtained from deglycosylated s-sortilin at a final concentration of 8.0 and 3.0 mg mL$^{-1}$ proNGF in a condition containing 0.18 M magnesium formate dihydrate pH 7.0, 18% PEG 3350 (w/v), 1 mM CaCl$_2$ and 1 mM L-Glutathione reduced and L-Glutathione oxidized, final pH 6.2. Crystal form 4 was obtained from s-sortilin at a final concentration of 8.0 and 3.0 mg mL$^{-1}$ proNGF in a condition containing 0.1 M HEPES pH 7.5, 1 mM CaCl$_2$ and 25% PEG 2000 monomethyl ether (w/v). Crystals were collected and flash-cooled in liquid nitrogen in the presence of reservoir solution supplemented with 25% ethylene glycol. Diffraction data were collected at 100 K at the Swiss Light Source (SLS Villigen, Switzerland) and the European Synchrotron Radiation Facility (ESRF Grenoble, France). Data were processed by MOSFLM or XDS and AIMLESS[45–47].

**Structure determination and refinement**. Resolution limits were determined by applying a cutoff based on the mean intensity correlation coefficient of half-data sets, CC1/2. The structure of mouse s-sortilin was solved by molecular replacement using either the structure of human s-sortilin (PDB code 3F6K; crystal form 4) or the structure of monomeric mouse s-sortilin (crystal form 4) as search model in Phaser[48]. Model building for sortilin was performed manually using COOT[49]. Structure refinement was performed using PHENIX[50] and REFMAC5[51] (see Table 1 for data set and refinement statistics). Molprobity[52] was used for structure validation. Structural analysis was performed using various programs of the CCP4 suite. Comparison of the monomer and dimer structures was done on the basis of an overlay of all monomer and dimer chains available. The electrostatic properties of both monomer and dimer forms at pH 7.4 and pH 5.5 were analyzed using the PDB2PQR server[53, 54] with a PARSE forcefield and the PROPKA software[28]. Figures were generated with PyMol (Schrödinger). Videos were generated using the Morph Conformation feature of Chimera[55].

**Size exclusion chromatography multi-angle light scattering**. Size exclusion chromatography multi-angle laser light scattering (SEC-MALS) was used to determine the oligomeric state of s-sortilin at pH 5.0 and 7.4. For each SEC-MALS run, 10 μl of 10 mg mL$^{-1}$ s-sortilin was injected into a Superdex 200 10/300 GL gel filtration column (GE Healthcare) and separated with a flow rate of 0.5 ml min$^{-1}$ in 25 mM HEPES pH 7.4, 150 mM NaCl or 25 mM 2-(N-morpholino)ethanesulfonic acid (MES) pH 5.0, 150 mM NaCl. For molecular weight characterization, light scattering was measured with a miniDAWN TREOS multi-angle light scattering detector (Wyatt), connected to a differential refractive index monitor (Shimadzu, RID-10A) for quantitation of the protein concentration. Chromatograms were collected, analyzed and processed by ASTRA6 software (Wyatt, using a calculated dn/dc value of 0.185 ml g$^{-1}$, determined from a dn/dc of 0.188 for the protein part, a dn/dc of 0.145 for the glycans and 8.3% glycosylation based on the native mass spectrometry data). The calibration of the instrument was verified by injection of 10 μl of 10 mg mL$^{-1}$ monomeric bovine serum albumine (BSA, Sigma-Aldrich).

**SEC-SAXS measurements and data analysis**. SEC-SAXS experiments were carried out on the BM29 beamline at ESRF Grenoble[56]. Interpretation of batch experiments of s-sortilin suffered from aggregation even at concentrations as low as 0.2 mg mL$^{-1}$. Given the sensitivity for batch SAXS for small amounts of large aggregates we used SEC-SAXS instead and the problem of protein aggregation was alleviated in these experiments. A volume of 40 μL wild-type s-sortilin at 12.2 mg mL$^{-1}$ and mutant s-sortilin A464E at 10 mg mL$^{-1}$ were loaded on a Superose 6 10/300 column (GE Healthcare) via a high performance liquid chromatography (HPLC) system, consisting of an in-line degasser (DGU-20A5R, Shimadzu, France), binary pump (LC-20ADXR, Shimadzu, France), valve for buffer selection and gradients, UV–VIS array photospectrometer (SPD-M20A, Shimadzu, France) and a conductimeter (CDD-10AVP, Shimadzu, France) attached directly to the sample-inlet valve of the BM29 sample changer[57]. Each sample was measured in two different conditions, either in 25 mM HEPES pH 7.4, 150 mM NaCl or 25 mM MES pH 5.5, 150 mM NaCl. The effect of neurotensin was measured by adding neurotensin from a 1 mM stock in 10 mM acetic acid pH 3.5 to a final 2:1

neurotensin:s-sortilin molar ratio. Samples were buffer exchanged and the column was equilibrated with 1.5 CV to the corresponding buffer and a stable background signal was confirmed before measurement. Measurements were performed at room temperature and a flow rate of 0.6 mL min$^{-1}$ was used for all sample measurements. All the SAXS data from the run were collected at a wavelength of 0.99 Å using a sample-to-detector (PILATUS 1 M, DECTRIS) distance of 2.81 m. The scattering of pure water was used to calibrate the intensity to absolute units[58]. The intensities were scaled such that the forward scattering corresponds directly to the concentration (in mg mL$^{-1}$) times the molar mass (in kDa) of idealized proteins, i.e., 1 a.u. = 8.03×10$^{-4}$ cm$^{-1}$, unless explicitly stated otherwise. 2400 frames (1 s each) were collected per 40 min run. Data reduction were performed automatically using the EDNA pipeline[59]. Frames in regions of stable $R_g$ were compared with CORMAP[60] to ensure signal stability in these ranges and 10–20 frames with good signal to noise were selected and averaged using PRIMUS[61] to yield a single averaged frame corresponding to the scattering of an individual SEC species. Protein concentrations were estimated based on the absorbance at 280 nm assuming a molecular extinction coefficient of 103 M$^{-1}$ cm$^{-1}$ for the monomer and of 206 M$^{-1}$ cm$^{-1}$ for the dimer.

Pair distance distribution functions were created with GNOM[62] and used to calculate 40 ab-initio models in C1 symmetry with DAMMIF[63]. The models were averaged, aligned and compared using DAMAVER[64]. As the differences between the predicted scattering curves (from WAXSiS[65, 66]) of the monomer s-sortilin crystal structure and the monomer sub-unit of the dimer s-sortilin crystal structure, as well as the differences between models with and without added glycans, were negligible in comparison to those observed experimentally at different pHs, the structural changes within the β-propeller domain where ignored for SAXS modeling. Missing residues were added to the monomer crystal structure with psfgen[67] and the resulting structure was relaxed using the energy minimization tool of sassie-web[68, 69]. This structure was used as a starting point for rigid body modeling using CORAL[61] and ensemble based modeling using EOM[70] at both pH 7.4 and pH 5.5. For the s-sortilin dimer structures, the relative positioning of the β-propeller domains was based on their arrangement in the crystal structure.

**Surface plasmon resonance**. Equilibrium binding studies were performed using an MX96 instrument (IBIS Technologies). Mouse NGF purified from submaxillary glands was purchased from Biorad. NGF, proNGF and proBDNF at 150, 200, and 250 μg mL$^{-1}$ were amine-coupled for 45 min at pH 4.5 to a planar-type P-COOH SensEye SPR sensor (IBIS Technologies) after 1-ethyl-3-(3-dimethylaminopropyl) carbodiimide hydrochloride/N-Hydroxysuccinimide (EDC/NHS) activation. Wt and A464E s-sortilin was flowed over the sensor chip, as analyte, in buffer containing either 25 mM HEPES pH 7.4 or 25 mM MES pH 5.0, 150 mM NaCl and 0.005% Tween 20. Temperature was kept constant at 25 °C. The data were analyzed using SprintX (IBIS Technologies) and SigmaPlot and modeled with a 1:1 Langmuir binding model to calculate the dissociation constant ($K_d$) and the maximum analyte binding ($B_{max}$).

**Analytical ultra centrifugation**. Sedimentation velocity experiments were carried out in a Beckman Coulter Proteomelab XL-A analytical centrifuge with An-60 Ti rotor (Beckman) at 42,000 revolutions per minute (r.p.m.). Three concentrations of s-sortilin, 1, 2, and 10 μM, were measured in 25 mM HEPES pH 7.4 and 150 mM NaCl at 20 °C. Absorbance was determined at 230 nm for the 1 and 2 μM samples and at 280 nm for the 10 μM sample. A total of 350 scans were collected per cell. Every sixth scan was used in continuous c(s) mode analysis in SEDFIT[71]. Sedimentation equilibrium experiments were carried out in a Beckman Coulter Proteomelab XL-I and a Beckman Optima XL-A analytical ultracentrifuge. Either 12 or 3 mm centerpieces with quartz windows were used, 12 mm for the lowest concentrations and 3 mm for the others. An-60 and An-50 Ti rotors (Beckman) were used to carry out the measurements. S-sortilin constructs were diluted with and dialyzed against buffer (either 25 mM HEPES pH 7.4 or 25 mM MES pH 5.5, 150 mM NaCl) using a 30 kDa MWCO membrane. The effect of neurotensin was measured by adding neurotensin from a 1 mM stock in 10 mM acetic acid pH 3.5 to a final 2:1 neurotensin:s-sortilin molar ratio before dialysis. Protein concentrations of 2, 10, and 50 μM were used. Sedimentation equilibrium runs were performed at 20 °C and at 7500, 14,000, and 20,000 r.p.m. Absorbance was determined at 250 and 280 nm using the respective buffer as reference. Extinction coefficients were determined by Protparam[72] based on the mature s-sortilin sequence and kept constant for each wavelength. Buffer density and viscosity were determined by SEDNTERP as 0.99823 g mL$^{-1}$ and 0.001002 Pa. s$^{-1}$, respectively. The partial specific volume for s-sortilin of 0.729 mL g$^{-1}$ is based on the amino acid sequence excluding the glycans and was determined with SEDNTERP. Analysis and fitting of the data were performed using the program SEDPHAT v.14.3[73].

**Fluorescence microscopy and in situ proximity ligation assay**. Constructs containing the sortilin residues identical to the crystal structure construct followed by a single transmembrane helix from human GPA33 were subcloned in pUPE07.30 and pUPE07.14 (cystatin secretion signal peptide, C-terminal myc-tag for 07.30 and flag-tag for 07.14, U-Protein Express) and transfected in adherent HEK293T (Large T antigen) cells, using Polyethylenimine (PEI, 1: 6 DNA: PEI ratio) in a 10 cm Petri dish, containing 3 × 10$^6$ cells in 8 ml Dulbecco's Modified

Eagle's Medium. A DNA titration of 1:100 (w/w) with dummy DNA was used[74] and both constructs were mixed 1:1. After 5 h, the transfection medium was displaced by culture medium. After two days about 150,000 cells were plated onto Menzel cover glasses (19 mm diameter, Fisher Emergo) in a 12-wells plate. Twenty-four hours later the cells were washed two times 5 min in phosphate buffer saline (PBS) and fixed for 30 min in 4% paraformaldehyde. After 3 times 5 min PBS washes, cells were permeabilized 3 min in 0.1% Triton in PBS, then blocked 30 min at 37 °C in 8% BSA in PBS. Cells were washed three times for 5 min in PBS before overnight incubation with primary antibodies (Rabbit anti-c-myc and Mouse anti-flag M2 monoclonal, Sigma, 1:300 dilution in 1% BSA, 1% Tween 20 in PBS (PBST)). Cells were washed three times 5 min in PBST before following the Duolink in situ protocol. Negative controls, by omitting either one of the sortilin constructs or by omitting the primary antibody, did not show any PLA events. Both Minus and Plus PLA probes interact with a rolling-circle nucleotide template when the distance between them is less than 40 nm. These complexes were ligated in the presence of a ligase in hybridization solution. The circular template was then amplified using a polymerase, while red-labeled probes hybridized the amplified sequence. Cover slips were mounted using Vectashield mounting medium with DAPI. Images were acquired using a Zeiss LSM 700 microscope. The analysis was done using FIJI. To minimize effects arising from differences in sortilin expression levels, only individual cells with a combined average pixel intensity for the myc and flag antibodies in the range of 3400–10,200 units were taken in account. The distribution of fluorescence intensity for cells transfected with wt or mutant sortilin was similar. A background intensity cutoff of 30 intensity units for the PLA events was applied. The robustness of the analysis was tested by three different cutoffs for the size of one PLA event (0.5 μm$^2$, 1 μm$^2$, and 2 μm$^2$). All cutoffs showed a significant difference in the number of PLA event between the wt and A464E mutant sortilin, with 1.5 times more PLA events in cells transfected with the wt construct compared to the cells containing the mutated construct. The non-parametric Mann–Whitney test was used.

**Native mass spectrometry**. Wt s-sortilin and A464E were produced recombinantly in HEK293-E and HEK293-ES cells, purified as described above and subsequently buffer exchanged to 150 mM ammonium acetate (pH 5.0 or pH 7.5) using Amicon Ultra-0.5 mL centrifugal filter units with a 30 kDa MWCO (Milipore). Next, the samples were diluted to about 4 μM final protein concentration, loaded into gold-coated borosilicate capillaries and analyzed by native nano-electrospray ionization MS using a modified quadrupole-time of flight mass spectrometer (MS Vision, Waters) operated in positive ion mode[75]. The instrument parameters were set as follows: 1.3–1.4 kV capillary voltage, 90 V sample cone voltage, 60 V extraction cone voltage, 30 V collision energy, 10 mbar source pressure, 1–1.5 × 10$^{-2}$ Xe gas pressure in the collision cell. Singly charged, mono-isotopic CsI cluster ions were used as an external mass calibrant. The reported standard deviations of the molecular weights were calculated from the different charge states of the respective species. The mass spectra were analyzed using MassLynx v4.1 (Waters). As the monomer and dimer m/z envelopes are well separated the relative abundances of s-sortilin monomer and dimer in the native mass spectrometry data were determined from the extracted ion currents for the m/z ranges of the monomer and of the dimer, which corresponds to the area under the respective charge state envelopes.

**MS-based glycan mapping**. Wt s-sortilin produced in HEK293-E and HEK293-ES cells were denatured in the presence of 8 M urea, reduced with dithiothreitol and alkylated with iodoacetamide. Subsequently, the samples were 10-fold diluted to reduce the urea concentration and to allow sequential proteolytic digestion with Glu-C (Roche, protease:substrate ratio (w/w) 1:75, 4 h at room temperature) and trypsin (Promega, protease:substrate ratio (w/w) 1:100, overnight at 37 °C). The peptide mixtures were desalted, dried under vacuum, reconstituted in 10% (v/v) formic acid and analyzed by nano-high performance liquid chromatography/tandem mass spectrometry (LC–MS/MS). The analyses were performed using either an ultra-HPLC Agilent 1200 system (Agilent Technologies) coupled on-line to an Orbitrap Fusion mass spectrometer (Thermo Fisher Scientific) or a Proxeon EASY-nLC 1000 system coupled on-line to an Orbitrap Elite mass spectrometer (both Thermo Fisher Scientific). In both case, peptides were separated by reversed-phase chromatography using in-house packed columns (Poroshell 120 EC-C18, 2.7 μm (Agilent Technologies)) and a 60 min gradient elution. All precursor ion (MS1) and fragment ion (MS2) mass spectra were acquired in the Orbitrap mass analyzer. On the Orbitrap Fusion, MS1 analysis was performed in top speed mode with 3 s cycle time and 140,000 mass resolution at m/z 200. Precursor ions ($z \geq 2$) were fragmented using sequential higher-energy collisional dissociation (HCD) and electron-transfer/higher energy collisional dissociation (EThcD) and MS2 scans were acquired with 30,000 mass resolution at m/z 200. On the Orbitrap Elite, MS1 analysis was performed using a mass resolution of 60,000 at m/z 200. The three most abundant precursor ions ($z \geq 2$) were subjected to sequential HCD-EThcD fragmentation and MS2 scans were recorded with 15,000 mass resolution at m/z 200. The MS data were analyzed using Byonic v2.6 (Protein Metrics), allowing 10 p. p.m. precursor mass tolerance and 20 p.p.m. fragment mass tolerance and forcing the software to skip low-quality mass spectra. For peptide identification, a concatenated target-decoy database was generated based on the amino acid sequences of s-sortilin and 47 common HEK cell contaminant proteins (identified in a separate Mascot search) with the following settings: proteolytic cleavage C-terminal

of Asp, Glu, Arg or Lys; up to 6 missed cleavage sites allowed; carbamidomethylation of Cys (as a fixed modification); oxidation of Met and N-glycosylation of Asn (as common variable modifications); O-glycosylation of Ser and Thr, phosphorylation of Ser and Thr, acetylation of N-termini and Lys (as rare variable modifications). Peptides were allowed to carry up to 4 common and 1 rare variable modification. Glycan trees were identified based on 2 Byonic glycan libraries containing the 6 most common O-glycans and 38 common biantennary N-glycans. Identified peptides were filtered using an automatic score cutoff, and are reported at 1% false-discovery rate. In addition, all MS2 spectra representing glycosylated s-sortilin peptides were manually verified.

**Data availability**. Coordinates and structure factors for s-sortilin structures 1–4 have been deposited in the Protein Data Bank with succession numbers 5NMT, 5NNI, 5NNJ, and 5NMR, respectively. All SAXS data are made available at the small angle scattering databank (SASBDB) with the accession codes SASDCW5 (dimeric s-sortilin at pH 7.4), SASDCX5 (monomeric s-sortilin at pH 5.5), SASDCY5 (dimeric s-sortilin at pH 5.5), SASDCZ5 (monomeric s-sortilin at pH 7.4), SASDCE7 (monomeric s-sortilin at pH 7.4 in the presence of neurotensin) and SASDCF7 (dimeric s-sortilin at pH 7.4 in the presence of neurotensin). Other data are available from the corresponding author upon reasonable request.

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

## Acknowledgements

We thank the staff of the SLS beamline PX and ESRF beamlines ID23–1 and BM29 for assistance with X-ray diffraction and scattering data collection. We thank Wim de Lau for providing us with the HEK293T cell line and the Biology Imaging Center (Utrecht University) for access to fluorescence microscopy equipment. We thank Camilla De Nardis for help with SAXS data analysis, Fan Liu for help with designing the MS acquisition methods and Joke Granneman for HEK293T cell culture. This work was funded by the Initial Training Network grant "ManiFold" from the EU under FP7 (grant agreement number 317371) and an Investment Grant (721.012.004) from the Netherland Organization for Scientific Research (NWO). Additional support was provided through the NWO-funded Roadmap Initiative Proteins@Work (184.032.201). B.J.C.J. is supported by an NWO Vidi grant (723.012.002). D.H.M. is funded by an EMBO long-term fellowship.

## Author contributions

N.L. and B.J.C.J. designed the experiments N.L. designed constructs, purified proteins and did all the structural biology (X-ray diffraction and SAXS), MALS and SPR experiments. N.L. and D.M.E.T.-W. performed the SE-AUC experiments and D.M.E.T.-W. analyzed the data. P.L. performed mass spectrometry experiments and analyzed the data

together with A.J.R.H.; N.L., and D.H.M. did the PLA assay and fluorescence microscopy. M.B. helped with SAXS data collection and did the analysis. B.J.C.J. supervised the project. N.L and B.J.C.J. wrote the manuscript with input from all authors.

## Additional information

**Competing interests:** The authors declare no competing financial interests.

