## [Peer Review file · Nature Communications]

Reviewers' comments:

Reviewer #1 (Remarks to the Author):

Janssen and co-workers study dimerisation of the transmembrane receptor Sortilin. Crystal structures reveal pH-dependent conformational changes and dimerisation of the protein at low pH. They further use various biophysical techniques to describe Sortilin dimerisation and ligand binding which is favoured by the monomeric protein. The manuscript is rather lengthy and the experiments are not clearly described. I suggest the following changes before considering publication of the manuscript.

Abstract

Line 15-17: It is difficult to understand this sentence.

Introduction

The introduction is rather lengthy. I suggest to shorten the introduction and only describe the facts that are important for the understanding of the manuscript.

I suggest to change some terms, e.g. unliganded (line 31) and endocytosed (line 44). Even though these terms exist they sound a bit stiff in the context.

Line 97: "interactions for several" change to: "interactions with several"

Results

First paragraph (page 4-5): The crystal structure is described, however, the resolution which is an important result when discussing crystal structures is missing. Only r.m.s.d. values are given.

Line 138-140: The discussion of the results is confusing. Please clarify.

Page 5, second paragraph: The explanation why dimeric Sortilin is oriented towards the cell-surface is not clear.

Page 6, first paragraph: The changes between monomer and dimer are rather small. Can these really be considered as conformational changes between the two forms? Please explain/discuss.

Page 7, line 209, 214 and 216: "predicted to be neutral" – please give a reference for this.

Page 8, first paragraph: Include how/which experiments were performed. It is difficult to follow the text without mentioning the techniques employed.

Line 238: Suddenly glycosylation is discussed. Please introduce this at an earlier point to help understanding why this is mentioned here.

Line 243: Please delete "biomolecular".

Line 244/245: "Similar injection concentrations" – please clarify. Was the protein concentration similar or the amount in the electrospray emitter?

Page 8, last paragraph: Deglycosylation of Sortilin might already affect its dimerisation behaviour.

The naturally modified protein might be a better measure to show pH-dependence.

Line 257: KD of 4 μ M. Please clarify which KD? Dimeric species?

Page 10, first paragraph: Native MS experiments of the mutant could also be used for quantification.

Line 308 and 310: "in the presence ". Please correct.

Line 319: proBDNF – please introduce abbreviations. Also check the entire manuscript, there are more abbreviations used which are not introduced.

Page 11, second paragraph, Line 346 etc.: 40 nm distance is quite large compared to the distances that were considered before as significant changes. Tags are usually rather flexible and misleading interactions might be observed.

Discussion

Page 12, Lines 381-382: Low resolution might also be a result of dynamic protein regions.

Line 401: change "it" to "its"

Line 343: change "from" to "form"

Methods

Line 652: "score cut" change to "score cut-off"

Figures

Figure 5G: The mass spectra look rather unresolved which is most likely due to the presence of glycoforms. However, the calculated error of the protein (complex') masses are very low. I would expect a higher error for spectra of this quality. In addition, it is not really clear how the masses for the lightest and heaviest glycoforms were obtained. There are no additional peaks visible in these spectra! The quantification of the peaks was estimated, however, there are tools available for simulation/quantification of native MS data.

Figure 6: Please correct numbering of the panels (e.g. panel I missing from the figure legend). The resolution of these spectra is very low. Are there any reasons why this is the case compared with Fig. 5G? Are glycoforms present?

Supplementary Figure 3: Please correct reference to Fig. 6I and 5G. I disagree that this mass spectrum shows broader peaks than the one shown in Figure 5G. The peak width seems to be the same, however the spectra in Fig 6I and Supp Fig 3 are less resolved.

Supplementary Figure 5: Lots of the assigned peaks seem to be in the background.

General comments

It is not clear what the function of dimeric Sortilin is. If this is considered the active form it is surprising that ligands only bind the monomeric form. Please discuss.

The manuscript is rather too long. Too many figures.

Reviewer #2 (Remarks to the Author):

Low pH-induced conformational change and dimerization of Sortilin triggers endocytosed ligand release.

Leloup et al., 2017.

The subject: The X-ray crystal structure of sortilin at two different pH's (acidic and neutral) and the biophysical/structural characterisation of sortilin self-association, specifically the exchange between monomers and dimers, induced by pH changes or the binding of effector ligands.

Key finding: The luminal segment of Sortilin undergoes conformational changes and dimerization on lowering the pH and, to quote the authors "sortilin dimerization and conformational change discharges ligands and triggers recycling."

In general, the investigation by Leloup et al is well written and the results presented are quite interesting. However, there are many questions that remain unanswered. These questions may, or may not, be resolved by providing more specific descriptions and analyses of the results as presented, but will likely also require additional experiments. Importantly, and especially for the SAXS sections of the manuscript, portions of the analyses are missing. For example, the analysis of the SAXS data in terms of mixtures is extremely basic and needs to be well extended so as to offer the reader models of the sortilin monomer/dimer system that fit the scattering data as opposed to what is currently presented in the manuscript, i.e., models that absolutely do not fit the SAXS data. In essence, the SAXS sections can be simply boiled down to a single statement:

'We observe a decrease in the Porod volume of the sortilin mixtures on increasing the pH from 5.5 to 7.4, mutating A464 to E, or the addition of neurotensin, suggesting some -sort of disassociation of the sortilin dimer.'

Yet, is it not the case that the SAXS data analysis and interpretation is kind-of the linchpin that brings together all of the available data into a coherent structural narrative that supports the key findings? What seems a little strange is that the authors have a great deal of information to perform a really good SAXS-based analysis – crystal structures, binding constants, MW estimates of both the glycosylated and deglycosylated forms of the protein, mass spec data, etc, etc, etc. Why not integrate this information into the SAXS data interpretation and generate solution-state models of the sortilin mixture system that fit the SAXS data (be they ab initio model mixtures or rigid-body models, etc)? There are plenty of SAXS modelling programs that can do this (e.g., as part of the ATSAS suite there is OLIGOMER, GASBORMX, SASREFMX, GLYCOSYLATION (if you want to build glycans onto your X-ray crystal structures), even single value decomposition approaches.)

As an aside, it is quite bizarre that of the biophysical methods used by the authors to structurally interrogate the sortilin system – X-ray crystallography, AUC, SPR, MS, MALLS/RI – it is only the SAXS section that *completely* lacks any kind of quality assurance. For example, the data, p(r) profiles and Kratky plots lacks errors, there are no error estimates on the extracted structural

parameters (R_g , D_{max} , V_p , etc), there are no statistical estimates of quality of the data-model fits or $p(r)$ vs r reciprocal space fits to the data (even the AUC gets a χ^2) and who know what statistical checks and tests were employed during data reduction, subtraction and processing to ensure that the SAXS profiles being analysed are in fact what the authors think they are analysing. So what happened here – SAXS is just some hand-wavy 'well it is kind of a blob that looks like an elephant' kind of thing? Please refer to, and take note of, the recommendations outlined by the wwPDB Small-angle scattering taskforce for the presentation and reporting of small-angle scattering data:

Jacques et al (2012) Acta Cryst D68:620-626
<https://doi.org/10.1107/S0907444912012073>

and:

Trewhella et al. (2013) Structure 21(6):875-881
<http://dx.doi.org/10.1016/j.str.2013.04.020>

Finally, why did the authors employ SEC-SAXS for this project? Based on the SEC-traces provided by the authors in the manuscript there is no real need for SEC. If anything using SEC-SAXS has enormously complicated the scattering analysis. If the authors just use boring old traditional SAXS on individual samples, through a concentration series (e.g., six protein concentrations), the mixture analysis would be greatly simplified, the data will be of a higher quality and there would be more information from which to draw solid conclusions. Using regular buffer-sample-buffer-sample SAXS:

- 1) They can accurately assess the protein concentration for concentration-dependent estimates of MW from the forward scattering intensity, $I(0)$. If placed on an absolute scale, the glycan contributions to the scattering intensities can be assessed eliminating MW estimates from V_p (or other scattering invariant MW approaches).
- 2) They can make sure they have the correct solvent for background subtraction, something they have not demonstrated in the manuscript for the SEC-SAXS.
- 3) They can obtain R_g as a function of protein concentration.
- 4) They can obtain MW (i.e., $I(0)$) as a function of protein concentration.
- 5) They can obtain V_p as a function of protein concentration.
- 6) They can perform SVD on all of this data to calculate pure monomer scattering and pure dimer scattering contributions from which they can also determine the volume fraction of monomers and dimers at each protein concentration and pH.
- 7) Then it would be possible to directly compare the X-ray crystal structure of the monomer and dimer and monomer/dimer mixtures to the SAXS data to make sure that the X-ray crystal structures are not simply capturing some weird conformation (e.g., influenced by lattice/packing forces) that is not actually realised in solution.
- 8) They can model using ab initio or rigid-body approaches against the concentration-series SAXS data in parallel, extract volume fraction information and obtain more data to support the conclusion that there are conformational changes occurring in sortilin on dimer disassociation.
- 9) They can assess the effects of changing the pH in incremental steps (at the moment they are limited to 2 pH values – why not do smaller increments?)
- 10) They can assess the effect of altering the neurotensin concentration on the association states or sortilin under various pH conditions.
- 11) They can obtain low-resolution models of the sortilin/ neurotrophin system and how this is affected by pH, etc etc, to back up the SPR investigation.
- 12) They can assess the effects of deglycosylating the protein on the pH dependent monomer-dimer equilibrium/ neurotensin interaction, etc.
- 13) All of the above could be measured at a synchrotron SAXS beam in about, what, 2 hour, i.e., the time it takes for, 2 SEC runs (of course the analysis would take longer, but so much stronger)?

Additional short comments for consideration:

Introduction.

Appears to be well referenced.

X-tal crystallography.

For the models determined using X-ray crystallography, especially when describing the shifts in the amino acid side chains, show the electron density.

It is stated that:

"These observations indicate that the large-scale pH-induced structural rearrangement in sSortilin may be the result of local, residue level, changes in charge."

Fine, but what happens to the bound solvent on monomer-dimer association/disassociation? Is there a possibility of a change in entropy as well on dimer formation?

For the biophysical analysis in solution, including SAXS.

In the text it is mentioned that MALLS was performed. The MW correlation and the SEC traces are shown in supplementary Figure 2. A UV trace is shown in both Figure 5 and Supplementary Figure 2.

1) In the supplementary figure show the RI and MALLS traces. As the authors calculated the MW from RI and MALLS, show the data.

2) For the SAXS, the authors must show the $I(0)$ trace (i.e., the $I(0)$ calculated from each individual subtracted frame taken during the time of the SEC-SAXS experiment). The issue is that in Figure 5A the dimer looks to be stable through the SEC column, whereas at pH 7.4, there is clearly poorly resolved dimers and monomers. However, in Sup Fig 2, the reverse is true - here there are well resolved monomers at pH 7.4 and poorly resolved dimers at pH 5. So what type of sample was used for the SEC-SAXS? If the authors do not show the $I(0)$ trace and the region of the trace that was used to generate their final SAXS data profiles, how does a reader know that they simply didn't selectively choose frames that were 'kind-of monomeric' at pH 7.4 and 'kind of dimeric' at pH 5.5 (BTW: a consistent R_g through a SEC-peak means nothing by itself. Refer to Box 1 of Nature Protocols 11, 2122-2153(2016)). So the authors need to comment on what is clearly a dimer disassociation on the column into monomers, i.e., an inability to resolve monomers from dimers and how this might affect the interpretation of the SAXS data and that the SEC elution range chosen for the analysis encompasses either the dimer or the monomer.

In Figure 5 B, how were the two SAXS profiles scaled relative to each other? In this representation the SEC-SAXS data for the pH 5.5 dimer looks to be under subtracted?

It is stated that:(line 234):

"indicative of more dimer at pH 5.5 compared to pH 7.4."

How much is 'more'? The authors should attempt performing a mixture/oligomer analysis or maybe single value decomposition/principle component analysis to estimate the volume fractions of the monomer and dimers in both pH conditions. 'More' is not really all that quantitative?

Regarding the MW determined from SAXS - it is indeed true, as stated by the authors' on line 238:

"The M_m based on the V_p is dependent on the shape, flexibility and glycosylation state of the protein and is not suitable here as an absolute measure, but can still be used as a relative indicator for sSortilin dimerization."

But how do you know this is the case? Put a reference to this statement? Were attempts made to obtain the the MW from $I(0)$ (the forward scattering intensity) and the concentration estimated by UV to experimentally back up this statement?

Why not incorporate the volume fractions observed from native MS into the mixture analysis of the SAXS? Why not incorporate the K_d determined from AUC into a mixture analysis for SAXS?

Does the calculation of a $p(r)$ vs r distribution for a mixture make sense?

The authors state on line 240:

"In addition, the fit of the calculated scattering curves from the sSortilin crystal structures to the SAXS data is best for the monomer to the SAXS data at pH 7.4 and for the dimer to the SAXS data at pH 5.5 (Fig. 5c and d)."

Yeah, fine, but what do you mean by 'best'. SAXS does have a level of scientific integrity attached to it. Sweeping terms like 'more' and 'best' can be summarised in terms like the χ^2 discrepancy, or Correlation Map p-values if the experimental errors are not available or misspecified (Nature Methods (2015) 12, 419–422). It is not sufficient to state in Crystallography 'the best model kind-of fits the diffraction data to a better resolution than we thought it would' so why are the terms 'more' and 'best' okay for SAXS?

What is the R_g of the monomer and the R_g of the dimer calculated from the X-ray crystal structure monomers and dimers? How does this compare with the experimental R_g from the SAXS and what does this mean? For example, what is the effect of the glycosylation on the fits to the SAXS data? Are the authors fitting the correct models to the data? It is entirely possible to build glycans attached to X-ray crystal structures and assess the effect of their addition on the modelled SAXS profiles and the fits to the experimental data.

Why did the authors not attempt ab initio bead modelling, for example using GASBOR that can take into account monomer-dimer association/disassociation?

Why did the authors not try SASREF, i.e., using their crystal structures to perform rigid-body model refinement against the SAXS data under monomer-dimer equilibrium conditions? At least from this that might obtain the volume fractions of the monomer and dimer from their SEC-SAXS data, or even systematically evaluate the monomer/dimer contributions through the SEC peak?

For all SAXS data.

- 1) Show the data points (i.e., $I(s)$) with the errors, both on the subtracted profiles and the Kratky plots.
- 2) Show the errors on the $p(r)$ profiles - you must reference which program you used to perform the inverse indirect Fourier transform.
- 3) Show the Guinier plots for all constructs in the appropriate sR_g range.
- 4) Reference which program was used to perform the model-fits to the SAXS data.
- 5) Deposit the data and models into a SAXS-database, for example SASBDB (sasbdb.org).

The Kratky plots are on a very odd scale. The s_{min} value of their experimental $I(s)s^2$ seems to limit to 0, when in fact it should be the case that the $I(0)$ (at $s = 0$) should occupy the 0,0 point

For the mA464 mutation section.

Again, why use SEC-SAXS on this? Batch mode SAXS at 8 sample concentrations would be a better approach - it would be possible to show the monomer-dimer equilibrium much more clearly in terms of MW from $I(0)$ and systematic changes in R_g vs concentration, etc.

It is stated on line 283:

"However, at very low scattering vectors ($0-0.1\text{nm}^{-1}$) sSortilin A464E at pH 5.5 fits better to the calculated dimer scattering than to that of the monomer."

The issue is that there is no data (okay maybe a couple of points) in the s -range of $0-0.1\text{nm}^{-1}$? Again define 'fits better' using some sort of statistical test.

The authors state on line 286:

"It is unlikely that this is due to the pH-induced conformational rearrangement within a monomer as the calculated scattering curves from a single sSortilin chain of the dimer and monomer crystal structures are identical, i.e. the conformation differences within a chain cannot be distinguished by SAXS analysis."

This is not the case at all. All because the calculated scattering intensities from the crystallographic models are similar does not mean SAXS is incapable of detecting conformational changes within monomers in the solution state (and once again how do you know the calculated scattering intensities from the monomer X-tal structures are in fact identical, if you do not provide a metric to evaluate this?) In fact one of the main strengths of SAXS is that X-ray crystal structures often do not fit the SAXS data because the crystal structure, by definition, only captures a single conformational state. As none of the X-ray crystal structures presented in this manuscript fit the SAXS data then the simple Occam's razor principle applies, i.e., the X-ray crystal structure does not represent the solution state. And this is actually quite true because the effect of glycosylation has not been taken into account in the models when fitting the experimental SAXS data, nor the volume fraction weighted contributions of the glycosylated monomer and dimer. In other words, based on the presented data, for all we know the crystal structure does not reflect what is going on in solution and the monomers are actually sampling different conformational states as recorded in and captured by the SAXS data?

For the ligand binding investigation:

On Line 310 it is stated:

"Furthermore, SEC coupled SAXS analysis of wt sSortilin in presence of neurotensin at pH 7.4 showed a tendency for a smaller R_g (based on the Guinier plot and on the $P(r)$), and smaller D_{max} . Thus, neurotensin stabilizes the monomer form of sSortilin and prevents dimer formation. It can be inferred from this data that pH induced Sortilin conformational change and dimerization will trigger release of neurotensin from Sortilin."

Where is the SAXS data and modelling to support this claim? Quoting R_g is insufficient as R_g is influenced by both size and shape. For example, if the neurotensin binds to the centre of the B-propeller, then you would expect R_g to go down as the R_g of a ring is larger compared to a disc with an equivalent diameter. So if neurotensin 'fills in the hole' then of course R_g goes down. It is necessary to do the data analysis.

What happens at pH 5.5? Does the neurotensin bind? Surely this is interesting? In the discussion it is mentioned around line 396 that: "The conformational change at acidic pH is required for release of neurotensin and, probably, other peptide-based ligands such as spadin, whereas the dimerization may sterically hinder other Sortilin ligands (Fig. 9)." So show the SAXS data to support this very potentially-exciting claim?

Why was the SAXS investigation not extended to the binding of sorting to NGF? Surely this would greatly support the SPR investigation? The following statement(s) are unclear:

"Unfortunately, the affinity of wt sSortilin to (pro)neurotrophins in acidic conditions could not be determined due to nonspecific binding of wt sSortilin to the SPR sensor surface....Taken together, our data indicate that dimerization of Sortilin at acidic pH prevents ligand binding."

For the SAXS methods.

There is a requirement to be far more specific in the methods section for SAXS.

It is not possible to perform automatic merging of the SAXS data using GNOM (GNOME as listed by the Authors)

All because a set of subtracted SEC-SAXS profiles have a consistent R_g does not mean you are isolating a single species.

What was the buffer used for the SEC-SAXS data subtraction and how was it selected? How do you know that the correct buffer was used to correct for background scattering contributions? The scattering length density of a buffer can change during the course of a SEX run and it is entirely possible to detect the effects small-molecule like fractionation of the buffer components in the buffer scattering profiles.

What statistical analyses were performed to ensure that the individual subtracted SEC-SAXS data that were then scaled and averaged were intact statistically equivalent. Considering the poor separation of dimers from monomers at low pH (refer to Sup Figure 2) this could prove to be very important.

References are required for Guinier analysis and all of the program used to determine $p(r)$ vs r , the Porod volume, radial averaging routines, scaling etc, etc, etc.

Reviewer #3 (Remarks to the Author):

In this manuscript, Leloup et al presents a structural approach for determining the mechanism of pH-induced ligand release of the Vps10p family receptor, sortilin. The authors demonstrate a series of pH-induced conformational changes and homodimerization of sortilin, utilizing a variety of biochemical techniques to support their findings. These results are potentially significant, as sortilin shows a diverse array of ligand binding and is associated with multiple clinical pathologies. However, while the structural and in vitro data are quite convincing, whether the proposed mechanism exist in vivo is still unclear from presented data.

1. In vivo phenotypes of the A464E monomeric mutant should be probed. According to the proposed model, this mutant should not release its ligand and thus defective in protein trafficking to lysosomes. Is it possible to test this hypothesis by assaying lysosomal trafficking of one of the sortilin ligands, such as progranulin (Neuron 68, 654–667 (2010))?
2. It is unclear whether conformational change alone is sufficient to disrupt ligand binding or whether dimerization is also required. Lower affinity binding of ProNGF and Pro-BDNF to the A464E mutant at pH 5.0 would suggest conformational changes at acidic conditions might also decrease the interaction between sortilin and its ligands.
3. The authors raise the question of whether sortilin dimerization plays a role in regulating its recycling and trafficking between cellular compartments. This is an interesting point and the logical progression is sensible. However, they do not attempt to address this in the manuscript. Interaction between WT vs A464E mutant of sortilin and GGAs could be tested and compared.
4. Pro-NGF is reported to bind to linear epitope on the surface of the beta propeller of sSortilin (J Biol Chem. 2010 Apr 16;285(16):12210-22), which is very different from neurotensin binding. This should be mentioned in the manuscript and discussed.
5. Binding between sortilin and pro-NGF has been shown to be mediated by the pro-domain of NGF

(Nature. 2004 Feb 26;427(6977):843-8; Neuropeptides, Volume 42, Issue 2, April 2008, Pages 205–214). Thus it is really puzzling that NGF has a binding KD lower than pro-NGF in Table 4. The reference for the relative affinity of the pro-NgF/NgF to sortilin is also mis-interpreted in page 11. In J. Mol. Biol. 396, 967–984 (2010) and Nature. 2004 Feb 26;427(6977):843-8, these affinity is of pro-NgF/NgF to sortilin/p75 complex rather to sortilin alone.

6. Also, in page 11, "Binding of NGF and proNGF to sSortilin A464E at acidic conditions was reduced by a factor of 2 compared to neutral pH while binding of proBDNF was reduced by a factor 4. Taken together, our data indicate that dimerization of Sortilin at acidic pH prevents ligand binding." How can lower binding of pro-NGF and NGF to the monomeric mutant A464E at acidic mutant support the proposed model that dimerization of sortilin at acidic pH prevents ligand binding?

Minor notes

1. The authors incorrectly list the soluble secreted protein, progranulin, as a signaling receptor: "Sortilin interacts with a diverse set of ligands such as signaling receptors (e.g. TrkB, 81 EGFR, progranulin)..."
2. Fig. 6G, very hard to distinguish three curves. Might want to change color.
3. Fig 6. Legend: format changes between lettering of each image. E.g. "A.", "B.", "C.", "(D)", "(E)", etc.
4. Having a negative control in the PLA assay will be helpful determining the specificity of the assay.

Reviewer #4 (Remarks to the Author):

Thank you for the opportunity of reviewing this paper by Janssen and coworkers. In my opinion this paper has the potential to represent a significant advance in our understanding of the molecular mechanisms underpinning endocytosis, and focuses on changes in the conformation state, and state of oligomerisation of the transmembrane sortilin system and its interaction with relevant ligands, controlled by pH. It is generally very well presented and illustrated. My concerns are with some important details with regards the techniques employed. Additional experiments though may be needed before this can be considered for publication in Nature Comms.

Major comments.

p17 & Supplementary Fig 2. SEC MALS. Why is an old mini-Dawn (with 3 angles) and not a regular MALS device (Dawn, Helios) with the full angular complement being used? The molecules are high molecular weight (>100,000 g/mol), asymmetric structures and would be expected to show a significant angular dependence, requiring an extrapolation to zero angle. Could the authors present this angular data in "Supplementary Figures" to give the reader confidence in the values claimed?

line524. How was this dn/dc value "estimated": the value quoted seems high. How is glycosylation taken into consideration? This is critical because as i am sure the authors are aware, errors in molar mass/ conformational parameters are affected by the square of this parameter.

p19. SPR. Values for Kd quoted correspond to one of the interacting species immobilized onto a surface. Did the authors check for anomalies? : What happens when s-sortilin is immobilized onto the surface and the ligands become the analyte? This would give us greater confidence in the values quoted.

p18 & Supplementary Fig 4. AUC. Important detail is missing. Why was sedimentation velocity not used to check for sample heterogeneity?

lines 566, 567 I am assuming the authors mean 2, 10 and 50 μ M and not mM. Even so, these are quite high concentrations (up to 5 mg/ml) and I am surprised the uv absorption signals stayed within range for many cases (although in some cases not so).

Why wasn't the more accurate interference optical system on the XL-I used (rather than noisy uv optics) which has no such Lambert-Beer restrictions? Detuning away from absorption maxima in an attempt to bring the signal down is a hazardous enterprise in my opinion when interpreting sedimentation equilibrium records: I would doubt the reliability of the data recorded at 250 and 300nm. How was thermodynamic non-ideality corrected for?.

What was the partial specific volume and how was it determined?. Was glycosylation taken into account?

In many case data is quoted to 4-5 significant figures (Tables 2,3) with no error.

Minor comments

1. Protein names are conventionally lower case: Sortilin should be sortilin throughout (except where it is the start of a sentence), suggest sSortilin is s-sortilin.

2. Line 515 and following. The modern convention is MALS (multi-angle light scattering) not MALLS as it is taken as a matter of course that lasers are being used.

I hope these comments prove helpful

Stephen Harding

Reviewer #5 (Remarks to the Author):

In this manuscript authors present crystal structures of the luminal part of the transmembrane receptor Sortilin (sSortilin) at two different pH ranges; pH 7.0 – 7.5 ("neutral pH") and pH 5.0 – 6.2 ("acidic pH"). At neutral pH, sSortilin predominantly exists in a monomeric form. At acidic pH, sSortilin changes conformation and forms a homodimeric species. These pH-induced changes in states were confirmed using a broad spectrum of biophysical techniques and the presence of the dimeric state was shown in cells using the proximity ligation assay (PLA).

The manuscript is well written and the authors have done a thorough analysis of the two different sSortilin forms observed at acidic and neutral pH. It is beyond doubt that sSortilin undergoes pH-induced conformational changes, which makes the model for ligand uptake at neutral pH and ligand release at acidic pH (e.g. within endosomes and lysosomes) very convincing. Also the dimerization of the sSortilin in acidic pH can provide a trigger for receptor recycling. These novel and attractive mechanisms could be also relevant for other receptors involved in internalization and endocytosis. Thus this study deserves publication, but there are a number of comments, that should be addressed before publication:

1. I would prefer to see RMSZ in the Table 1 instead of RMSD of bond length and angles, the Ramachandran outliers and rotamer outliers expressed in number of residuals instead of percentage. Also it is a good practice to add MolProbity score of each structure as the bottom row.

2. I would advise the authors to change the coloring of their structure models within the figures; it is difficult to distinguish between the pink vs red colors used for monomer and dimer, respectively. Also the blue vs light-blue color are too similar to visualize the differences between monomer and dimer structure.

3. Figure 1 nicely illustrates the relative position of the sSortilin domains, also in respect to the

Cell surface. If visually possible, it would be nice to add the binding site of Neurotensin, which is only detailed in Figure 7. Alternatively, a zoomed-out sSortilin + Neurotensin complex structure could be presented in Figure 7.

4. Similar to point 3., the N-glycosylation sites (or at least N241 located at the dimer interface) could be indicated in one of the figures (e.g. Figure 1)

5. For obtaining crystal forms 2, 3 and 4, sSortilin was initially co-crystallized with proneurotrophins (Material and methods; lines 484-486). However, these proneurotrophins are not mentioned in the crystallization conditions. This seems somewhat confusing; did sSortilin crystallize in the presence of proneurotrophin (which one(s)?) and the complex was not observed in the crystal structure?

6. Figure 5A shows elution profiles for sSortilin in two different pH's, elution profiles for the sSortilin in the same condition also shown in Sup. Figure 2. However the retention volumes are very different in these two figures. I guess that the elution profiles in Figure 5A (and Figure 6A) are from Superose 6 10/30 column that was used in SEC-SAXS experiments and elution profiles in Sup. Figure 2 are from Superdex 200 10/300 used in SEC-MALLS experiments, but this is not mentioned in the main text or figure legends.

7. According to my knowledge, the native MS profile does not provide a direct quantitative information regarding abundances of different protein species, since the ionization efficiency of different protein species may not always be the same in different conditions. Also, ionized spray may disturb monomer-dimer equilibrium not in the same way at pH 5.0 and at pH 7.5. So the difference in MS profiles that shown in Figure 5G by themselves don't confirm "that Sortilin has a propensity to dimerize at acidic pH" (line 244-246). However this conclusion can be drawn based on comparison of MS profiles from the WT protein with the profiles from A464E mutant (Figure 6I).

8. Sup. Figure 3 shown the MS profile of sSortilin produced in HEK293-E cells that was purified and measured at pH 7.5 in contrast to the sSortilin produced in HEK293-ES (Figure 5G) and A464E mutant (Figure 6I). It is not clear for me why the protein was purified at pH 5.5 for a MS measurement at pH 7.5 (Figure 5G and 6I) and where is this purification procedure mentioned?

9. It will be appreciated if authors will include MS profile of the sSortilin produced in HEK293-E cells that was purified at pH 5.5 and measured at pH 5 in Sup. Figure 3, as it was done for sSortilin produced in HEK293-ES (Figure 5G) and A464E mutant (Figure 6I).

10. The distributions of the number of PLA events per cell in Figure 8C do not look like normal distribution to me. Did the authors try to test the normality of the distributions? If the distributions are not normal I would suggest to use non-parametric test (for example Mann-Whitney test) for proper sample comparison.

11. Lines 327-328 the authors state "The value that we determined for NGF-Sortilin interaction agrees with that of Nykjaer et al. and that of proNGF with Sortilin is similar to that of Feng et al". However in both references the affinity of the Sortilin to proNGF is much higher than the affinity to NGF. In the experiment presented by the authors it is other way around. The reason could be the different source of the proteins as authors suggested, or the experimental setup. It is hard to say without extra controls or alternative binding experiments.

12. Weakened affinity of A464E mutant to the ligands at pH 5.0 compared to pH 7.4 authors interpreted as evidence that "dimerization of Sortilin at acidic pH prevents ligand binding" (lines 334-338). However the WT sSortilin at pH 5.0, according to the authors, interacts unspecific with the chip surface, and it is shown to be mostly in dimeric form at this pH. The A464E mutant from the other hand has substantial monomeric fraction in the acidic pH. So weakened affinity could be due

to the fact that dimers interact in an unspecific way with the chip and that only monomeric fraction binds specifically to the ligands.

13. Authors didn't show the dissociation part of the SPR experiment so the release of the ligand in different pH's can't be evaluated.

14. Taking in account comments 11-13 I don't think that these SPR experiments are adding any useful information to this manuscript.

15. Authors state that dimerization and conformational change provide a double mechanism for ligand release at acidic pH. However I don't think they show enough evidence for this statement. The conformational change at acidic pH seems to be sufficient, based on the presented data, to explain the ligand release at acidic pH.

16. Lines 113-115 "In addition, the pH-induced dimerization brings the sSortilin C-termini and the cytosolic segments in close proximity which could provide the signal for cytosolic adaptor proteins to shuttle Sortilin to various intracellular compartments". I assume that the meaning is that cytosolic segments come to the close proximity to each other due to sSortilin dimerization and this provide the signal for cytosolic adaptor proteins.

Point-by-point response to comments and suggestions on manuscript NCOMMS-17-02577

Reply to reviewers' comments:

Reviewer #1 (Remarks to the Author):

Janssen and co-workers study dimerisation of the transmembrane receptor Sortilin. Crystal structures reveal pH-dependent conformational changes and dimerisation of the protein at low pH. They further use various biophysical techniques to describe Sortilin dimerisation and ligand binding which is favoured by the monomeric protein. The manuscript is rather lengthy and the experiments are not clearly described. I suggest the following changes before considering publication of the manuscript.

Abstract

Q: Line 15-17: It is difficult to understand this sentence.

A: To clarify this we have changed this sentence from “The transmembrane protein Sortilin, a β -propeller containing endocytosis receptor, internalizes a diverse set of ligands, with roles in cell differentiation and homeostasis, and is recycled to the *trans*-Golgi network or cell surface” to “The transmembrane protein sortilin, a β -propeller containing endocytosis receptor, internalizes a diverse set of ligands with roles in cell differentiation and homeostasis.”.

Introduction

Q: The introduction is rather lengthy. I suggest to shorten the introduction and only describe the facts that are important for the understanding of the manuscript.

A: To address this we have shortened the introduction substantially. More specifically we have removed the following sections: “A new role for Sortilin in cancer via export of the TrkB-EGFR-Sortilin complex (TES, Tropomyosin receptor kinase B- Epidermal Growth factor Receptor-Sortilin) and regulation of Sonic Hedgehog (Shh) has recently been reported 18,19” and “e.g. lysosome targeting, recycling to the cell surface and cycling between the TGN and endosomes 15,21,20. Deletion of the Sortilin cytosolic tail or sorting-motif mutations modulate Sortilin endocytosis and shuttling 20,22. The adaptors GGA1 and retromer complex have been shown to homodimerize at high concentrations in solution 23–25 and this dimerization has been suggested to be important for binding adaptor cargo such as Sortilin 25” and also “In addition, it has been suggested that glycosylation of the luminal segment of Sortilin plays a modulating role in sorting ligands; a highly glycosylated form of Sortilin is responsible for neurotensin endocytosis from the cell surface and a less glycosylated, intracellular form, is responsible for the sorting of internalized neurotensin to the TGN 6”.

Q: I suggest to change some terms, e.g. unliganded (line 31) and endocytosed (line 44). Even though these terms exist they sound a bit stiff in the context.

A: We have changed “unliganded” (line 31) to “free” and “endocytosed” (line 44) to “internalized”

Q: Line 97: “interactions for several” change to: “interactions with several”

A: This change has been addressed as requested.

Results

Q: First paragraph (page 4-5): The crystal structure is described, however, the resolution which is an important result when discussing crystal structures is missing. Only r.m.s.d. values are given.

A: To address this we have included the maximum resolutions of the datasets in the first sentence of the results: “We determined the structures of the luminal segment of mouse Sortilin, s-sortilin, from crystals grown at neutral pH (pH 7.5, one crystal form, 2.1 Å maximum resolution) and acidic pH (pH ranging from 5.0 to 6.2, three crystal forms, maximum resolution ranging from 2.3 to 4.0 Å) (Fig. 1, Table 1 and Suppl. Fig. 1).”

Q: Line 138-140: The discussion of the results is confusing. Please clarify.

A: To clarify this we have changed this sentence from “β-Propeller blades 1,4,6,7,8,9 and 10 are involved in dimer formation with blade 1 interacting with 7 and 8, blade 4 interacting with 6, and blade 9 interacting with 10 between the dimer chains” to “β-Propeller blades 1,4,6,7,8,9 and 10 are involved in dimer formation. The two-fold symmetry axis that describes the s-sortilin homodimer passes through the dimer parallel to the dimerization interface and exits at blades 4 and 5 on one side and blades 9 and 10 on the other side of the dimer. As a consequence, the following β-propeller blades interact with each other across the dimerization interface: blade 1 interacts with blades 7 and 8 of the other chain, blade 4 interacts with blade 6 of the other chain, and blade 9 interacts with blade 10 of the other chain.”.

Q: Page 5, second paragraph: The explanation why dimeric Sortilin is oriented towards the cell-surface is not clear.

A: The full length sortilin is a type I transmembrane protein and is surface attached. To clarify this, we have changed the first sentence of this paragraph from “The sSortilin dimer structures reveal how Sortilin could be oriented on the cell-surface” into “The s-sortilin dimer structures reveal how the full-length transmembrane sortilin could be oriented on the cell-surface”.

In addition, we have rewritten the description of the possible orientation of sortilin on the cell-surface to clarify our reasoning further. We have changed the following section in this paragraph “The dimer crystal structures lack nine residues to the transmembrane helix and therefore the sortilin dimer is, most likely, oriented with the β-propeller facing perpendicular to the cell-surface. In such an orientation, the C-termini, the 10CC-b domains and β-propeller blades 4 and 5 are close to the membrane (Fig. 1b). Interestingly, the interface on the sortilin dimer that would be close to the cell-surface is lined ...” to “The dimer crystal structures lack nine residues to the transmembrane helix. Most likely the two-fold axis that describes the sortilin dimer is oriented perpendicular to the cell surface. In this orientation, the sortilin β-propellers face the cell surface in a perpendicular fashion and the C-termini, the 10CC-b domains and β-propeller blades 4 and 5 are closest to the cell surface whereas blades 9 and 10 would be furthest away from it (Fig. 1b). Interestingly, the interface on the sortilin dimer that faces the cell-surface in this proposed orientation is lined ...”

Q: Page 6, first paragraph: The changes between monomer and dimer are rather small. Can these really be considered as conformational changes between the two forms? Please explain/discuss.

A: The supplementary morph movie shows the extent of the conformational changes in the monomer and the dimer. In particular for the β-propeller these changes are unprecedented as this

domain is believed to be a stable fold. These changes can indeed be considered a conformational change as all four independent dimer structures are very similar to each other and all five monomer structures are very similar to each other (as for example shown in supplementary figure 1). We used the center of mass of the blades to calculate these changes in an unbiased manner. In a similar fashion, the distance between two centers of mass of monomers differ by at most 0.3 Å versus the 2-3 Å differences we observe between the monomer and dimer structures.

Q: Page 7, line 209, 214 and 216: “predicted to be neutral” – please give a reference for this.

A: This was predicted with the PROPKA software, which is reference 29 in the methods section. To clarify this in the results section reference 29: “Czodrowski, P., Dramburg, I., Sotriffer, C. A. & Klebe, G. Development, Validation, and Application of Adapted PEOE Charges to Estimate p K. 437, 424–437 (2006)” was added as requested.

Q: Page 8, first paragraph: Include how/which experiments were performed. It is difficult to follow the text without mentioning the techniques employed.

A: To address this we have added which experiments were performed at the relevant places. More specifically we have changed the first two sentences of this paragraph “We verified that sSortilin undergoes a reversible, pH-induced, monomer-dimer transition in solution (Fig. 5). At pH 5.5, the monomer-dimer equilibrium is shifted towards dimer; at similar concentration the size exclusion retention volume of sSortilin at pH 5.5 is decreased compared to its retention volume at pH 7.4 (Fig. 5a).” to “At pH 5.5, the monomer-dimer equilibrium is shifted towards dimer; at similar concentration, the size exclusion chromatography (SEC) retention volume of s-sortilin at pH 5.5 is decreased compared to its retention volume at pH 7.4 (Fig. 4a).”

Q: Line 238: Suddenly glycosylation is discussed. Please introduce this at an earlier point to help understanding why this is mentioned here.

A: To clarify this we now mention in the introduction that the sortilin luminal segment is N-linked glycosylated and in the first sentence of the results that we determined the structures of the glycosylated luminal segment of sortilin.

Q: Line 243: Please delete “biomolecular”.

A: We have now deleted biomolecular as requested.

Q: Line 244/245: “Similar injection concentrations” – please clarify. Was the protein concentration similar or the amount in the electrospray emitter?

A: The experiments were done at similar protein concentrations. To clarify this, we have changed “At similar injection concentrations” to “At similar protein concentrations”.

Q: Page 8, last paragraph: Deglycosylation of Sortilin might already affect its dimerisation behaviour. The naturally modified protein might be a better measure to show pH-dependence.

A: We agree that our glycosylated construct is a better measure to show pH-dependence compared to the deglycosylated form. The sedimentation equilibrium analytical ultracentrifugation, size exclusion chromatography small angle scattering analysis and multi angle light scattering experiments all show the pH-dependent dimerization behaviour of glycosylated s-sortilin. Only for the sedimentation equilibrium analytical ultracentrifugation we have also used deglycosylated s-

sortilin, in addition to the glycosylated form, this shows that the glycosylation may stabilise the dimer form as deglycosylated s-sortilin has less propensity to dimerize.

Q: Line 257: KD of 4 μ M. Please clarify which KD? Dimeric species?

A: We were referring to the KD of dimerization. To clarify this, we have changed “a KD of 4 μ M” to “a KD of dimerization of 4 μ M”.

Q: Page 10, first paragraph: Native MS experiments of the mutant could also be used for quantification.

A: We have provided relative abundances of the monomer versus the dimer derived from the native mass spectrometry data by comparing s-sortilin wild type at pH 7.4 with pH 5.0 both at the same concentration (inset in figure 4C) and by comparing s-sortilin A464E at pH 7.4 with pH 5.0 at the same concentration (figure 4D). This shows how dimerization is dependent on the pH.

We have not attempted to determine K_D values with native mass spectrometry as such approaches often suffer from gas phase artefacts, most importantly different ionization efficiencies of the different species (as correctly pointed out by reviewer 5). Moreover, MS-based K_D determination would require spraying s-sortilin at a wide range of concentrations, which we found is not possible for s-sortilin. Instead, as reported in the manuscript, we determined the K_D of homodimerization in solution with the widely used sedimentation equilibrium analytical ultracentrifugation technique.

Q: Line 308 and 310: “in the presence “. Please correct.

A: To correct this we have changed “in presence” to “in the presence”.

Q: Line 319: proBDNF – please introduce abbreviations. Also check the entire manuscript, there are more abbreviations used which are not introduced.

A: To address this we have added the abbreviations of proBDNF and NGF when referred to for the first time as in “the binding of proBrain-Derived Neurotrophic Factor (proBDNF) to s-sortilin” and “Nerve Growth Factor (NGF)”.

Q: Page 11, second paragraph, Line 346 etc.: 40 nm distance is quite large compared to the distances that were considered before as significant changes. Tags are usually rather flexible and misleading interactions might be observed.

A: With this assay, we are probing dimer versus monomer sortilin. Proximity Ligation Assay events will only be observed when two s-sortilin proteins are within 40 nm distance of each other. This is very likely the case for the s-sortilin dimer based on our structures. It is much less likely that two monomer sortilin proteins are within this 40 nm close contact. Nonetheless, as the reviewer points out, some interactions may still be observed by purely monomer protein. To address this, we have added to this paragraph the following: “Alternatively, it is possible that two monomer sortilin molecules are within 40 nm proximity to give rise to a PLA event and that the counts from the sortilin A464E mutant are mainly from monomer protein.”.

Discussion

Q: Page 12, Lines 381-382: Low resolution might also be a result of dynamic protein regions.

A: At this site, the electron density arising from the protein is well resolved and no dynamic protein regions are apparent at this site. To clarify this, we have changed “Poorly resolved electron density is

present in this mouse structure at the neurotensin binding site which may correspond to a low occupancy small molecule co-purified from the expression medium, but this has not been modelled (Suppl. Fig. 11).” to “Unmodelled poorly resolved electron density is present in this mouse structure at the well-resolved neurotensin binding site. This additional electron density may correspond to a low occupancy small molecule co-purified from the expression medium, but this has not been modelled (Suppl. Fig. 11).”.

Q: Line 401: change “it” to “its”

A: Done as requested.

Q: Line 343: change “from” to “form”

A: Done as requested in line 434.

Methods

Q: Line 652: “score cut” change to “score cut-off”

A: Done as requested.

Figures

Q: Figure 5G: The mass spectra look rather unresolved which is most likely due to the presence of glycoforms. However, the calculated error of the protein (complex’) masses are very low. I would expect a higher error for spectra of this quality. In addition, it is not really clear how the masses for the lightest and heaviest glycoforms were obtained. There are no additional peaks visible in these spectra! The quantification of the peaks was estimated, however, there are tools available for simulation/quantification of native MS data.

A: In electrospray ionization mass spectrometry (ESI-MS), every protein species is detected as a series of differently charged ions (charge state envelope). Based on the m/z position of every charge state, a molecular weight can be calculated. The reported error is the standard deviation of the molecular weights calculated from each single charge state in a charge state envelope. The error reflects the precision of the molecular weight assignment and a low error is indicative of a correct assignment. As correctly mentioned by the reviewer, the heterogeneity of the different s-sortilin glycoforms is reflected by the relatively unresolved peaks. However, since the glycan heterogeneity is constant for all detected charge states, the molecular weight can be determined with very high precision. Both the molecular weight and the standard deviations were confirmed with MassLynx v4.1. To clarify this, we have added the following to the Native Mass Spectrometry methods section: “The reported standard deviations of the molecular weights were calculated from the different charge states of the respective species.”.

To clarify how the masses for the lightest and heaviest glycoform were obtained we have added expanded views of the most abundant charge states for the monomer and dimer to supplementary figure 5. This shows the m/z peaks for the lightest and heaviest glycoform besides the more populated two main glycoforms. We have added a figure legend for this panel to supplementary figure 5 and refer to this figure panel in the main text.

We were able to calculate the relative abundancies of the monomer and the dimer directly from the raw mass spectra due to the fact that the monomer and dimer envelopes are well separated. As the

reviewer points out simulations are often used to aid MS-based quantification. Such simulations are based on peak fitting and subsequent integration to approximate the experimental mass spectrum. This is mainly necessary for complex mass spectra with overlapping charge state envelopes. In case of s-sortilin, however, the monomer and dimer envelopes are well separated, thus there is no need to simulate the data to derive information on abundances. Instead, we were able to directly calculate the relative abundances in the raw mass spectra from the ion current detected for the m/z range of the monomer and of the dimer (which corresponds to the area under the charge state envelope). This is an accurate approach to derive quantitative information from MS and it is also used in peptide-centric MS approaches (referred to as MS1- or XIC-based quantification). We referred to estimated relative abundances because native MS can only provide apparent relative quantities (e.g. due to different ionization efficiencies of the different species). To clarify that we have calculated (and not estimated) the relative abundances of the monomer and dimer in the raw mass spectra we have changed in the Native mass spectrometry methods section the following sentence “The mass spectra were analyzed using MassLynx v4.1 (Waters) and the relative abundances of s-sortilin monomer and dimer were estimated based on their respective extracted ion currents.” to “The mass spectra were analyzed using MassLynx v4.1 (Waters). As the monomer and dimer m/z envelopes are well separated the relative abundances of s-sortilin monomer and dimer in the native mass spectrometry data were determined from the extracted ion currents for the m/z ranges of the monomer and of the dimer, which corresponds to the area under the respective charge state envelopes.”.

Q: Figure 6: Please correct numbering of the panels (e.g. panel I missing from the figure legend). The resolution of these spectra is very low. Are there any reasons why this is the case compared with Fig. 5G? Are glycoforms present?

A: The numbering in the figure legend has now been corrected. The mass spectra have now been combined in figure 4, following other changes to the manuscript.

The difference in resolution of the mass spectrometry spectra is arising from the difference in protein glycosylation pattern when protein is produced in HEK293-E cells or in HEK293-ES cells (that are deficient in N-acetylglucosyltransferase). In the Protein expression and purification methods section we explain “HEK293-ES cells produce proteins with shorter, more homogeneous high mannose glycans (“short” glycan type), while HEK293-E cells produce native-like protein with hybrid glycans (“native” glycan type)”. Also in supplementary figure 3 we emphasize the different glycosylation patterns depending on the s-sortilin source. To clarify this difference in figure 4C and figure 4D we have now added to the figure legend of 4C “resulting in shorter, more homogeneous oligo mannose glycans” and “resulting in larger, less homogeneous hybrid glycans”.

Q: Supplementary Figure 3: Please correct reference to Fig. 6I and 5G. I disagree that this mass spectrum shows broader peaks than the one shown in Figure 5G. The peak width seems to be the same, however the spectra in Fig 6I and Supp Fig 3 are less resolved.

A: The references to figure 6I and 5G have now been corrected. Due to other changes to the manuscript, these figures are now respectively figures 4D and 4C.

We agree with the referee that the wording here is confusing. To address this, we have changed the figure legend of supplementary figure 3 (now supplementary figure 5) from “and shows much broader peaks compared to sSortilin produced in HEK293-ES cells” to “Compared to wt s-sortilin produced in HEK293-ES cells, less well resolved peaks and higher molecular weights are observed for s-sortilin produced in HEK293-E cells, confirming the presence of longer and more heterogeneous glycan trees”.

Q: Supplementary Figure 5: Lots of the assigned peaks seem to be in the background.

Glycopeptides tend to show non-optimal fragmentation behaviour, complicating the distinction of very low abundant fragment ions from background noise. To address this challenge, the annotation was performed using the software program Byonic v2.6 with automated spectra quality control to disregard low-quality spectra. The spectra shown in Supplementary Figure 5 (now supplementary figure 7) were automatically annotated by Byonic. Although Byonic can be used to analyse any kind of bottom-up proteomics data, it contains several dedicated features to facilitate high-fidelity glycan mapping (e.g. by taking into account all characteristic glycan fragment ions). To further increase identification confidence, we applied (1) a target-decoy approach using a strict false-discovery rate (FDR) cut-off of 1% and (2) a strict fragment ion tolerance of 20 ppm, ascertaining that even low abundant fragment ions are highly accurate in mass. Taken together, our identifications are backed up by highly accurate peptide and fragment ion masses, reported at score cut-off corresponding to a strict 1% FDR, and additionally validated by manual inspection of the fragment ion spectra. To emphasize the quality control, we have changed in the MS-Based Glycan Mapping methods section “Identified peptides were filtered using an automatic score cut-off and all MS2 spectra representing glycosylated s-sortilin peptides were manually verified.” to “Identified peptides were filtered using an automatic score cut-off and are reported at 1% false-discovery rate. In addition, all MS2 spectra representing glycosylated s-sortilin peptides were manually verified.” The narrow fragment ion tolerance and the use of automated spectra quality control are already reported in the MS-based Glycan Mapping section as in “The MS data were analyzed using Byonic v2.6 (Protein Metrics), allowing 10 ppm precursor mass tolerance and 20 ppm fragment mass tolerance and forcing the software to skip low-quality mass spectra.”.

General comments

Q: It is not clear what the function of dimeric Sortilin is. If this is considered the active form it is surprising that ligands only bind the monomeric form. Please discuss.

A: We are the first to report that sortilin dimerizes. In addition, we show that sortilin dimerizes more readily at low pH. Others have shown that sortilin is important for endocytosis of ligands. Our data and that of others reveal that the monomer form of sortilin binds ligands and that the dimer form does not. Taken together this indicates that the monomer form of sortilin can bind ligands outside of the cell where the pH is neutral. This leads to internalization of the sortilin-ligand complex. The ligands are released from sortilin due to the sortilin dimerization and conformational change induced by the low pH in the endosomes. So both monomer and dimer forms are relevant for the function of sortilin. The function of monomer and dimer sortilin is summarized in figure 8 (previously figure 9). To emphasize the role of monomer and dimer form of sortilin further we have changed the legend of figure 8 from “At the cell surface, sortilin is in an equilibrium between monomer and dimer. Upon endocytosis, sortilin dimerizes due to the acidification along the endosomal pathway and thereby releases its ligand” to “At the cell surface, sortilin is in an equilibrium between monomer and dimer, but ligands are bound preferentially to the monomer form. Ligand bound sortilin is internalized. Upon endocytosis, sortilin dimerizes due to the acidification along the endosomal pathway and thereby releases its ligand”.

Q: The manuscript is rather too long. Too many figures.

A: To address this we have moved figure 3 to the supplementary figures. We have changed the numbering of the figures and supplementary figures throughout the manuscript.

Reviewer #2 (Remarks to the Author):

Low pH-induced conformational change and dimerization of Sortilin triggers endocytosed ligand release.

Leloup et al., 2017.

The subject: The X-ray crystal structure of sortilin at two different pH's (acidic and neutral) and the biophysical/structural characterisation of sortilin self-association, specifically the exchange between monomers and dimers, induced by pH changes or the binding of effector ligands.

Key finding: The luminal segment of Sortilin undergoes conformational changes and dimerization on lowering the pH and, to quote the authors "sortilin dimerization and conformational change discharges ligands and triggers recycling."

Q: In general, the investigation by Leloup et al is well written and the results presented are quite interesting. However, there are many questions that remain unanswered. These questions may, or may not, be resolved by providing more specific descriptions and analyses of the results as presented, but will likely also require additional experiments. Importantly, and especially for the SAXS sections of the manuscript, portions of the analyses are missing. For example, the analysis of the SAXS data in terms of mixtures is extremely basic and needs to be well extended so as to offer the reader models of the sortilin monomer/dimer system that fit the scattering data as opposed to what is currently presented in the manuscript, i.e., models that absolutely do not fit the SAXS data. In essence, the SAXS sections can be simply boiled down to a single statement:

'We observe a decrease in the Porod volume of the sortilin mixtures on increasing the pH from 5.5 to 7.4, mutating A464 to E, or the addition of neurotensin, suggesting some-sort of disassociation of the sortilin dimer.'

Yet, is it not the case that the SAXS data analysis and interpretation is kind-of the linchpin that brings together all of the available data into a coherent structural narrative that supports the key findings? What seems a little strange is that the authors have a great deal of information to perform a really good SAXS-based analysis – crystal structures, binding constants, MW estimates of both the glycosylated and deglycosylated forms of the protein, mass spec data, etc, etc, etc. Why not integrate this information into the SAXS data interpretation and generate solution-state models of the sortilin mixture system that fit the SAXS data (be they ab initio model mixtures or rigid-body models, etc)? There are plenty of SAXS modelling programs that can do this (e.g., as part of the ATSAS suite there is OLIGOMER, GASBORMX, SASREFMX, GLYCOSYLATION (if you want to build glycans onto your X-ray crystal structures), even single value decomposition approaches.)

A: We have now re-interpreted the SAXS results and based on this analysis substantially rewritten the SEC-SAXS results section and material and methods. As the reviewer suggests, we have put more care in extraction of the SAXS curves for the different species and in additional modelling. The reanalysis of the SAXS data shows that, as previously reported, s-sortilin undergoes a pH induced monomer dimer transition and that the A464E dimerization interface mutation shifts the equilibrium towards more monomers. Based on the relatively narrow and homogeneous elution profiles and the

stable scattering signal throughout the peak we have now identified the peak of s-sortilin at pH 5.5 as dimer and the peak of s-sortilin A464E as monomer both at pH 5.5 and at pH 7.4 (see also Figure 5). The SAXS signal throughout the peak of s-sortilin at pH 7.4 varies significantly and both the radius of gyration and the estimated mass decrease at higher retention volumes. The s-sortilin dimer present at pH 7.4 has a conformation that is different to the dimer at pH 5.5. Both *ab initio* bead modelling in DAMMIF and rigid-body modeling with CORAL indicate that both dimer conformations are compatible with the core dimerization unit consisting of the two β -propellers in our crystal structure but the positions of the 10CC-a and 10CC-b domains are different between the two SAXS determined pH structures and different from the crystal structures (even when including glycans). In addition, the SAXS data indicates that the conformation of the s-sortilin monomer at pH 7.4 is different to the conformation of the s-sortilin A464E monomer mutant at pH 5.5 (note that we did not observe evidence for the presence of monomers at pH 5.5 of the wt s-sortilin sample). This difference may be described as a difference in position of the 10CC-a and 10CC-b domain in the two SAXS-determined structures that is different from the one present in the crystal.

As the s-sortilin crystal structures are different to the SAXS determined solution states (even when including glycans) we could not make use of OLIGOMER. The dimer (at least at pH 5.5) does not seem to be in P2 symmetry which prevented the use of GASBORMX and SASREFMX. Instead we have used DAMMIF for *ab initio* bead modelling and CORAL for rigid body modelling.

*Q: As an aside, it is quite bizarre that of the biophysical methods used by the authors to structurally interrogate the sortilin system – X-ray crystallography, AUC, SPR, MS, MALLS/RI – it is only the SAXS section that *completely* lacks any kind of quality assurance. For example, the data, $p(r)$ profiles and Kratky plots lacks errors, there are no error estimates on the extracted structural parameters (R_g , D_{max} , V_p , etc), there are no statistical estimates of quality of the data-model fits or $p(r)$ vs r reciprocal space fits to the data (even the AUC gets a χ^2) and who know what statistical checks and tests were employed during data reduction, subtraction and processing to ensure that the SAXS profiles being analysed are in fact what the authors think they are analysing. So what happened here – SAXS is just some hand-wavy ‘well it is kind of a blob that looks like an elephant’ kind of thing? Please refer to, and take note of, the recommendations outlined by the wwPDB Small-angle scattering taskforce for the presentation and reporting of small-angle scattering data: Jacques et al (2012) Acta Cryst D68:620-626 <https://doi.org/10.1107/S0907444912012073>*

and:

Trewhella et al. (2013) Structure 21(6):875-881
<http://dx.doi.org/10.1016/j.str.2013.04.020>

A: We agree with the reviewer that the quality assurance concerning the SAXS data was missing in the manuscript. Errors and χ^2 values have been added where appropriate, as well as a more detailed SAXS table, Guinier plots, SEC-SAXS chromatograms, and all data necessary to gauge the quality of the SAXS data and analysis.

Q: Finally, why did the authors employ SEC-SAXS for this project? Based on the SEC-traces provided by the authors in the manuscript there is no real need for SEC. If anything using SEC-SAXS has enormously complicated the scattering analysis. If the authors just use boring old traditional SAXS on individual samples, through a concentration series (e.g., six protein concentrations), the mixture analysis would be greatly simplified, the data will be of a higher quality and there would be more

information from which to draw solid conclusions. Using regular buffer-sample-buffer-sample SAXS:

- 1)They can accurately assess the protein concentration for concentration-dependent estimates of MW from the forward scattering intensity, $I(0)$. If placed on an absolute scale, the glycan contributions to the scattering intensities can be assessed eliminating MW estimates from V_p (or other scattering invariant MW approaches).*
- 2)They can make sure they have the correct solvent for background subtraction, something they have not demonstrated in the manuscript for the SEC-SAXS.*
- 3)They can obtain R_g as a function of protein concentration.*
- 4)They can obtain MW (i.e., $I(0)$) as a function of protein concentration.*
- 5)They can obtain V_p as a function of protein concentration.*
- 6)They can perform SVD on all of this data to calculate pure monomer scattering and pure dimer scattering contributions from which they can also determine the volume fraction of monomers and dimers at each protein concentration and pH.*
- 7)Then it would be possible to directly compare the X-ray crystal structure of the monomer and dimer and monomer/dimer mixtures to the SAXS data to make sure that the X-ray crystal structures are not simply capturing some weird conformation (e.g., influenced by lattice/packing forces) that is not actually realised in solution.*
- 8)They can model using ab initio or rigid-body approaches against the concentration-series SAXS data in parallel, extract volume fraction information and obtain more data to support the conclusion that there are conformational changes occurring in sortilin on dimer disassociation.*
- 9)They can assess the effects of changing the pH in incremental steps (at the moment they are limited to 2 pH values – why not do smaller increments?)*
- 10)They can assess the effect of altering the neurotensin concentration on the association states or sortilin under various pH conditions.*
- 11)They can obtain low-resolution models of the sortilin/ neurotrophin system and how this is affected by pH, etc etc, to back up the SPR investigation.*
- 12)They can assess the effects of deglycosylating the protein on the pH dependent monomer-dimer equilibrium/ neurotensin interaction, etc.*
- 13)All of the above could be measured at a synchrotron SAXS beam in about, what, 2 hour, i.e., the time it takes for, 2 SEC runs (of course the analysis would take longer, but so much stronger)?*

A: We agree with the reviewer on the power of SAXS measurements on individual samples in batch. We initially attempted this for s-sortilin and found indications of aggregation even at low concentrations. For example, already at concentrations as low as 0.2 mg/mL at pH 7.4 the smallest radius of gyration determined from batch data was about 5 nm, which given the results from the SEC runs (with a maximum radius of gyration of 3.7 nm for s-sortilin dimer at pH 7.4) is clearly an aggregation artefact. Furthermore, the Guinier analysis also showed deviation from linearity. Given the sensitivity for SAXS for small amounts of large aggregates we used SEC-SAXS instead. As expected this problem was alleviated in the SEC-SAXS experiments. To address this in the manuscript we have added to the SAXS section in the methods the following “Interpretation of batch experiments of s-sortilin suffered from aggregation even at concentrations as low as 0.2 mg/ml. Given the sensitivity for batch SAXS for small amounts of large aggregates we used SEC-SAXS instead and the problem of protein aggregation was alleviated in these experiments”.

Additional short comments for consideration:

Introduction.

Appears to be well referenced.

X-tal crystallography.

Q: For the models determined using X-ray crystallography, especially when describing the shifts in the amino acid side chains, show the electron density.

A: To address this we have added a supplementary figure (supplementary figure 3) in which we show the electron density for all the amino acid side chain that we describe in figure 3.

It is stated that:

“These observations indicate that the large-scale pH-induced structural rearrangement in sSortilin may be the result of local, residue level, changes in charge.”

Q: Fine, but what happens to the bound solvent on monomer-dimer association/disassociation? Is there a possibility of a change in entropy as well on dimer formation?

A: From our crystal structure of the monomer at 2.1 Å we see that there are twelve bound water molecules that are freed upon dimerization of s-sortilin. Possibly more water molecules are freed as the 2.1 Å maximum resolution of the monomer data may limit the number of bound water molecules that can be identified. In terms of freeing bound solvent molecules, entropy is increased upon dimerization. On the other hand, flexibility of loops and side chains that become buried in the dimerization interface and thus become less flexible, decrease the entropy of the protein upon dimerization. From the structures, it is not possible to determine the net result on entropy for the dimerization of s-sortilin.

For the biophysical analysis in solution, including SAXS.

In the text it is mentioned that MALLS was performed. The MW correlation and the SEC traces are shown in supplementary Figure 2. A UV trace is shown in both Figure 5 and Supplementary Figure 2.

Q: 1) In the supplementary figure show the RI and MALLS traces. As the authors calculated the MW from RI and MALLS, show the data.

A: The RI and MALS traces are now shown in an added panel in the supplementary figure (now Supl. Fig. 4). The legend was adjusted accordingly.

Q: 2) For the SAXS, the authors must show must show the $I(0)$ trace (i.e., the $I(0)$ calculated from each individual subtracted frame taken during the time of the SEC-SAXS experiment). The issue is that in Figure 5A the dimer looks to be stable through the SEC column, whereas at pH 7.4, there is clearly poorly resolved dimers and monomers. However, In Sup Fig 2, the reverse is true - here there are well resolved monomers at pH 7.4 and poorly resolved dimers at pH 5. So what type of sample was used for the SEC-SAXS? If the authors do not show the $I(0)$ trace and the region of the trace that was used to generate their final SAXS data profiles, how does a reader know that they simply didn't selectively choose frames that were 'kind-of monomeric' at pH 7.4 and 'kind of dimeric' at pH 5.5 (BTW: a consistent R_g through a SEC-peak means nothing by itself. Refer to Box 1 of Nature Protocols 11, 2122–2153(2016). So the authors need to comment on what is clearly a dimer disassociation on the column into monomers, i.e., an inability to resolve monomers from dimers and how this might affect the interpretation of the SAXS data and that the SEC elution range chosen for the analysis encompasses either the dimer or the monomer.

A: The new figure 5 now shows more complete SEC-SAXS chromatograms and the issue of signal stability is explicitly addressed in the materials and methods section as followed: “Samples were buffer exchanged and the column was equilibrated with 1.5 CV to the corresponding buffer and a stable background signal was confirmed before measurement.”

Q: In Figure 5 B, how were the two SAXS profiles scaled relative to each other? In this representation The SEC-SAXS data for the pH 5.5 dimer looks to be under subtracted?

A: This figure has been replaced by a normalized Kratky plot whose features are described in the main text (see Fig. 5f and 5j). The “offset” at higher q most likely stems from the presence of glycans and flexible protein regions and not from under subtraction.

It is stated that:(line 234):

“indicative of more dimer at pH 5.5 compared to pH 7.4.”

Q: How much is ‘more’? The authors should attempt performing a mixture/oligomer analysis or maybe single value decomposition/principle component analysis to estimate the volume fractions of the monomer and dimers in both pH conditions. ‘More’ is not really all that quantitative?

A: We have replaced this statement in the manuscript with a more extensive analysis of our SEC-SAXS data as described in the “S-sortilin shape depends on pH” section.

Regarding the MW determined from SAXS - it is indeed true, as stated by the authors’ on line 238:

“The Mm based on the Vp is dependent on the shape, flexibility and glycosylation state of the protein and is not suitable here as an absolute measure, but can still be used as a relative indicator for sSortilin dimerization.”

Q: But how do you know this is the case? Put a reference to this statement? Were attempts made to obtain the the MW from I(0) (the forward scattering intensity) and the concentration estimated by UV to experimentally back up this statement?

A: We have replaced the discussion of volumes by a more extensive analysis of the SEC-SAXS data. We have now also determined the MW from the UV and SAXS data as described in the “S-sortilin shape depends on pH” section. This analysis is consistent with s-sortilin being a dimer at pH 5.5 and the mutant s-sortilin A464E being a monomer both at pH 7.4 and pH 5.5.

Q: Why not incorporate the volume fractions observed from native MS into the mixture analysis of the SAXS? Why not incorporate the Kd determined from AUC into a mixture analysis for SAXS?

A: This is indeed a valuable suggestion. However s-sortilin at pH 7.4 is not at equilibrium during the SEC run as the dimer and monomer species are being separated. This will hamper a reliable use of the Kd in such an analysis.

Q: Does the calculation of a p(r) vs r distribution for a mixture make sense?

A: This is actually not an issue for the updated manuscript. Pair distribution function of mixtures indeed makes sense (and is even common for vesicles, etc.), but of course one would need to keep in mind that it is the $p(r)$ of a mixture when interpreting it.

The authors state on line 240:

“In addition, the fit of the calculated scattering curves from the sSortilin crystal structures to the SAXS

data is best for the monomer to the SAXS data at pH 7.4 and for the dimer to the SAXS data at pH 5.5 (Fig. 5c and d)."

Q: Yeah, fine, but what do you mean by 'best'. SAXS does have a level of scientific integrity attached to it. Sweeping terms like 'more' and 'best' can be summarised in terms like the χ^2 discrepancy, or Correlation Map p-values if the experimental errors are not available or miss-specified (Nature Methods (2015) 12, 419–422). It is not sufficient to state in Crystallography 'the best model kind-of fits the diffraction data to a better resolution than we thought it would' so why are the terms 'more' and 'best' okay for SAXS?

What is the R_g of the monomer and the R_g of the dimer calculated from the X-ray crystal structure monomers and dimers? How does this compare with the experimental R_g from the SAXS and what does this mean? For example, what is the effect of the glycosylation on the fits to the SAXS data? Are the authors fitting the correct models to the data? It is entirely possible to build glycans attached to X-ray crystal structures and assess the effect of their addition on the modelled SAXS profiles and the fits to the experimental data.

A: The manuscript now shows the prediction scattering curves of both the monomer and the dimer and addresses the fact that the known glycosylation does not affect them much. We have now added in the SEC-SAXS material and methods section the following: "The models were averaged, aligned and compared using DAMAVER (Volkov & Svergun, 2003, <https://doi.org/10.1107/S0021889803000268>). As the differences between the predicted scattering curves (from WAXSiS, Chen et al. 2014, doi: 10.1016/j.bpj.2015.03.062, Knight et al., 2015, doi: 10.1093/nar/gkv309) of the monomer s-sortilin crystal structure and the monomer sub-unit of the dimer s-sortilin crystal structure, as well as the differences between models with and without added glycans, were negligible in comparison to those observed experimentally at different pHs (data not shown), the structural changes within the β -propeller domain were ignored for SAXS modeling". We also list the expected R_g and D_{max} values for both structures in the SAXS table.

Q: Why did the authors not attempt ab initio bead modelling, for example using GASBOR that can take into account monomer-dimer association/disassociation?

A: As the reanalysed SAXS curves are representing pure states, we now discuss (DAMMIF based) bead modelling as detailed in the "S-sortilin shape depends on pH" section and in the materials and methods.

Q: Why did the authors not try SASREF, i.e., using their crystal structures to perform rigid-body model refinement against the SAXS data under monomer-dimer equilibrium conditions? At least from this that might obtain the volume fractions of the monomer and dimer from their SEC-SAXS data, or even systematically evaluate the monomer/dimer contributions through the SEC peak?

A: We have addressed this issue by CORAL based modelling, to also take into account parts that are missing in the crystal structures.

Q: For all SAXS data.

1) Show the data points (i.e., $I(s)$) with the errors, both on the subtracted profiles and the Kratky plots.

2) Show the errors on the $p(r)$ profiles - you must reference which program you used to perform the inverse indirect Fourier transform.

3) Show the Guinier plots for all constructs in the appropriate sR_g range.

- 4) Reference which program was used to perform the model-fits to the SAXS data.
5) Deposit the data and models into a SAXS-database, for example SASBDB (sasbdb.org).

A: Points 1-3 have been implemented as requested (see Fig.5).

Point 4 is now described in the Material and Methods section: “The intensities were scaled such that the forward scattering corresponds directly to the concentration (in mg/mL) times the molar mass (in kDa) of idealized proteins, i.e. 1 a.u. = $8.03 \cdot 10^{-4} \text{ cm}^{-1}$, unless explicitly stated otherwise. 2400 frames (1 s each) were collected per 40 min run. Data reduction was performed automatically using the EDNA pipeline (Brennich et al., 2016, DOI: 10.1107/S1600576715024462). Frames in regions of stable Rg were compared with CORMAP (Franke et al., 2015, doi:10.1038/nmeth.3358) to ensure signal stability in these ranges and 10 to 20 frames with good signal to noise were selected and averaged using PRIMUS (Petoukhov et al., 2012; doi: 10.1107/S0021889812007662) to yield a single averaged frame corresponding to the scattering of an individual SEC species. Protein concentrations were estimated based on the absorbance at 280 nm assuming a molecular extinction coefficient of $103 \text{ M}^{-1} \text{ cm}^{-1}$ for the monomer and of $206 \text{ M}^{-1} \text{ cm}^{-1}$ for the dimer. Pair distance distribution functions were created with GNOM (Svergun, 1992, <https://doi.org/10.1107/S0021889892001663>) and used to calculate 40 ab-initio models in C1 symmetry with DAMMIF (Franke & Svergun, 2009, doi: 10.1107/S0021889809000338). The models were averaged, aligned and compared using DAMAVER (Volkov & Svergun, 2003, <https://doi.org/10.1107/S0021889803000268>). As the differences between the predicted scattering curves (from WAXSiS, Chen et al. 2014, doi: 10.1016/j.bpj.2015.03.062, Knight et al., 2015, doi: 10.1093/nar/gkv309) of the monomer s-sortilin crystal structure and the monomer sub-unit of the dimer s-sortilin crystal structure, as well as the differences between models with and without added glycans, were negligible in comparison to those observed experimentally at different pHs (data not shown), the structural changes within the β -propeller domain were ignored for SAXS modeling. Missing residues were added to the monomeric crystal structure with psfgen (Humphrey et al., 1996, <https://www.ncbi.nlm.nih.gov/pubmed/8744570>) and the resulting structure was relaxed using the energy minimization tool of sassie-web (Curtis et al., 2012, doi: 10.1016/j.cpc.2011.09.010 ; Perkins et al., 2016, DOI: 10.1107/S160057671601517X). This structure was used as a starting point for rigid body modeling using CORAL (Petoukhov et al., 2012, doi: 10.1107/S0021889812007662) and ensemble based modeling using EOM (Tria et al., 2015, DOI:10.1107/S205225251500202X) at both pH 7.4 and pH 5.5. For the s-sortilin dimer structures, the relative positioning of the β -propeller domains was based on their arrangement in the crystal structure”.

Point 5: The SAXS data and models have been deposited in the SASBDB with following accession codes: SASDCW5 (Dimeric Sortilin at pH 7.4), SASDCX5 (Monomeric Sortilin at pH 5.5), SASDCY5 (Dimeric Sortilin at pH 5.5), SASDCZ5 (Monomeric Sortilin at pH 7.4)

Q: The Kratky plots are on a very odd scale. The s_{min} value of their experimental $I(s)s^2$ seems to limit to 0, when in fact it should be the case that the $I(0)$ (at $s = 0$) should occupy the 0,0 point

A: We have addressed this issue and updated the figure (see Fig.5).

Q: For the mA464 mutation section.

Again, why use SEC-SAXS on this? Batch mode SAXS at 8 sample concentrations would be a better approach - it would be possible to show the monomer-dimer equilibrium much more clearly in terms of MW from $I(0)$ and systematic changes in Rg vs concentration, etc.

A: Batch SAXS measurements of the s-sortilin A464E mutant showed clear hints of aggregation. We therefore chose to use SEC-SAXS experiments in which the aggregation issue was alleviated.

It is stated on line 283:

“However, at very low scattering vectors (0-0.1nm⁻¹) sSortilin A464E at pH 5.5 fits better to the calculated dimer scattering than to that of the monomer.”

Q: The issue is that there is no data (okay maybe a couple of points) in the s-range of 0-0.1nm⁻¹? Again define ‘fits better’ using some sort of statistical test.

A: We have replaced this comment in the manuscript by a reanalysis of the SEC-SAXS data as described in the “S-sortilin shape depends on pH” section.

The authors state on line 286:

“It is unlikely that this is due to the pH-induced conformational rearrangement within a monomer as the calculated scattering curves from a single sSortilin chain of the dimer and monomer crystal structures are identical, i.e. the conformation differences within a chain cannot be distinguished by SAXS analysis.”

Q: This is not the case at all. All because the calculated scattering intensities from the crystallographic models are similar does not mean SAXS is incapable of detecting conformational changes within monomers in the solution state (and once again how do you know the calculated scattering intensities from the monomer X-tal structures are in fact identical, if you do not provide a metric to evaluate this?) In fact one of the main strengths of SAXS is that X-ray crystal structures often do not fit the SAXS data because the crystal structure, by definition, only captures a single conformational state. As none of the X-ray crystal structures presented in this manuscript fit the SAXS data then the simple Occam’s razor principle applies, i.e., the X-ray crystal structure does not represent the solution state. And this is actually quite true because the effect of glycosylation has not been taken into account in the models when fitting the experimental SAXS data, nor the volume fraction weighted contributions of the glycosylated monomer and dimer. In other words, based on the presented data, for all we know the crystal structure does not reflect what is going on in solution and the monomers are actually sampling different conformational states as recorded in and captured by the SAXS data?

A: Indeed, we now argue that the position of the C-terminal 10CC-a and 10CC-b domains in solution depend on pH and are not well described by the crystal structure. We have added the following to the “S-sortilin shape depends on pH” section: “These findings indicate that the position of the C-terminal 10CC-a and 10CC-b domains are more flexible in solution. However, at pH 7.4 they seems to prefer out of plane positions whereas at pH 5.5 in plane positions are favored, implying that their observed positions in the crystal structures might be primarily due to pH change and only secondarily due to dimerization.

For the ligand binding investigation:

On Line 310 it is stated:

“Furthermore, SEC coupled SAXS analysis of wt sSortilin in presence of neurotensin at pH 7.4 showed a tendency for a smaller Rg (based on the Guinier plot and on the P(r)), and smaller Dmax. Thus, neurotensin stabilizes the monomer form of sSortilin and prevents dimer formation. It can be inferred from this data that pH induced Sortilin conformational change and dimerization will trigger release of neurotensin form Sortilin.”

Q: Where is the SAXS data and modelling to support this claim? Quoting Rg is insufficient as Rg is influenced by both size and shape. For example, if the neurotensin binds to the centre of the B-propeller, then you would expect Rg to go down as the Rg of a ring is larger compared to a disc with

an an equivalent diameter. So if neurotensin 'fills in the hole' then of course Rg goes down. It is necessary to do the data analysis.

A: The argument of Neurotensin changing the monomer-dimer equilibrium is now addressed on the more appropriate level of chromatograms. The elution peak from the SEC column has become more narrow for s-sortilin with neurotensin by reducing the size of the shoulder stemming from the s-sortilin dimer (Supplementary Figure 9). We also show that the SAXS curve of the monomer in complex with neurotensin does not change compared to the neurotensin free monomer, which given the relatively small size of neurotensin (13 amino acids) is not surprising.

Q: What happens at pH 5.5? Does the neurotensin bind? Surely this is interesting? In the discussion it is mentioned around line 396 that: "The conformational change at acidic pH is required for release of neurotensin and, probably, other peptide-based ligands such as spadin, whereas the dimerization may sterically hinder other Sortilin ligands (Fig. 9)." So show the SAXS data to support this very potentially-exciting claim?

A: We have not directly tested if neurotensin binds to sortilin at pH 5.5. We have shown that neurotensin binds to sortilin at pH 7.4. The neurotensin binding site undergoes a conformational change in the transition of sortilin from pH 7.4 to acidic pH. Others have shown that neurotensin competes with other peptide-based ligands such as spadin in binding to sortilin. Binding of spading to sortilin is substantially weakened at pH 5.0. We inferred from this data that the low pH induced conformational change of sortilin plays an important role in the ligand release.

Q: Why was the SAXS investigation not extended to the binding of sorting to NGF? Surely this would greatly support the SPr investigation? The following statement(s) are unclear:

"Unfortunately, the affinity of wt sSortilin to (pro)neurotrophins in acidic conditions could not be determined due to nonspecific binding of wt sSortilin to the SPR sensor surface....Taken together, our data indicate that dimerization of Sortilin at acidic pH prevents ligand binding."

A: We have not attempted SEC-SAXS measurements of sortilin NGF (or proNGF) complexes as the sortilin NGF (or sortilin proNGF) complex was not stable on the size exclusion column. Others have shown that ligand binding to wt sortilin is drastically reduced at acidic pH (Gustafsen, C. *et al.*, *Cell Metab.* **19**, 310–318 (2014); Petersen, C. M. *et al.* *EMBO J.* **18**, 595–604 (1999); Petersen, C. M. *et al.*, *J. Biol. Chem.* **272**, 3599–3605 (1997); Gustafsen, C. *et al.*, *J. Neurosci.* **33**, (2013); Conticello, S. G. *et al.*, *J. Biol. Chem.* **278**, 26311–26314 (2003)). These previous observations are detailed in the introduction. We show that the NGF, proNGF and proBDNF ligands can still interact with sortilin A464E at acidic pH although somewhat weakened and we show that sortilin A464 has still some, albeit much weaker, propensity to dimerize. To clarify how our data in combination with that of others support the model that dimerization of sortilin at acidic pH prevents ligand binding we have rewritten the last two sentences of the second paragraph of the "s-sortilin homodimerization prevents ligand binding" section to "The somewhat weakened ligand affinity for s-sortilin A464E at pH 5.0 compared to pH 7.4 may be due to the conformational change, the remaining albeit much reduced dimerization propensity, or both properties of s-sortilin A464E. Taken together, our data in combination with that of others that show drastically reduced wt sortilin – ligand interactions at acidic pH (Gustafsen, C. *et al.*, *Cell Metab.* **19**, 310–318 (2014); Petersen, C. M. *et al.* *EMBO J.* **18**, 595–604 (1999); Petersen, C. M. *et al.*, *J. Biol. Chem.* **272**, 3599–3605 (1997); Gustafsen, C. *et al.*, *J. Neurosci.* **33**, (2013); Conticello, S. G. *et al.*, *J. Biol. Chem.* **278**, 26311–26314 (2003)) indicate that dimerization and conformational change of sortilin at acidic pH prevents ligand binding".

For the SAXS methods.

Q: There is a requirement to be far more specific in the methods section for SAXS.

A: This methods section is now extended considerably.

Q: It is not possible to perform automatic merging of the SAXS data using GNOM (GNOME as listed by the Authors)

A: This statement in the methods section has been corrected.

Q: All because a set of subtracted SEC-SAXS profiles have a consistent R_g does not mean you are isolating a single species.

A: This is correct, and the new analysis indeed compares the complete SAXS curves.

Q: What was the buffer used for the SEC-SAXS data subtraction and how was it selected? How do you know that the correct buffer was used to correct for background scattering contributions? The scattering length density of a buffer can change during the course of a SEX run and it is entirely possible to detect the effects small-molecule like fractionation of the buffer components in the buffer scattering profiles.

What statistical analyses were performed to ensure that the individual subtracted SEC-SAXS data that were then scaled and averaged were intact statistically equivalent. Considering the poor separation of dimers from monomers at low pH (refer to Sup Figure 2) this could prove to be very important.

A: We now clearly state in the Material and Methods section that the stability of the background was confirmed before actual experiments were performed, that the background subtraction was done using the beamline's standard pipeline and that signal stability was ensured in the ranges that were averaged: "Data reduction was performed automatically using the EDNA pipeline (Brennich et al., 2016, DOI: 10.1107/S1600576715024462). Frames in regions of stable R_g were compared with CORMAP (Franke et al., 2015, doi:10.1038/nmeth.3358) to ensure signal stability in these ranges and 10 to 20 frames with good signal to noise were selected and averaged using PRIMUS (Petoukhov et al., 2012; doi: 10.1107/S0021889812007662) to yield a single averaged frame corresponding to the scattering of an individual SEC species."

Q: References are required for Guinier analysis and all of the program used to determine $p(r)$ vs r , the Porod volume, radial averaging routines, scaling etc, etc, etc.

A: References have been added as requested by the referee throughout the SEC-SAXS part in the main text and the materials and methods section.

Reviewer #3 (Remarks to the Author):

In this manuscript, Leloup et al presents a structural approach for determining the mechanism of pH-induced ligand release of the Vps10p family receptor, sortilin. The authors demonstrate a series of pH-induced conformational changes and homodimerization of sortilin, utilizing a variety of biochemical techniques to support their findings. These results are potentially significant, as sortilin shows a diverse array of ligand binding and is associated with multiple clinical pathologies. However, while the structural and in vitro data are quite convincing, whether the proposed mechanism exist in vivo is still unclear from presented data.

Q: 1. In vivo phenotypes of the A464E monomeric mutant should be probed. According to the

proposed model, this mutant should not release its ligand and thus defective in protein trafficking to lysosomes. Is it possible to test this hypothesis by assaying lysosomal trafficking of one of the sortilin ligands, such as progranulin (Neuron 68, 654–667 (2010))?

A: We have indeed not shown any *in vivo* phenotypes of the A464E monomer mutant. It may take a relatively large effort to get this *in vivo* mouse mutant set up. In addition, it is not clear what effect this mutation has on its *in vivo* expression level. For example, it may be that due to its defective recycling, sortilin A464E itself ends up being degraded in the lysosomes and is effectively removed from the cell surface. Note that we prevent internalization and trafficking effects from occurring in our cellular Proximity Ligation Assay by omitting the sortilin cytosolic tail from the constructs. We feel it may be a logical inference from our data and that of others that the sortilin dimerization and conformational change induce the ligand release. To emphasise that we have not shown *in vivo* that the release of ligands is induced by the sortilin dimerization and conformational change we have changed last sentence of the abstract from “More generally, this work reveals a novel double mechanism for low pH-induced ligand release by endocytosis receptors.” to “More generally, this work may reveal a novel double mechanism for low pH-induced ligand release by endocytosis receptors.”.

Q: 2. It is unclear whether conformational change alone is sufficient to disrupt ligand binding or whether dimerization is also required. Lower affinity binding of ProNGF and Pro-BDNF to the A464E mutant at pH 5.0 would suggest conformational changes at acidic conditions might also decrease the interaction between sortilin and its ligands.

A: We agree with the referee that this is a possibility. In addition, we show that dimerization at acidic pH is not entirely disrupted for the A464E mutant. The somewhat lower affinity for the proNGF, proBDNF and NGF ligands to s-sortilin A464E at pH 5.0 compared to pH 7.4 could be due to the conformational change or due to the remaining but much reduced dimerization propensity or both properties of s-sortilin A464E. To address this in the manuscript we have added to the second paragraph of the “s-sortilin homodimerization prevents ligand binding” section the following: “The somewhat weakened ligand affinity for s-sortilin A464E at pH 5.0 compared to pH 7.4 may be due to the conformational change, the remaining albeit much reduced dimerization propensity, or both properties of s-sortilin A464E.”.

Q: 3. The authors raise the question of whether sortilin dimerization plays a role in regulating its recycling and trafficking between cellular compartments. This is an interesting point and the logical progression is sensible. However, they do not attempt to address this in the manuscript. Interaction between WT vs A464E mutant of sortilin and GGAs could be tested and compared.

A: We have indeed not shown experimentally that the dimerization of sortilin at acidic pH affects interaction with GGAs, but infer this from our data. To emphasise that we have not experimentally shown an effect of sortilin dimerization on GGA interaction we have added to the second last paragraph of the discussion “but this hypothesis has not been experimentally verified” as in “In this mechanism, dimerization of sortilin would provide the trigger for shuttling of sortilin to different compartments (Fig. 8), but this hypothesis has not been experimentally verified.”

Q: 4. Pro-NgF is reported to bind to linear epitope on the surface of the beta propeller of sSortilin (J Biol Chem. 2010 Apr 16;285(16):12210-22), which is very different from neurotensin binding. This should be mentioned in the manuscript and discussed.

A: The linear epitope in blade 2 as identified by Andersen et al. is not located in the sortilin dimerization interface and is not changing its conformation substantially when comparing the monomer structure with that of the dimer. The preceding loop, residues 155 to 162, are changing

conformation due to dimerization as it is near the dimerization interface. The epitope is located about 28 Å away from the dimerization interface and the NGF part of proNGF is 60 Å long. It is currently not known how proNGF exactly interacts with sortilin but it is possible that residues that are buried in the dimer are contributing to proNGF binding in the sortilin monomer. To address this we have added to the discussion section the following: “A linear epitope in blade 2 of sortilin has previously been identified as a binding site for proNGF [J Biol Chem. 2010 Apr 16;285(16):12210-22]. This epitope is located 28 Å away from the dimerization interface and the NGF part of proNGF is 60 Å long. It is currently not known how proNGF exactly interacts with sortilin but residues that become buried in the dimer may contribute to binding proNGF in the sortilin monomer”.

Q: 5. Binding between sortilin and pro-NGF has been shown to be mediated by the pro-domain of NGF (Nature. 2004 Feb 26;427(6977):843-8; Neuropeptides, Volume 42, Issue 2, April 2008, Pages 205–214). Thus it is really puzzling that NGF has a binding KD lower than pro-NGF in Table 4. The reference for the relative affinity of the pro-NgF/NgF to sortilin is also mis-interpreted in page 11. In J. Mol. Biol. 396, 967–984 (2010) and Nature. 2004 Feb 26;427(6977):843-8, these affinity is of pro-NgF/NgF to sortilin/p75 complex rather to sortilin alone.

A: It has been shown by us and others (Nykjaer, 2004, Nature, 427:843-8 and Feng, 2010, JMB, 396:967–84) that both proNGF and mature NGF bind to sortilin, but the reported affinities differ substantially. Indeed, as the reviewer correctly puts forward, others have shown that proNGF has a higher affinity for sortilin compared to NGF whereas we find a higher affinity of NGF for sortilin compared to proNGF. To emphasise that we find this difference in contrast to what was reported earlier we have added to the second paragraph of the “s-sortilin homodimerization prevents ligand binding” section the following: “but in contrast to these earlier reports we find that NGF interacts with higher affinity to s-sortilin compared to proNGF” as in “The value that we determined for NGF-sortilin interaction agrees with that of Nykjaer et al. and that of proNGF with sortilin is similar to that of Feng et al., but in contrast to these earlier reports we find that NGF interacts with higher affinity to s-sortilin than proNGF does. The differences in affinity may come from differences in protein origin. We expressed and purified proNGF in HEK293 cells compared to proNGF produced in E. Coli and Sf9 cells, and we used mouse proteins instead of the human versions”.

Both Nykjaer, 2004, Nature, 427:843-8 and Feng, 2010, JMB, 396:967–84 report affinities of proNGF and NGF to sortilin alone (see left two top panels in figure 1a of Nykjaer et al. and figure 6a and 6c in Feng et al.). The affinity constants reported in these two papers are referred to in our manuscript in the second paragraph of the “s-sortilin homodimerization prevents ligand binding” section.

Q: 6. Also, in page 11, “Binding of NGF and proNGF to sSortilin A464E at acidic conditions was reduced by a factor of 2 compared to neutral pH while binding of proBDNF was reduced by a factor 4. Taken together, our data indicate that dimerization of Sortilin at acidic pH prevents ligand binding.” How can lower binding of pro-NGF and NGF to the monomeric mutant A464E at acidic mutant support the proposed model that dimerization of sortilin at acidic pH prevents ligand binding?

A: Others have shown that ligand binding to wt sortilin is drastically reduced at acidic pH (Gustafsen, C. et al., *Cell Metab.* **19**, 310–318 (2014); Petersen, C. M. et al. *EMBO J.* **18**, 595–604 (1999); Petersen, C. M. et al., *J. Biol. Chem.* **272**, 3599–3605 (1997); Gustafsen, C. et al., *J. Neurosci.* **33**, (2013); Conticello, S. G. et al., *J. Biol. Chem.* **278**, 26311–26314 (2003)). These previous observations are detailed in the introduction. We show that the NGF, proNGF and proBDNF ligands can still interact with sortilin A464E at acidic pH although somewhat weakened and we show that sortilin A464 has still some, albeit much weaker, propensity to dimerize. To clarify how our data in combination with that of others support the model that dimerization of sortilin at acidic pH prevents ligand binding we have rewritten the last two sentences of the second paragraph of the “s-sortilin

homodimerization prevents ligand binding” section to “The somewhat weakened ligand affinity for s-sortilin A464E at pH 5.0 compared to pH 7.4 may be due to the conformational change, the remaining albeit much reduced dimerization propensity, or both properties of s-sortilin A464E. Taken together, our data in combination with that of others that show drastically reduced wt sortilin – ligand interactions at acidic pH (Gustafsen, C. *et al.*, *Cell Metab.* **19**, 310–318 (2014); Petersen, C. M. *et al.* *EMBO J.* **18**, 595–604 (1999); Petersen, C. M. *et al.*, *J. Biol. Chem.* **272**, 3599–3605 (1997); Gustafsen, C. *et al.*, *J. Neurosci.* **33**, (2013); Conticello, S. G. *et al.*, *J. Biol. Chem.* **278**, 26311–26314 (2003)) indicate that dimerization and conformational change of sortilin at acidic pH prevents ligand binding”.

Minor notes

Q: 1. The authors incorrectly list the soluble secreted protein, progranulin, as a signaling receptor: “Sortilin interacts with a diverse set of ligands such as signaling receptors (e.g. TrkB, 81 EGFR, progranulin)...”.

A: We have corrected this by removing progranulin from the above mentioned list and by now referring to progranulin as a signaling protein as in “signaling proteins such as Sonic Hedgehog, (pro)neurotrophins, progranulin and neurotensin.”

Q: 2. Fig. 6G, very hard to distinguish three curves. Might want to change color.

Q: 3. Fig 6. Legend: format changes between lettering of each image. E.g. “A.”, “B.”, “C.”, “(D)”, “(E)”, etc.

A: Figure 6 has been modified upon reanalysis of the data suggested by reviewer 2. The two questions above have been alleviated by this modification.

Q: 4. Having a negative control in the PLA assay will be helpful determining the specificity of the assay.

A: Three different negative controls were performed and none of them showed any PLA events. To clarify this we have added to the “Fluorescence microscopy and in situ proximity ligation assay” in the methods the following “Negative controls, by omitting either one of the sortilin constructs or by omitting the primary antibody, did not show any PLA events”.

Reviewer #4 (Remarks to the Author):

Thank you for the opportunity of reviewing this paper by Janssen and coworkers. In my opinion this paper has the potential to represent a significant advance in our understanding of the molecular mechanisms underpinning endocytosis, and focuses on changes in the conformation state, and state of oligomerisation of the transmembrane sortilin system and its interaction with relevant ligands, controlled by pH. It is generally very well presented and illustrated. My concerns are with some important details with regards the techniques employed. Additional experiments though may be needed before this can be considered for publication in Nature Comms.

Major comments.

Q: p17 & Supplementary Fig 2. SEC MALS. Why is an old mini-Dawn (with 3 angles) and not a regular MALS device (Dawn, Helios) with the full angular complement being used? The molecules are high molecular weight (>100,000 g/mol), asymmetric structures and would be expected to show a

significant angular dependence, requiring an extrapolation to zero angle. Could the authors present this angular data in "Supplementary Figures" to give the reader confidence in the values claimed? line524. How was this dn/dc value "estimated": the value quoted seems high. How is glycosylation taken into consideration? This is critical because as i am sure the authors are aware, errors in molar mass/ conformational parameters are affected by the square of this parameter.

A: S-sortilin has an Rg smaller than 4 nm (determined with SAXS, as reported in the table 3 in the manuscript) and well below the typical minimum size of 10 nm at which angular dependence of the scattering would become useful in our setup to actually determine the Rg from the slope. To give the reader confidence in the values reported we have included the scattering intensities from the three measured angles at the highest protein concentration in supplementary figure 4 (right panels). The linear fit through the three data points is relatively flat indicating, as expected, an isotropic profile i.e. minimal angular dependence of the scattering. Please note that the average mass is determined from the average across the eluted SEC peak.

We calculated the dn/dc value based on the composition of the protein (i.e. the amino acids) according to Zhao et al. (Zhao et al, Biophysical Journal, 100, 2309-17) and taking into account 8.3 % average glycan mass, as determined from the native mass spectrometry experiments. The dn/dc of the protein part is 0.188 (91.7 %) and that of the glycans is 0.145 (8.3 %) giving the reported dn/dc of 0.185. We have used the differential refractive index monitor for quantitation of the protein concentration as stated in the methods section. In effect the molecular weight determination we employed, using both the light scattering and refractive index detectors, depends linearly on the dn/dc. To clarify how we have calculated the dn/dc of 0.185 we have changed in the method section "using an estimated dn/dc value of 0.185 ml/g" to "using a calculated dn/dc value of 0.185 ml/g, determined from a dn/dc of 0.188 for the protein part, a dn/dc of 0.145 for the glycans and 8.3 % glycosylation based on the native mass spectrometry data".

Q: p19. SPR. Values for Kd quoted correspond to one of the interacting species immobilized onto a surface. Did the authors check for anomalies? : What happens when s-sortilin is immobilized onto the surface and the ligands become the analyte? This would give us greater confidence in the values quoted.

A: We did indeed also perform SPR experiments with s-sortilin immobilized onto the surface and the ligands as analyte. However, nonspecific binding of the analytes to the sensor chip surface prevented us from interpreting these results.

Q: p18 & Supplementary Fig 4. AUC. Important detail is missing. Why was sedimentation velocity not used to check for sample heterogeneity?

A: We have now included sedimentation velocity experiments (supplementary figure 6A) with s-sortilin at three different concentrations (1 μ M, 2 μ M and 10 μ M). This shows that at 1 μ M one species is present corresponding to a s-sortilin monomer. At 2 and 10 μ M this species remains the most predominant one, as reflected by the peak area, but also a species at a higher sedimentation coefficient becomes apparent. The methods section and figure legend has been updated accordingly.

Q: lines566, 567 I am assuming the authors mean 2, 10 and 50uM and not mM. Even so, these are quite high concentrations (up to 5 mg/ml) and I am surprised the uv absorption signals stayed within range for many cases (although in some cases not so).

Why wasn't the more accurate interference optical system on the XL-I used (rather than noisy uv optics) which has no such Lambert-Beer restrictions? Detuning away from absorption maxima in an attempt to bring the signal down is a hazardous enterprise in my opinion when interpreting

sedimentation equilibrium records: I would doubt the reliability of the data recorded at 250 and 300nm.

A: The referee is correct that the concentrations reported in the methods for the AUC experiment were indeed micromolar and not millimolar. This has been corrected in the manuscript.

For particles absorbing light, like proteins, XL-A is very useful, because of its selectivity and the possibility to measure at different wavelengths. It is a standard procedure in sedimentation equilibrium experiments to use a combination of absorbance measurements at multiple wavelengths and different loading concentrations to increase the dynamic range of the concentration gradient detected in the SE experiment. We used scans measured at 250 and 280nm, which are at a minimum and maximum in absorption, respectively (see figure 1), to ensure that the absorption does not detune easily. We did not use the data collected at 300 nm in our global analysis, this has been corrected in the methods.

Figure 1. Absorbance of 50 and 10 μM s-sortilin in a 3mm light path AUC cell

To ensure the uv absorption signal stayed within a linear range we collected AUC data at 50 and 10 μM s-sortilin in 3 mm centerpieces and data at 2 μM s-sortilin in 12 mm centerpieces (see figure 2). Even up to a relatively high concentration of 50 μM there is a linear dependency of uv absorption and s-sortilin concentration. In addition, for the equilibrium plots only data with a uv absorbance smaller than 0.6 A for 250 nm and smaller than 1.0 A for 280 nm was taken into account which is well within the linear range of uv absorption.

Figure 2. Absorbance at three wavelengths as a function of the sortilin concentration (data taken from wavelength scans in AUC). Data collected in 12 mm centerpieces (at 2 μ M) were divided by a factor of 4 to compare with the 10 and 50 μ M concentrations. Note that the data collected at 300 nm was not used in the global analysis as including this did not influence the outcome of the results.

Interference optics is a very accurate system that works well to obtain data with low noise and this optical system is also preferable to use for higher concentrations (where the Lambert-Beer relation no longer holds). However, everything, including the solvent and the buffer components can contribute to the signal. An intensive dialysis is needed to make a reference sample that is exactly equal to the real sample apart from the proteins to be measured. In addition, any mechanical imperfection that causes shifts in the optical path lengths in the nm range or larger, will generate a signal. Mechanical ageing of the cell assembly may prevent these variations in path length and this works well for sedimentation velocity experiments. For sedimentation equilibrium experiments, however, it turns out that over longer times (order of weeks) minor changes in light path still occur. To prevent such problems we prefer absorbance measurements over interference measurements for sedimentation equilibrium experiments.

Q: How was thermodynamic non-ideality corrected for?

A: Thermodynamic non-ideality does probably not play a role in our AUC experiment. In particular at high concentrations (above 1 mg/mL for globular particles such as s-sortilin) thermodynamic non-ideality may play a role. We have verified that this is not the case for s-sortilin by comparing the results of the global analysis of all the data with that using only data below 1 mg/ml (i.e. up to an absorbance of 0.15 for 250 nm and 0.4 for 280 nm). Resulting differences in the K_D are less than 3 %, indicating that thermodynamic non-ideality does not play a significant role here.

Q: What was the partial specific volume and how was it determined?. Was glycosylation taken into account?

A: The partial specific volume of 0.729 mL/g was calculated based on the amino acid sequence of s-sortilin with the program SEDNTERP. The glycosylation was not taken into account. To clarify this in the manuscript we have added to the “Analytical Ultra Centrifugation” section in the methods the following: “The partial specific volume for s-sortilin of 0.729 mL/g is based on the amino acid sequence excluding the glycans and was determined with SEDNTERP”.

Minor comments

Q: 1. Protein names are conventionally lower case: Sortilin should be sortilin throughout (except where it is the start of a sentence), suggest sSortilin is s-sortilin.

A: To address this we have changed “Sortilin” to “sortilin” and we have changed “sSortilin” to “s-sortilin” throughout the manuscript.

Q: 2. Line 515 and following. The modern convention is MALS (multi-angle light scattering) not MALLS as it is taken as a matter of course that lasers are being used.

A: We have changed MALLS to MALS and “multi-angle laser light scattering” to “multi-angle light scattering” throughout the manuscript.

I hope these comments prove helpful

Stephen Harding

Reviewer #5 (Remarks to the Author):

In this manuscript authors present crystal structures of the luminal part of the transmembrane receptor Sortilin (sSortilin) at two different pH ranges; pH 7.0 – 7.5 (“neutral pH”) and pH 5.0 – 6.2 (“acidic pH”). At neutral pH, sSortilin predominantly exists in a monomeric form. At acidic pH, sSortilin changes conformation and forms a homodimeric species. These pH-induced changes in states were confirmed using a broad spectrum of biophysical techniques and the presence of the dimeric state was shown in cells using the proximity ligation assay (PLA).

The manuscript is well written and the authors have done a thorough analysis of the two different sSortilin forms observed at acidic and neutral pH. It is beyond doubt that sSortilin undergoes pH-induced conformational changes, which makes the model for ligand uptake at neutral pH and ligand release at acidic pH (e.g. within endosomes and lysosomes) very convincing. Also the dimerization of the sSortilin in acidic pH can provide a trigger for receptor recycling. These novel and attractive mechanisms could be also relevant for other receptors involved in internalization and endocytosis. Thus this study deserves publication, but there are a number of comments, that should be addressed before publication:

Q: 1. I would prefer to see RMSZ in the Table 1 instead of RMSD of bond length and angles, the Ramachandran outliers and rotamer outliers expressed in number of residuals instead of percentage. Also it is a good practice to add MolProbity score of each structure as the bottom row.

A: We have now included in table 1 RMSZ for the bond length and angle, the Ramachandran outliers and rotamer outliers expressed in number of residuals and the MolProbity score. We have also submitted updated structures to the protein data bank and have updated Tabel 1 accordingly.

Q: 2. I would advise the authors to change the coloring of their structure models within the figures; it is difficult to distinguish between the pink vs red colors used for monomer and dimer, respectively. Also the blue vs light-blue color are too similar to visualize the differences between monomer and dimer structure.

A: We have changed the pink vs red and the blue vs light-blue coloring of the structures throughout the manuscript for clearer representation.

Q: 3. *Figure 1 nicely illustrates the relative position of the sSortilin domains, also in respect to the Cell surface. If visually possible, it would be nice to add the binding site of Neurotensin, which is only detailed in Figure 7. Alternatively, a zoomed-out sSortilin + Neurotensin complex structure could be presented in Figure 7.*

A: We have added a zoomed-out representation of the s-sortilin – neurotensin complex to figure 7 (which is renumbered to figure 6 in the current version).

Q: 4. *Similar to point 3., the N-glycosylation sites (or at least N241 located at the dimer interface) could be indicated in one of the figures (e.g. Figure 1)*

A: We have now indicated the location of N241 in supplementary figure 2.

Q: 5. *For obtaining crystal forms 2, 3 and 4, sSortilin was initially co-crystallized with proneurotrophins (Material and methods; lines 484-486). However, these proneurotrophins are not mentioned in the crystallization conditions. This seems somewhat confusing; did sSortilin crystallize in the presence of proneurotrophin (which one(s)?) and the complex was not observed in the crystal structure?*

A: Crystal forms 2, 3 and 4 were grown in the presence of proneurotrophins (proNGF for crystal form 3 and proBDNF for crystal form 2). The complex was indeed not observed in the crystal structure. Apparently only the s-sortilin crystallized in these conditions. To clarify this, we have changed lines 484-485 from “Crystal forms 2, 3 and 4 (also see table 1) were initially set up as a complex with proneurotrophins” into “Crystal forms 2, 3 and 4 (also see table 1) were grown from a 1:1 molar ratio mixture of s-sortilin with proneurotrophins, but proneurotrophins were not present in the crystals” and we have added details of the proneurotrophins to the crystallization conditions.

Q: 6. *Figure 5A shows elution profiles for sSortilin in two different pH's, elution profiles for the sSortilin in the same condition also shown in Sup. Figure 2. However the retention volumes are very different in these two figures. I guess that the elution profiles in Figure 5A (and Figure 6A) are from Superose 6 10/30 column that was used in SEC-SAXS experiments and elution profiles in Sup. Figure 2 are from Superdex 200 10/300 used in SEC-MALLS experiments, but this is not mentioned in the main text or figure legends.*

A: s-Sortilin at pH 5.0 interacts with the Superdex 200 column as revealed by the delayed elution and tailing beyond the total column volume, this does not happen with the Superose 6 column. To clarify this, we have added to the figure legend of suppl fig 2 “s-Sortilin at pH 5.0 interacts with the Superdex 200 column as revealed by the delayed elution and tailing beyond the total column volume”. We have also added the column information to the figure legends of figure 4A (previously Fig. 6A) and suppl. figure 2.

Q: 7. *According to my knowledge, the native MS profile does not provide a direct quantitative information regarding abundances of different protein species, since the ionization efficiency of different protein species may not always be the same in different conditions. Also, ionized spray may disturb monomer-dimer equilibrium not in the same way at pH 5.0 and at pH 7.5. So the difference in MS profiles that shown in Figure 5G by themselves don't confirm “that Sortilin has a propensity to dimerize at acidic pH” (line 244-246). However this conclusion can be drawn based on comparison of MS profiles from the WT protein with the profiles from A464E mutant (Figure 6I).*

A: We agree with the reviewer that the ionization efficiency of different protein species may be different in different conditions, see also our reply to referee 1 (question: “Page 10, first paragraph: Native MS experiments of the mutant could also be used for quantification”). To address this we have removed the statement “that Sortilin has a propensity to dimerize at acidic pH” (line 244-246) from the text.

Q: 8. *Sup. Figure 3 shown the MS profile of sSortilin produced in HEK293-E cells that was purified and measured at pH 7.5 in contrast to the sSortilin produced in HEK293-ES (Figure 5G) and A464E mutant (Figure 6I). It is not clear for me why the protein was purified at pH 5.5 for a MS measurement at pH 7.5 (Figure 5G and 6I) and where is this purification procedure mentioned?*

A: We purified s-sortilin at pH 5.5 and subsequently measured at pH 7.5 to show s-sortilin dimerization is reversible, the reverse experiment was also done. We have since shown the reversible nature of dimerization with other methods such as AUC as well. We decided to only show the native MS data from the material purified at pH 5.5 as the reversibility is much better determined by the AUC analysis and as the s-sortilin does not form aggregates at pH 5.5. To clarify this, we have added to the “Protein expression and purification” section in the methods the following “For crystallization experiments, this was followed by size exclusion chromatography (SEC) on a Superdex 200 HiLoad 16/60 column (GE Healthcare) in 150 mM NaCl, 20 mM HEPES pH 7.0, for all other experiments the SEC was performed in 25 mM MES pH 5.5, 150 mM NaCl because in our hands s-sortilin does not form aggregate or precipitate at pH 5.5, while contaminants precipitate, forming a white powder on the side of the tube which is easily spun down and separated from pure s-sortilin which remains in solution.”.

For consistency, we have now performed the native MS experiment with s-sortilin produced in HEK293-E cells that was purified at pH 5.5 and measured at pH 7.5 similar to all other native MS experiments reported. We show this new experiment in suppl. figure 5 instead of the experiment of s-sortilin purified at pH 7.4 and measured at pH 7.5.

Q: 9. *It will be appreciated if authors will include MS profile of the sSortilin produced in HEK293-E cells that was purified at pH 5.5 and measured at pH 5 in Sup. Figure 3, as it was done for sSortilin produced in HEK293-ES (Figure 5G) and A464E mutant (Figure 6I).*

A: As requested we have now added this experiment in suppl. figure 5. As expected this experiment shows the same pattern as the experiment with s-sortilin produced in HEK293-ES cells, i.e. more dimers at pH 5.0 compared to pH 7.4.

Q: 10. *The distributions of the number of PLA events per cell in Figure 8C do not look like normal distribution to me. Did the authors try to test the normality of the distributions? If the distributions are not normal I would suggest to use non-parametric test (for example Mann–Whitney test) for proper sample comparison.*

A: The distribution for the wt sortilin is normal. The distribution for the A464E mutant is approximately normal. We decided to assume a normal distribution and use the student’s t-test. Note that applying the non-parametric Mann-Whitney test to the data also shows a significant difference between the wt sortilin and the mutant with $p < 0.01$. To address this we have added the following to “Fluorescence microscopy and in situ proximity ligation assay” in the methods section: “The distribution for the wt sortilin is normal. The distribution for the A464E mutant is approximately normal. Note that applying the non-parametric Mann-Whitney test to the data also shows a significant difference between the wt sortilin and the mutant with $p < 0.01$.”

Q: 11. *Lines 327-328 the authors state “The value that we determined for NGF-Sortilin interaction*

agrees with that of Nykjaer et al. and that of proNGF with Sortilin is similar to that of Feng et al". However in both references the affinity of the Sortilin to proNGF is much higher than the affinity to NGF. In the experiment presented by the authors it is other way around. The reason could be the different source of the proteins as authors suggested, or the experimental setup. It is hard to say without extra controls or alternative binding experiments.

A: Indeed, as the reviewer correctly puts forward, others have shown that proNGF has a higher affinity for sortilin compared to NGF whereas we find a higher affinity of NGF for sortilin compared to proNGF. To emphasise that we find this difference in contrast to what was reported earlier we have added to the second paragraph of the "s-sortilin homodimerization prevents ligand binding" section the following: "but in contrast to these earlier reports we find that NGF interacts with higher affinity to s-sortilin compared to proNGF" as in "The value that we determined for NGF-sortilin interaction agrees with that of Nykjaer et al. and that of proNGF with sortilin is similar to that of Feng et al., but in contrast to these earlier reports we find that NGF interacts with higher affinity to s-sortilin than proNGF does. The differences in affinity may come from differences in protein origin. We expressed and purified proNGF in HEK293 cells compared to proNGF produced in E. Coli and Sf9 cells, and we used mouse proteins instead of the human versions".

Q: 12. Weakened affinity of A464E mutant to the ligands at pH 5.0 compared to pH 7.4 authors interpreted as evidence that "dimerization of Sortilin at acidic pH prevents ligand binding" (lines 334-338). However the WT sSortilin at pH 5.0, according to the authors, interacts unspecific with the chip surface, and it is shown to be mostly in dimeric for at this pH. The A464E mutant from the other hand has substantial monomeric fraction in the acidic pH. So weakened affinity could be due to the fact that dimers interact in an unspecific way with the chip and that only monomeric fraction binds specifically to the ligands.

A: We agree with the referee that this is a possibility. To address this, we have added the following to this section: "The somewhat weakened ligand affinity for s-sortilin A464E at pH 5.0 compared to pH 7.4 may be due to the conformational change, the remaining albeit much reduced dimerization propensity, or both properties of s-sortilin A464E".

Q: 13. Authors didn't show the dissociation part of the SPR experiment so the release of the ligand in different pH's can't be evaluated.

A: The SPR experiment we performed is an equilibrium experiment that does not rely on collection of the dissociation part. The k_D is derived from fitting the SPR response at equilibrium for a range of analyte concentrations to a Langmuir binding model. We therefore did not collect the dissociation part and instead applied a regeneration buffer to rapidly dissociate the analyte from the ligand.

Q: 14. Taking in account comments 11-13 I don't think that these SPR experiments are adding any useful information to this manuscript.

A: Our SPR results show that the A464E mutant s-sortilin, that has less propensity to dimerize, is capable of binding ligands at acidic and at neutral pH whereas other have shown that wt sortilin only binds ligands at neutral pH and loses its affinity at acidic pH. To emphasise this, we have added the following to the SPR section in the results: "our data in combination with that of others that show drastically reduced wt sortilin – ligand interactions at acidic pH (Gustafsen, C. *et al.*, *Cell Metab.* **19**, 310–318 (2014); Petersen, C. M. *et al.* *EMBO J.* **18**, 595–604 (1999); Petersen, C. M. *et al.*, *J. Biol. Chem.* **272**, 3599–3605 (1997); Gustafsen, C. *et al.*, *J. Neurosci.* **33**, (2013); Conticello, S. G. *et al.*, *J. Biol. Chem.* **278**, 26311–26314 (2003)) indicate that dimerization and conformational change of sortilin at acidic pH prevents ligand binding".

Q: 15. Authors state that dimerization and conformational change provide a double mechanism for ligand release at acidic pH. However I don't think they show enough evidence for this statement. The conformational change at acidic pH seems to be sufficient, based on the presented data, to explain the ligand release at acidic pH.

A: The A464E mutant, that has a lower propensity to dimerize is still capable of binding the ligands NGF, proNGF and proBDNF at acidic pH. We believe we can infer from this data that also the dimerization plays a role in ligand release. To emphasise that this statement is a hypothesis we have changed the last sentence in the abstract to “this work may reveal a novel double mechanism for low pH-induced ligand release by endocytosis receptors” and we have changed in the discussion the sentence “Sortilin dimerization and conformational change provide a double mechanism for ligand release at acidic pH” to “Sortilin dimerization and conformational change may provide a double mechanism for ligand release at acidic pH”.

Q: 16. Lines 113-115 “In addition, the pH-induced dimerization brings the sSortilin C-termini and the cytosolic segments in close proximity which could provide the signal for cytosolic adaptor proteins to shuttle Sortilin to various intracellular compartments”. I assume that the meaning is that cytosolic segments come to the close proximity to each other due to sSortilin dimerization and this provide the signal for cytosolic adaptor proteins.

A: The referee is correct. To clarify this, we have changed this sentence to: “In addition, the pH-induced dimerization brings the sortilin cytosolic segments in close proximity of each other which could provide the signal for cytosolic adaptor proteins to shuttle sortilin to various intracellular compartments”.

REVIEWERS' COMMENTS:

Reviewer #1 (Remarks to the Author):

The authors have mostly answered my questions.

Reviewer #2 (Remarks to the Author):

Thank you to the authors for revising the SAXS sections of the manuscript so as to present the data and interpretation in a more scientific fashion. However, it remains the opinion of this reviewer that the authors do not adequately address – or explicitly state – the complications encountered when interpreting the SAXS data in terms of mixtures. Why not simply rewrite the SAXS section in the context of mixtures? Is there anything wrong with this approach if that is what the data is suggesting?

For example on p9 line 276 it is stated that:

"For wt s-sortilin, at pH 7.4, the elution peak is very broad and asymmetric. The SAXS signal varies significantly throughout the peak and both the radius of gyration and the estimated mass decrease at higher retention volumes. All other runs, i.e. s-sortilin at pH 5.5 and of s-sortilin A464E at pH 7.4 and pH 5.5, exhibit relatively narrow and more homogeneous peaks with stable scattering signal throughout the peak...the peak of wt s-sortilin at pH 5.5 can be identified as s-sortilin dimer and those of s-sortilin A464E at both pH values as s-sortilin monomer."

So are the authors actually stating is that the SEC-SAXS traces demonstrate that the averaged SAXS profiles presented in the main text are derived from components that have not been separated on the SEC column, especially for the wild-type protein? As a result, is it highly likely that the profiles are representative of the scattering from volume-fraction weighted monomer-dimer mixtures? So how does this impact the interpretation of single rigid-body models and single ab initio bead models as presented in the manuscript (and deposited into SASBDB)? Is it not the case that presenting single models of the monomer and the dimer in solution does not adequately represent what is going on in solution? Single models are nice to have, but in this case it maybe very dangerous to interpret the SAXS using a 'single model' view as such an approach can lead to quite bizarre conclusions.

For example, the authors state on page 10, line 290:

"The SAXS data indicate that the shapes of the s-sortilin dimers at pH 7.4 and pH 5.5 are not identical nor are the shapes of the s-sortilin monomers at pH 7.4 and pH 5.5. This suggests that s-sortilin can adopt four different conformations, with the monomer and dimer conformations at pH 7.4 different to the monomer and dimer conformations at pH 5.5."

Really? This is a bold claim. Take for example the SAXS data for the 'monomeric' form of the A464E mutant protein at pH 5.5 deposited into SASBDB (accession code: SASDCX5). It is straightforward to show that the pH 5.5 A464E 'monomer' scattering can, in fact, be described (reasonably well, $p = 0.001$; $\chi^2 = 1.3$) as a simple mixture of 76 % pH 7.4 A464E 'monomer' scattering (SASDCZ5) plus 25 % pH 5.5 dimer scattering (SASDCY5). Below/attached is the predicted $I(s)$ vs s for the mixture of SASDCZ5 plus SASDCY5 (yellow) compared to the SASDCX5 data (red).

A similar analysis shows that the pH 7.4 'dimer' (SASDCW5) can be described as 38% pH 7.4 A464E monomer scattering plus 62% pH 5.5 dimer scattering ($p = 0.007$; $\chi^2 = 0.75$). So are the differences in conformation as stated by the authors actually differences in conformation or, more likely, just different mixture composition(s)? This should be addressed in the manuscript.

The recommendation is to:

1) Revise and simplify the SAXS section, keeping it within the realms of what you can and cannot

tell from the data, without over-interpreting the profiles. GASBORmx should work – you even have an isoscattering point at around $s = 0.67$ inverse nm across all profiles suggesting some sort of common intermediate (another big sign mixtures are present!). Also, if you are going to model a dimer with GASBORMX, it might be best to use P2 symmetry? Model in parallel against all the data at the same time? Use say 730 amino acids in the modelling (native protein sequence plus 40ish additional residues to take into account the missing glycan mass)...etc.

2) Also include the sortilin-neurotensin SEC-SAXS data in SASBDB?

Reviewer #4 (Remarks to the Author):

The authors have addressed my comments - exhaustively - and I am happy to recommend acceptance

Reviewer #5 (Remarks to the Author):

The authors did very extensive and thorough work to answer the comments. The extra experiments, additional data and corrections made the manuscript more clear and precise. I have only two minor comments that should be addressed without my revision prior to publication.

Authors answer 1: We have now included in table 1 RMSZ for the bond length and angle, the Ramachandran outliers and rotamer outliers expressed in number of residuals and the MolProbity score. We have also submitted updated structures to the protein data bank and have updated Tabel 1 accordingly.

Comment: I can't find RMSZ in table 1.

Authors answer 10 : The distribution for the wt sortilin is normal. The distribution for the A464E mutant is approximately normal. We decided to assume a normal distribution and use the student's t-test. Note that applying the non-parametric Mann-Whitney test to the data also shows a significant difference between the wt sortilin and the mutant with $p < 0.01$. To address this we have added the following to "Fluorescence microscopy and in situ proximity ligation assay" in the methods section: "The distribution for the wt sortilin is normal. The distribution for the A464E mutant is approximately normal. Note that applying the non-parametric Mann-Whitney test to the data also shows a significant difference between the wt sortilin and the mutant with $p < 0.01$."

Comment: Authors don't need to assume normality of the distribution, they can test for it, using one of the normality tests. And taking in account that Mann-Whitney test shows significant difference the normality assumption is excessive.

Authors answer 13: The SPR experiment we performed is an equilibrium experiment that does not rely on collection of the dissociation part. The K_D is derived from fitting the SPR response at equilibrium for a range of analyte concentrations to a Langmuir binding model. We therefore did not collect the dissociation part and instead applied a regeneration buffer to rapidly dissociate the analyte from the ligand.

Comment: Dissociation equilibrium constant (K_D) is a ration of binding and dissociation rates, $K_D = k_{on}/k_{off}$. If dimer prevents bind the ligand, I would expect the binding rate to be slower in lower pH, since ligand needs more time to find the monomer to bind. From the other hand, if the conformational change is the reason for lower affinity (higher K_D), then most likely the dissociation rate will be faster in lower pH. If both dimerization and conformational change are important I would expect both rates to be different in different pHs. So the dissociation part of the SPR

experiment can provide insight to the mechanism of this interaction. However, since the authors now present a double mechanism for ligand release as an option, this part of experiment is not crucial for publication.

Point-by-point response to comments and suggestions on manuscript NCOMMS-17-02577A

Reply to reviewers' comments:

Reviewer #1 (Remarks to the Author):

The authors have mostly answered my questions.

Reviewer #2 (Remarks to the Author):

Thank you to the authors for revising the SAXS sections of the manuscript so as to present the data and interpretation in a more scientific fashion. However, it remains the opinion of this reviewer that the authors do not adequately address – or explicitly state – the complications encountered when interpreting the SAXS data in terms of mixtures. Why not simply rewrite the SAXS section in the context of mixtures? Is there anything wrong with this approach if that is what the data is suggesting?

For example on p9 line 276 it is stated that:

"For wt s-sortilin, at pH 7.4, the elution peak is very broad and asymmetric. The SAXS signal varies significantly throughout the peak and both the radius of gyration and the estimated mass decrease at higher retention volumes. All other runs, i.e. s-sortilin at pH 5.5 and of s-sortilin A464E at pH 7.4 and pH 5.5, exhibit relatively narrow and more homogeneous peaks with stable scattering signal throughout the peak...the peak of wt s-sortilin at pH 5.5 can be identified as s-sortilin dimer and those of s-sortilin A464E at both pH values as s-sortilin monomer."

So are the authors actually stating is that the SEC-SAXS traces demonstrate that the averaged SAXS profiles presented in the main text are derived from components that have not been separated on the SEC column, especially for the wild-type protein? As a result, is it highly likely that the profiles are representative of the scattering from volume-fraction weighted monomer-dimer mixtures? So how does this impact the interpretation of single rigid-body models and single ab initio bead models as presented in the manuscript (and deposited into SASBDB)? Is it not the case that presenting single models of the monomer and the dimer in solution does not adequately represent what is going on in solution? Single models are nice to have, but in this case it maybe very dangerous to interpret the SAXS using a 'single model' view as such an approach can lead to quite bizarre conclusions.

For example, the authors state on page 10, line 290:

"The SAXS data indicate that the shapes of the s-sortilin dimers at pH 7.4 and pH 5.5 are not identical nor are the shapes of the s-sortilin monomers at pH 7.4 and pH 5.5. This suggests that s-sortilin can adopt four different conformations, with the monomer and dimer conformations at pH 7.4 different to the monomer and dimer conformations at pH 5.5."

Really? This is a bold claim. Take for example the SAXS data for the ,monmeric' form of the A464E mutant protein at pH 5.5 deposited into SASBDB (accession code: SASDCX5). It is straightforward to show that the pH 5.5 A464E ,monomer' scattering can, in fact, be described (reasonably well, $p=0.001$; $\chi^2 = 1.3$) as a simple mixture of 76 % pH 7.4 A464E ,monomer' scattering (SASDCZ5) plus 25 % pH 5.5 dimer scattering (SASDCY5). Below/attached is the predicted $I(s)$ vs s for the mixture of SASDCZ5 plus SASDCY5 (yellow) compared to the SASDCX5 data (red).

A similar analysis shows that the pH 7.4 ,dimer' (SASDCW5) can be described as 38% pH 7.4 A464E monomer scattering plus 62% pH 5.5 dimer scattering ($p = 0.007$; $\chi^2 = 0.75$). So are the differences

in conformation as stated by the authors actually differences in conformation or, more likely, just different mixture composition(s)? This should be addressed in the manuscript.

The recommendation is to:

Q: 1) Revise and simplify the SAXS section, keeping it within the realms of what you can and cannot tell from the data, without over-interpreting the profiles. GASBORmx should work – you even have an isoscattering point at around $s = 0.67$ inverse nm across all profiles suggesting some sort of common intermediate (another big sign mixtures are present!). Also, if you are going to model a dimer with GASBORMX, it might be best to use P2 symmetry? Model in parallel against all the data at the same time? Use say 730 amino acids in the modelling (native protein sequence plus 40ish additional residues to take into account the missing glycan mass)...etc.

A: We thank the reviewer for his valuable suggestions and his analysis of the SAXS data. It is indeed possible that the SAXS data of A464E at pH 5.5 and the wt pH 7.4 dimer part represent a mixture of monomer and dimer protein. As the referee points out SAXS data for both experiments can be modelled as a mixture of A464E pH 7.4 monomer scattering and wt pH 5.5 dimer scattering. There is however no peak shift apparent between the A464E pH 5.5 and pH 7.4 SEC peaks and both peaks are symmetrical and lack any shoulders. Assuming about 25% dimer is present in the A464E pH 5.5 data, and no dimer in the pH 7.4 data a peak shift or asymmetric peak shape would be expected. To address this ambiguity and follow the reviewers recommendation we have included the monomer-dimer OLIGOMER analysis in the manuscript as follows: "This either suggests that s-sortilin can adopt four different conformations, with the monomer and dimer conformations at pH 7.4 different to the monomer and dimer conformations at pH 5.5 or, alternatively, that the s-sortilin dimer at pH 7.4 contains some monomer and the A464E monomer at pH 5.5 contains some dimer. The A464E pH 5.5 monomer SAXS data can be described reasonably well as a mixture of 77 % A464E pH 7.4 monomer scattering plus 23 % wt pH 5.5 dimer scattering ($p = 0.0074$, A464E pH 7.4 monomer scattering plus 61 % wt pH 5.5 dimer scattering ($p = 0.007$,"

To simplify the SAXS section (and to condense the manuscript as requested by the editor) we have shortened and moved the DAMMIF and CORAL based analysis of the SAXS data to supplementary Fig. 8. In addition to reflect that there may be ambiguity in the data, we have pointed out that some of the data can be described as arising from a mixture of monomer and dimer s-sortilin.

We followed the reviewer's suggestion and attempted modeling the set of four SAXS curves with GASBORmx. The modelling eventually converged for all datasets except for A464E pH 7.4 (for the A464E pH 7.4 data the χ^2 of the solution is very high, about 1000, and thus the model does not represent the data). There were also distinct deviations, in particular at small angles, to the extent that for all datasets, except the A464E pH 7.4 the CORMAP derived p-value is 0 ($p=0$). This indicates a fit with substantial systematic deviations. In addition, even with several runs it was not possible to obtain models that did not interlace and the models are not representative of what can be expected of a protein.

Q: 2) Also include the sortilin-neurotensin SEC-SAXS data in SASBDB?

A: The neurotensin data have now been submitted to the SASDB: SASDCE7 (monomeric s-sortilin at pH 7.4 in the presence of neurotensin) and SASDCF7 (dimeric s-sortilin at pH 7.4 in the presence of neurotensin). Note that a new project was created for this. This was accordingly indicated in the "Data availability section" in the manuscript.

Reviewer #4 (Remarks to the Author):

The authors have addressed my comments - exhaustively - and I am happy to recommend acceptance

Reviewer #5 (Remarks to the Author):

The authors did very extensive and thorough work to answer the comments. The extra experiments, additional data and corrections made the manuscript more clear and precise. I have only two minor comments that should be addressed without my revision prior to publication.

Authors answer 1: We have now included in table 1 RMSZ for the bond length and angle, the Ramachandran outliers and rotamer outliers expressed in number of residuals and the MolProbity score. We have also submitted updated structures to the protein data bank and have updated Tabel 1 accordingly.

Q: Comment: I can't find RMSZ in table 1.

A: This is indeed an error from our side and we thank the referee for noting this. We have now updated the crystallographic table with the requested RMSZ values.

Authors answer 10 : The distribution for the wt sortilin is normal. The distribution for the A464E mutant is approximately normal. We decided to assume a normal distribution and use the student's t-test. Note that applying the non-parametric Mann-Whitney test to the data also shows a significant difference between the wt sortilin and the mutant with $p < 0.01$. To address this we have added the following to "Fluorescence microscopy and in situ proximity ligation assay" in the methods section: "The distribution for the wt sortilin is normal. The distribution for the A464E mutant is approximately normal. Note that applying the non-parametric Mann-Whitney test to the data also shows a significant difference between the wt sortilin and the mutant with $p < 0.01$."

Q: Comment: Authors don't need to assume normality of the distribution, they can test for it, using one of the normality tests. And taking in account that Mann-Whitney test shows significant difference the normality assumption is excessive.

A: To address this comment we have now indicated in the manuscript that we are only using the Mann-Whitney test and not the Student's t-test.

Authors answer 13: The SPR experiment we performed is an equilibrium experiment that does not rely on collection of the dissociation part. The K_D is derived from fitting the SPR response at equilibrium for a range of analyte concentrations to a Langmuir binding model. We therefore did not collect the dissociation part and instead applied a regeneration buffer to rapidly dissociate the analyte from the ligand.

Comment: Dissociation equilibrium constant (K_d) is a ration of binding and dissociation rates, $K_d = k_{on}/k_{off}$. If dimer prevents bind the ligand, I would expect the binding rate to be slower in lower pH, since ligand needs more time to find the monomer to bind. From the other hand, if the conformational change is the reason for lower affinity (higher K_d), then most likely the dissociation rate will be faster in lower pH. If both dimerization and conformational change are important I would expect both rates to be different in different pHs. So the dissociation part of the SPR experiment can provide insight to the mechanism of this interaction. However, since the authors now present a double mechanism for ligand release as an option, this part of experiment is not crucial for publication.